# GraphMERT: Efficient and Scalable Distillation of Reliable Knowledge Graphs from Unstructured Data

**Margarita Belova**                                            *margarita.bel@princeton.edu*
*Department of Electrical and Computer Engineering, Princeton University*

**Jiaxin Xiao**                                                    *jx0800@princeton.edu*
*Department of Electrical and Computer Engineering, Princeton University*

**Shikhar Tuli**                                          *stuli@alumni.princeton.edu*
*Department of Electrical and Computer Engineering, Princeton University*

**Niraj K. Jha**                                                      *jha@princeton.edu*
*Department of Electrical and Computer Engineering, Princeton University*

**Reviewed on OpenReview:** *https://openreview.net/forum?id=tnXSdDhvqc*

## Abstract

Researchers have pursued neurosymbolic artificial intelligence (AI) applications for nearly three decades. A marriage of the neural and symbolic components can lead to rapid advancements in AI. Yet, the field has not realized this promise since most neurosymbolic AI frameworks fail to scale. In addition, the implicit representations and approximate reasoning of purely neural approaches limit interpretability and trust. Knowledge graphs (KGs), a gold-standard representation of explicit semantic knowledge, can address the symbolic side of the problem. However, automatically deriving reliable KGs from text corpora remains an open problem. We address the above challenges by introducing GRAPHMERT, a tiny graphical encoder-only model that distills high-quality KGs from unstructured text corpora and its own internal representations. Together, GRAPHMERT and its equivalent KG form a modular neurosymbolic stack: neural learning of abstractions; symbolic KGs for verifiable reasoning. GRAPHMERT + KG is the first efficient and scalable neurosymbolic model to achieve state-of-the-art benchmark accuracy along with superior symbolic representations relative to baselines. More concretely, we target *reliable* domain-specific KGs that are both (1) factual (with provenance) and (2) valid (ontology-consistent relations with domain-appropriate semantics). When an off-the-shelf large language model (LLM), e.g., Qwen3-32B, generates domain-specific KGs, it falls short on the reliability front due to prompt sensitivity, shallow domain expertise, and hallucinated relations. Thus, practitioners should avoid employing LLM-generated KGs in high-stakes domains, e.g., medicine, law, business, education, etc. On text obtained from PubMed papers related to diabetes, our KG extraction pipeline with a small 80M-parameter GRAPHMERT yields a KG with a 69.8% FActScore; a 32B-parameter baseline LLM yields a KG that achieves only a 40.2% FActScore. The GRAPHMERT-extracted KG also achieves a significantly higher ValidityScore of 68.7%, compared to an LLM-generated baseline (43.0%), demonstrating its ability to preserve ontology alignment. KG cleaning further improves factuality, with GRAPHMERT reaching 76.9% FActScore, compared to 55.6% for the LLM baseline. GRAPHMERT can then treat the augmented KG as the seed KG and refine it further. Finally, human experts can edit and audit the extracted KGs, further increasing their reliability. This is nearly impossible with purely neural representations. Hence, GRAPHMERT enables efficient, scalable, transparent (interpretable and explainable), attributable (with provenance), accountable (with governance), editable, auditable, and continually improvable state-of-the-art neurosymbolic AI. The code is available at https://github.com/jha-lab/graphmert_umls.

*Index Terms*: Hallucinations, interpretability, knowledge graphs, language models, neurosymbolic methods, retrieval-augmented generation.

# 1 Introduction

Artificial intelligence (AI) has long oscillated between two dominant paradigms: symbolic reasoning and neural learning (Shavlik et al., 1991; d'Avila Garcez et al., 2002). Symbolic systems excel at explicit (rule-based) inference, providing interpretability and strong exact reasoning. This assumes precise and consistent symbolic abstractions. However, such systems struggle with noisy or ambiguous data. Neural approaches, by contrast, thrive on large-scale and general-purpose pattern recognition, often outperforming hand-coded explicit representations (Sutton, 2019). Nevertheless, neural networks operate as black boxes, offering little transparency in their decision-making (Sharkey et al., 2025). Their representations are approximate: ambiguous, difficult to control, and not grounded in explicit rules. Thus, each paradigm, taken alone, has critical gaps. Neurosymbolic AI is a synthesis of the two paradigms, aiming to combine the flexibility of neural models with the rigor and interpretability of symbolic systems. By uniting these complementary strengths, it opens up a path toward systems capable of both scalable learning and sound reasoning — a longstanding ambition of the field (Garcez & Lamb, 2023; Towell, 1994; d'Avila Garcez et al., 2019).

Large language models (LLMs) have generated enormous excitement, but their reasoning is ultimately probabilistic and often unable to perform causal inference, opaque to humans, and prone to hallucinations, especially during multi-step reasoning (Marcus, 2018; Ji et al., 2023). Their opaqueness raises serious concerns about whether such models can be trusted (von Eschenbach, 2021; Alzubaidi et al., 2023). LLMs trained on general text corpora may also fail to adapt to specialized domains or incorporate new knowledge without undergoing expensive retraining (Zhao et al., 2025). These limitations highlight the need for external, explicit sources of factual grounding in high-stakes use cases (**?**).

Unifying knowledge graphs (KGs) with LLMs could help overcome some of these limitations, as KGs offer a natural complement (Ibrahim et al., 2024). By encoding knowledge in a structured, symbolic form of head-relation-tail triples with explicit and verifiable relations, KGs offer interpretability, auditability, and domain-specific depth that LLMs lack (Pan et al., 2023a). Thus, KGs can guide LLM inference and enable robust evolution of background knowledge, while LLMs contribute flexible reasoning, efficient handling of ambiguity through approximate inference, and serve as a natural language interface (Pan et al., 2023b). This synergy can enable KG-guided exploration and learning, support agentic workflows driven by interaction with an editable knowledge base, and enhance the trustworthiness of LLMs in high-stakes application domains by reinforcing factuality and enabling immediate knowledge updates.

Yet, constructing a KG from scratch in a new domain is a notoriously arduous task (Zhong et al., 2023; Ji et al., 2022). The process typically involves cleaning and pre-processing heterogeneous data, multi-step knowledge acquisition, and post-processing. Ensuring that the resulting KG is reliable and factual is even more challenging, often requiring extensive manual inspection and crowdsourcing (Wang et al., 2021). In fields where no ground-truth KG or domain-specific benchmarks exist, this task becomes infeasible without expert intervention. This makes existing approaches unscalable.

Given these obstacles, we propose characteristics that an effective KG construction method should provide:

1. **Factuality and provenance:** Triples grounded in source text with verifiable citations.

2. **Validity:** Adherence to ontological schema constraints, appropriate term granularity, and domain-correct relations.

3. **Automation:** End-to-end extraction without expert oversight; usable by non-specialists.

4. **Scalability:** Robust performance as data volume and compute grow.

5. **Domain generality:** Principles that transfer across subject areas.

6. **Global integration:** Cross-document linking of concepts, not just within local spans.

We refer to KGs that are both factual and valid as *reliable*; hence, we classify methods that produce factual and valid triples as reliable.

Recently, KG generation with decoder-based LLMs has become the most widely used method (Xu et al., 2024), especially for commonsense knowledge extraction (West et al., 2022). Nonetheless, we demonstrate that LLMs are unable to construct reliable domain-specific KGs. The factuality challenge faced by LLMs intersects with several other pressing issues, among them hallucinations, outdated knowledge, and lack of depth in given domains (e.g., in health, law, and finance applications) (Wang et al., 2025a).

Despite their ease of use, versatility, broad knowledge coverage (Minaee et al., 2025), LLMs as KG constructors underperform on domain-specific datasets (Zhu et al., 2024) and are susceptible to domain-irrelevant noise in the context (Chen et al., 2024) and hallucinations (Ghanem & Cruz, 2025), often resulting in inclusion of fabricated facts (Huang et al., 2025a). Since LLMs are oblivious to their training sources (Khalifa et al., 2024), KG distillation from weights of an LLM obfuscates knowledge provenance (Pan et al., 2023a). Fine-tuned models exhibit enhanced accuracy and extraction fidelity in the domains they are adapted to. However, fine-tuning requires labeled training data and negatively impacts generalization in heterogeneous domains (Ghanem & Cruz, 2025). Finally, KG extraction from text corpora with off-the-shelf LLMs is not global in the sense that the extracted triples are confined to a single text chunk presented in the context window.

The current approaches fail to meet all six requirements for KGs we listed above (Hofer et al., 2024). A method that satisfies all six could unlock numerous downstream applications. However, a reliable KG can be extracted only if the data are of high quality (Rejeleene et al., 2024; Geiger et al., 2020; Wang et al., 2023). Yet, such data are scarce. This leads us to the central question:

*How can we build a reliable domain-specific KG from limited high-quality sources?*

To address this challenge, we propose a novel framework, GRAPHMERT (Graphical Multidirectional Encoder Representations from Transformers), for reliable KG extraction from small high-quality domain-specific data, in domains where no large expert-annotated text corpora for triple extraction is available. GRAPHMERT relies on an encoder-only transformer that distills a symbolic representation from its weights. It jointly learns cross-modal representations: semantic — from a small expert-curated initial seed KG, and syntactic — from unstructured sentence-level text by minimizing the standard masked language modeling (MLM) and the proposed masked node modeling (MNM) losses. We automatically extract a KG that captures factual knowledge by training GRAPHMERT on high-quality texts with a small seed KG (e.g., 100+ triples per relation). The framework is domain-agnostic and only requires a seed KG along with a small, high-quality, domain-specific dataset (e.g., $\sim$ 100M tokens). To the best of our knowledge, *GRAPHMERT-powered KG extraction is the first framework that possesses all six characteristics of an effective KG mentioned earlier:*

1. Factuality and provenance: We implement triple extraction at the sentence level. One can trace back each triple to its source sequence, thus supporting knowledge provenance. FActScore, which quantifies the factuality of a KG (more details in Sec. 6.2), for the GRAPHMERT-generated KG (69.8%) is much higher than that of an LLM-generated one (40.2%).

2. Validity: The resulting KG preserves the relation usage patterns imposed by the ontological structure of the seed KG, which enhances the validity of the relations in the extracted KG relative to the LLM-extracted baseline KG. ValidityScore, which quantifies the ontological alignment of a KG (more details in Sec. 6.2), for our KG (68.8%) is much higher than that of the baseline (43.0%).

3. Automation: It does not need manual feature selection, rule handcrafting, or human experts in the loop. It leverages a neural-to-symbolic converter, i.e., GRAPHMERT, to automatically and reliably generate a KG.

4. Scalability: We obtain training data only from credible sources and the compact GRAPHMERT (with just 80M parameters) eliminates the need for pretraining on large unverified text, making the approach much more practical than employing expensive LLMs (with billions or trillions of parameters). It can be scaled when provided with more data and given extra compute resources.

5. Domain generality: It relies on domain-agnostic principles. We do not hard-code any domain-specific parameters to the proposed GRAPHMERT pipeline.

6. Global integration: It can connect global concepts across the whole dataset throughout training, in contrast to extracting disconnected information from isolated text.

The rest of the article is organized as follows. In Sec. 2, we review different KG extraction techniques and motivate the need for a reliable KG. In Sec. 3, we provide a brief motivational example. In Sec. 4, we give a detailed overview of our proposed GRAPHMERT framework and its architecture. In Sec. 5, we describe the experimental setup. In Sec. 6, we provide experimental results. In Sec. 7, we discuss the limitations of our methodology and discuss future work. Finally, we conclude in Sec. 8.

## 2 Background and Related Work

In this section, we review prior research that is relevant to our work, including applications of neurosymbolic AI and KGs. We then examine existing KG extraction methods, their limitations, and how our framework addresses these gaps. Finally, we provide the technical background on graph transformer architectures that is necessary to understand the remainder of this work.

### 2.1 Structured Representation Learning

The paradigm of extracting interpretable or geometric structures from data and subsequently leveraging them for downstream tasks is a foundational strategy in AI. While modern deep learning often relies on end-to-end processing where intermediate representations remain opaque (Bengio et al., 2014), a growing body of literature advocates a two-stage paradigm in which structured representations are first derived from raw inputs and then employed to support downstream reasoning or decision-making (Battaglia et al., 2018; Bronstein et al., 2021). This method first distills raw inputs into structured, often interpretable, representations, which are then used to guide reasoning or decision-making.

In the visual domain, research demonstrates that explicitly extracting prototypical parts or geometric structures is crucial for robust performance (Chen et al., 2019; Kadkhodaie et al., 2024). This strategy has proven equally vital in natural language processing (NLP). For instance, dependency trees are often extracted to guide graph neural networks, enabling them to capture long-range semantic dependencies for relation extraction (Zhang et al., 2018). Similarly, rationale extraction frameworks enforce interpretability by first selecting relevant text segments before making a prediction, ensuring decisions are grounded in explicit evidence (Lei et al.). In the realm of complex reasoning, neuro-symbolic methods retrieve and use subgraphs from external knowledge bases to augment language models, effectively grounding their answers in structured facts (Yasunaga et al., 2021). Furthermore, in causal inference, discovering the underlying causal graph is often a prerequisite for correctly estimating treatment effects and ensuring that models remain robust to distribution shifts (Schölkopf et al., 2021).

This discussion converges on a central insight: Explicit structure improves robustness, interpretability, and reasoning. KGs, in particular, provide a symbolic representation that captures rich semantic relations across text, enabling broad applications (Appendix A).

### 2.2 Knowledge Graphs

A KG $\mathcal{G} = (V, E)$ can be viewed as a directed graph where nodes $V$ represent real-world entities and directed edges $E \subseteq V \times V$ represent relationships between them (see Fig. 1 for a toy example). Each directed edge $e = (u, v) \in E$ connects two nodes $u, v \in V$ and encodes a relationship $r$ between the corresponding entities. Semantically, a KG can be thought of as a set of triples $\mathcal{G} = \langle h, r, t \rangle = \langle head, relation, tail \rangle$, where head and tail denote two KG entities connected by a directed relation.

### 2.3 The Representation Dilemma: Neural or Symbolic?

Traditional AI research overwhelmingly associates reasoning with purely symbolic systems, including expert systems (van Melle, 1978; Lindsay et al., 1993) or logic-based AI (McCarthy, 1980; Colmerauer & Roussel, 1996). For decades, this paradigm shaped AI practice under the premise that human intelligence could be

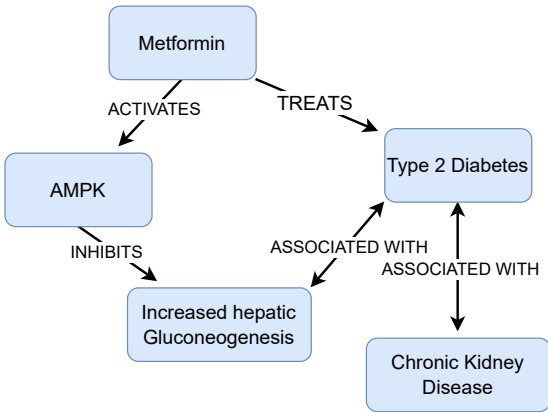

Figure 1: A toy KG example from the medical domain.

reduced to formal logic operating on symbols (Newell & Simon, 1976; Haugeland, 1985). Symbolic methods offer clarity and structure by explicitly encoding rules over discrete concepts (Newell & Simon, 1976) and are reliable, given suitable abstractions. The symbolic approach governed the AI field till the 90s, when its drawbacks became evident: Symbolic systems struggle with ambiguity, contextualization, and the fluidity of real-world knowledge (Harnad, 1990). In addition, computational complexity limits scalability of systems that are already prone to brittleness: Complete symbolic grounding of a knowledge base leads to a worst-case combinatorial explosion (Chen & Suen, 1994).

In contrast, neural approaches rely on multidimensional embeddings as their representations and approximate knowledge grounding through gradient-based learning over a continuous parameter space (LeCun et al., 2015). They are robust against outliers and inaccuracies in data, and scale learning and inference well (Hitzler et al., 2022). Neural systems are efficient learners but forfeit transparency: Their decision pathways remain opaque and lack the verifiable interpretability of symbolic inference (Tran et al., 2025). They tend to memorize, thus undermining reliable generalization beyond observed facts, especially in out-of-distribution domains. Furthermore, while probabilistic and approximate inference accommodates ambiguities, it yields imprecise logical inference.

The central question remains: Which representation best fits a given task? Due to these complementary advantages and limitations, researchers are increasingly focused on neurosymbolic integration (Tran et al., 2025) for complex AI challenges that cannot be solved by either approach individually. To bridge this divide, KGs serve as a natural interface that connects implicit neural learning with explicit symbolic reasoning.

Building on this perspective, we next illustrate how KGs function as complementary infrastructure in sensitive domains. In sensitive domains, auditable and editable KGs can serve as a persistent knowledge base: facts can be inspected and updated directly (Pan et al., 2023a). LLMs, by contrast, embed knowledge implicitly in parameters, making tracing and verification difficult (Akyurek et al., 2022). Updating LLMs is resource-intensive, requiring fine-tuning or retrieval-augmented generation (RAG), and resolving contradictions demands complex methods like context-aware decoding. Verification of generated content against a factual KG can help reduce the risk of inaccuracies or hallucinations (Luo et al., 2024; Hron et al., 2024). Furthermore, a KG offers the "ability to forget": Information can, in principle, be erased if required by legal regulations or upon user request. By contrast, removing knowledge from LLM requires complex interventions and sophisticated strategies that risk catastrophic unlearning (Si et al., 2023), raising concerns about access to harmful content, user privacy, and copyright violations (Tian et al., 2024). However, since KGs themselves can be incomplete or biased, their usage in critical fields requires careful construction and maintenance.

For an extended discussion on how KGs unify these representations, and an overview of their applications in neurosymbolic frameworks, we refer the reader to Appendix A.

### 2.4 Existing KG-extraction Methods

Next, we briefly review existing KG-extraction methods. For further details, we direct the reader to a comprehensive review on KG construction (Zhong et al., 2023). We discuss the following categories: (1) task-specific NLP methods, (2) triple embedding-based, and, the most recent, (3) generative, or LLM-based, which we discuss in greater detail, given its current prominence.

Given the growing attention to LLM-driven KG construction (Zhu et al., 2024), we further differentiate *KG extraction* from *KG generation*, to clarify their respective advantages, limitations, and application scopes. We classify methods where an LLM plays a pivotal role in the KG construction pipeline, either generating triples conditioned on input texts or distilling knowledge from model weights, under the category of KG generation. We refer to all other approaches as KG extraction.

#### 2.4.1 Task-specific NLP Methods Scale Badly

Early rule-based information extraction systems demanded heavy feature engineering and domain expertise. Modern pipelines sequentially chain machine learning components—named entity recognition, coreference resolution, and relation extraction (Jaradeh et al., 2023)—often relying on structured or semi-structured data (Hofer et al., 2024). These systems require sophisticated text preprocessing heuristics (Rao et al., 2024), e.g., as in the case of conditional random fields. Traditional fine-tuned BERT for relation extraction or sequence-to-sequence generation relies on large labeled corpora. As a consequence, it requires substantial data annotation in each new domain. Long short-term memories and convolutional neural networks introduce locality bias. Another drawback is that errors propagate over the pipeline. Overall, these methods can be accurate, but are very labor-intensive, not fully automatic, and hard to scale.

#### 2.4.2 Triple Embeddings are Local, Closed-domain, and Miss Long-range Dependencies

An embedding-based approach seeks to train ML methods on KGs to capture semantic and structural patterns of the graph into embeddings by optimizing a scoring function. Embeddings enable the model to predict missing links (triple completion) and estimate the likelihood of new relations (link prediction; Cheng et al. 2024). However, this approach suffers from several limitations, including selection bias, lack of scalability, brittleness to KG errors, and limited external/world knowledge (Xia et al., 2025). Because most KG embedding models operate on local triple patterns, they struggle to compose long multi-hop chains, handle negation, or respect ontological constraints — in particular, when relations are n-ary, qualified (e.g., temporal, provenance), or context-dependent. They also assume a largely closed-world, static graph: Cold-start entities/relations and evolving KGs typically require expensive retraining or ad-hoc heuristics, and performance degrades under distribution shift. More concretely, embedding-based approaches face the following limitations.

**Sparsity, limited information, and vocabulary:** The scale of the largest publicly available KGs, e.g., Wikidata (118M+ entities), PubGraph (385M+ entities; Ahrabian et al. 2023), is of the order of $10^8$ entities. The scale is incompatible with text corpora sizes: In 2024, top-tier LLMs reported up to $10^{13}$-token training datasets (Villalobos et al., 2024), and the pretraining corpora of leading projects can collectively surpass 700TB (Liu et al., 2024b).

**Insufficient utilization of semantic information:** Learning an embedding representation that incorporates equally good graph structural and semantic information remains challenging. Multiple efforts are targeted at developing architectures and approaches that produce embeddings that are not overly localized (Rao & Wang, 2023), incorporate multiple relation types (Lu et al., 2023), and better integrate contextual (including semantic) information for improved reasoning (Shi et al., 2025). This suggests that embedding methods are useful in task-specific applications based on KGs, but on their own, fall short in extending KGs.

**Generalizability:** Embedding methods do not generalize well across different KGs. Each KG is characterized by its own set of relations, attributes, and ontology, making it impossible or impractical to unite many KGs for training. In practice, embeddings entangle schema-specific signals (relation vocabularies, type

hierarchies, qualifier formats). Hence, representations learned on one KG transfer poorly to another with different ontologies or naming conventions. Cross-KG use then requires costly alignment steps — entity linking, relation mapping, and negative-sampling redesign — and even after alignment, out-of-vocabulary entities/relations and schema drift often degrade performance (Chen et al., 2023b).

### 2.4.3 LLM-based KG Generation is Not Reliable

With the tremendous success of LLMs on all kinds of NLP tasks, modern research on KG extraction is skewed towards extracting relational knowledge from LLM weights using prompts (Zhu et al., 2024; Carta et al., 2023; Li et al., 2025; Gupta et al., 2025). The widespread adoption of LLMs can be attributed to their versatility across tasks, adaptability to diverse domains, and ease of use, which together make them attractive as general-purpose AI systems. LLMs capture relational knowledge unevenly: They are more accurate for some types (e.g., commonsense or hierarchical 'is-a' links) and less accurate for others (e.g., detailed encyclopedic facts or multi-hop traversal relations; Pan et al. 2023a). In addition, a range of inherent drawbacks raises concerns about the use of LLMs for reliable domain-specific KG generation. The drawbacks we discuss next appear to be intrinsic to generative methods of KG construction.

**LLMs are brittle with respect to prompts:** Instruction fine-tuning does not fully address this problem (Zhou et al., 2024). KG extraction with prompts is biased towards prompt structure (Cao et al., 2021). LLMs are sensitive to task-framing: answer consistency can shift with small syntactic changes (Hagström et al., 2023), with slight prompt variations (Mousavi et al., 2024; Wang et al., 2024a), and even with respect to basic logical constraints (Ghosh et al., 2025). Retrieval augmentation can mitigate inconsistency and knowledge-cutoff issues (Shuster et al., 2021), but introduces new failure modes: conflicts between retrieved evidence and parametric knowledge of an LLM (Zhang et al., 2025; Zeng et al., 2025), imperfections in retrieval and ranking (Jin et al., 2024), and weak relevance estimation, resulting in incorrect utilization of the retrieved knowledge (Wang et al., 2024e).

**LLMs hallucinate:** Beyond reproducibility, LLMs hallucinate outputs that are nonsensical or unfaithful to the provided source content (Ji et al., 2023; Zhang et al., 2024; Li et al., 2024). Hallucinations persist regardless of model size or training data scale (Shuster et al., 2021). The current consensus is that hallucinations are an inherent, unavoidable property of LLMs. Some scholars formally show that, from the theoretical viewpoint, hallucinations may be innate to probabilistic generative methods (Xu et al., 2025). All methods to strengthen LLM reasoning, including chain-of-thoughts with self-consistency, fine-tuning, augmented generation, and greedy decoding, improve accuracy, but are unable to eliminate nontrivial hallucinations (Kim et al., 2025). In the context of KG generation, LLMs struggle with inverse inference (the "reversal curse"): They may learn ⟨A, is-a, B⟩ yet fail to infer the inverse relation (Berglund et al., 2024), undermining triple extraction in domains where inverse relations are common and semantically critical.

**LLMs are not factually accurate:** Factuality and hallucination are distinct: Factuality errors occur when a model fails to learn or apply factual knowledge accurately, whereas hallucinations are ungrounded or unfaithful content relative to the provided source (Wang et al., 2024f). As demonstrated in our motivating example (Sec. 3), even state-of-the-art models can return factually divergent answers to equivalent queries. Because models acquire factual knowledge during pretraining and can add more via continued pretraining (Chang et al., 2025), rigorous dataset cleaning is essential. Yet, verifying or synthesizing high-quality data at the LLM scale is infeasible in practice, creating a fundamental size-quality trade-off (see Sec. 2.5). No current LLM offers the factuality needed for trust. Even the most advanced commercial systems make significant factual errors, which spike on rare entities and remain less factual than humans (Min et al., 2023), with mistakes persisting even in search-augmented tools (Mehrotra & Marchman, 2024). Consequently, any method that relies on LLMs needs robust verification. In high-stakes domains (e.g., medicine (Thirunavukarasu et al., 2023), autonomous driving, aeronautics, and cybersecurity), verification alone is insufficient: Outputs must also be interpretable and explainable for decision-makers. However, LLMs provide little insight into their decision rationale (Madsen et al., 2024; Ye & Durrett, 2022; Bowman, 2023), falling short of the guarantees these applications demand.

## 2.5 Impact of Data Quality and Dataset Size

Prior work shows that state-of-the-art LLM capabilities emerge only when model size, dataset scale, and compute reach sufficient magnitude (Brown et al., 2020; Kaplan et al., 2020; Wei et al., 2022). As a result, modern pretraining corpora often favor scale over domain fidelity. Given LLM propensity to memorization, flawed data sources with misinformation and biases are a primary driver of hallucinations (Huang et al., 2025b). Domain adaptation with fine-tuning (Hu et al., 2022) can improve factuality and coherence, but risk catastrophic forgetting and cross-domain interference (Wang et al., 2024d); continued pretraining adapts knowledge more smoothly to a target domain, but demands substantial additional data. In medicine, researchers warn that scarcity of diverse, high-quality data at the scale required by LLMs leads to a "garbage in, garbage out" dynamic and remains a key barrier to clinical LLM deployment (Thirunavukarasu et al., 2023). Compounding this, large-scale public text corpora lack established cross-verification mechanisms. Privacy and copyright concerns restrict access to private datasets for training purposes (Pereira et al., 2022; Wang et al., 2024b). In cases where models are trained on closed-source training data that are not accessible for scrutiny, the lack of transparency blocks the public's ability to conduct a thorough investigation (Nguyen et al., 2024).

As a response to the limited availability of verified sources (Gandhi et al., 2024; Wang et al., 2024c) and unaffordable training costs, a growing line of work seeks to maximize LLM performance with less data, emphasizing the quality-quantity trade-off. The post-GPT-3 trajectory of the NLP field (Brown et al., 2020) prioritized ever-larger unlabeled web corpora, arguably underweighting data quality. Recent efforts formalize data-quality criteria (Zheng et al., 2025) and show that small, high-quality datasets can outperform substantially larger but unvalidated corpora (Iskander et al., 2024); likewise, compact models trained on carefully curated, domain-specific text sometimes surpass frontier general-purpose models on in-domain tasks (Kadosh et al., 2024; Chen et al., 2023a).

Our approach also advocates for "data quality first": We contend that high-quality data — not the sheer volume of data — is crucial to creating a reliable KG.

## 2.6 Integrating Knowledge Graphs with LLMs

While LLMs excel at fluent generation, they are limited by their reliance on implicit parametric knowledge, which can lead to hallucinations and obsolescence (Tonmoy et al., 2024). In contrast, KGs provide explicit, deterministic representations of factual information but lack generative flexibility and adaptability (Yang et al., 2025). Recent research has, therefore, focused on unifying these complementary paradigms, primarily through KG-enhanced pre-training and fine-tuning, as well as RAG.

### 2.6.1 KG-enhanced Pre-training and Fine-tuning

Early approaches sought to internalize structural knowledge directly into the weights of the model. K-BERT explicitly injects domain knowledge by stitching KG triples into the sentence constituent tree (Liu et al., 2020). Subsequent work DRAGON (Yasunaga et al., 2022) advances this by employing a cross-modal encoder that enables information to flow bidirectionally between text segments and KG subgraphs during pre-training. More recent studies demonstrate that domain-specific KGs can also be leveraged during fine-tuning to enhance LLM performance and factual consistency (Kamel et al., 2025; Jiang et al., 2024; Chen et al., 2025). While these approaches are effective in improving task performance and mitigating factual gaps, they typically require carefully curated KGs and additional training overhead, which can limit scalability and adaptability to rapidly evolving knowledge.

### 2.6.2 Retrieval-augmented Generation

RAG enhances the capabilities of LLMs by connecting them to external data sources. In a typical RAG system, a user's query is used to retrieve relevant documents or text chunks from a large corpus. This retrieved information is then combined with the original query into a prompt for the LLM, which generates a response grounded in the provided context (Ram et al., 2023). This approach is particularly effective when the knowledge base is too large to fit within the LLM context window. The most common retrieval

method, vector RAG, involves embedding the text corpus into a vector space and retrieving chunks that are semantically closest to the query vector (Gao et al., 2024). However, this approach struggles with queries that require a holistic understanding of the entire dataset.

**GraphRAG:** To strengthen standard retrieval, many systems incorporate KGs (Guo et al., 2023; Deng et al., 2024; Sun et al., 2023). GraphRAG (Edge et al., 2024) leverages the inherent modularity of KGs to support global sense-making in two stages: indexing and querying. In the indexing stage, an LLM builds an entity-level KG from source documents, then partitions the graph into a hierarchy of nested communities based on the density of connections between entities. The LLM then produces bottom-up summaries for each community, yielding a hierarchical summary tree that aggregates local insights into global ones. In the querying stage, GraphRAG extracts a subgraph based on the pre-generated community summaries and the query, and uses that subgraph as context to generate answers. With its hierarchical design, GraphRAG reports improvements over conventional vector-based RAG (Edge et al., 2024).

**Using GraphRAG for KG evaluation:** The global sense-making capability of GraphRAG also enables a unique method for evaluating the quality of the KG and benchmarking it across various tasks. The accuracy of the answers to user queries is directly dependent on the coverage, validity, and factuality of the KG. A poorly constructed KG (e.g., one with incomplete entities or incorrect relationships) results in fragmented, inaccurate, or nonsensical responses. Therefore, by assessing the quality of GraphRAG output to queries, one can indirectly measure the quality of the KG itself.

### 2.7 Graph Attention Networks and Graph Transformer Architecture

The transformer architecture (Vaswani et al., 2017) has emerged as the dominant paradigm for NLP tasks. Yet, the native transformer self-attention module handles only sequential input. To enable training of a transformer on graphical input, the graph structure must either be encoded into the input or the attention module must be modified. We do both: (1) encode relation embedding into input graph sequences (see Sec. 4.2.1) and (2) modify attention weights to reflect spatial distance in input graphs (see Sec. 4.2). To implement our model, we take inspiration from Graphormer (Ying et al., 2021) but design alternative graph encodings tailored to language tasks.

## 3 Motivational Example

What follows is a motivating example to demonstrate the importance of reliability in our proposed pipeline. To do this, we design a simple "reverse test" using Unified Medical Language System (UMLS; Bodenreider 2004) triples. First, we sample a ground-truth triple from UMLS: ⟨`chronic kidney disease`, `has_finding_site`, `kidney structure`⟩ [SNOMED CT United States Edition vocabulary in UMLS lists kidney structure as the only finding site for chronic kidney disease (CKD)]. Next, we manually create a sequence that implies a weak connection between CKD and cerebellar gray matter abnormalities (so that it is appropriate for the `associated_with` relation, i.e., ⟨`chronic kidney disease`, `associated_with`, `cerebellar gray matter`⟩ would be an ontologically appropriate triple). We craft this sequence so that it reflects recent medical studies on brain imaging in patients with CKD (Xiao et al., 2024), which show indirect abnormalities in the cerebellar gray matter. Even with this indirect correlation, the logical triple should remain ⟨`chronic kidney disease`, `has_finding_site`, `kidney structure`⟩, as found in UMLS.

Next, we prompt strong general-purpose LLMs (Gemini 2.5 Pro, Claude Sonnet 4.5, GPT-5, and Grok 4) to infer the triple from the sentence. Among all these LLMs, only Claude Sonnet 4.5 was able to give the correct tail. We present the Gemini 2.5 Pro response below, and place Claude Sonnet 4.5, GPT-5, and Grok 4 responses in Appendix B. Despite fluent rationales, these models frequently hallucinate relations or return ontologically invalid outputs, yielding triples that are non-factual or misaligned with UMLS constraints. Their outputs seem to show spurious correlations instead of semantic connections. In contrast, our GRAPHMERT model, trained as detailed in the following sections, recovers the correct UMLS triple from the same sentence. This example underscores that adhering to biomedical ontologies matters: Terms like "gray matter" should be used with an `associated_with` relation rather than being miscast with

`finding_site`; structure-aware training is essential for preventing such errors. This is only possible with the proposed GRAPHMERT pipeline.

---

**Prompt.** Please complete the following medical KG triple (head, relation, tail):
(`chronic kidney disease, has_finding_site, ...`)

based on the sequence:

Chronic kidney disease (CKD) is a renal disorder. However, neuroimaging cohorts of people with CKD have reported subtle, group-level abnormalities within the cerebellar gray matter. Studies note modest reductions or regional alterations in cerebellar gray-matter volume, cortical thickness, and diffusion metrics. These observations are heterogeneous and not diagnostic, yet they place observable findings at the cerebellar gray matter in CKD. Thus, CKD may be associated with cerebellar gray-matter changes.

---

**Gemini 2.5 Pro:** (chronic kidney disease, has_finding_site, cerebellar gray matter)

The provided text explicitly states that neuroimaging studies "place observable findings at the cerebellar gray matter in CKD."

---

Next, based on the top 20 predicted tokens from our model with top-5 softmax probabilities of 0.789, 0.173, 0.01, 0.003, 0.003, our pipeline yields "kidneys" (with Gemini 2.5 Pro as a helper LLM), which corresponds to the UMLS gold answer:

---

**GraphMERT:**
structure, kidney, system, function, renal, disease, heart, kidneys, of, physiology, arterial, underlying, pancreas, organ, impaired, health, cerebrovascular, pathophysiology, injury, morphology

**Gemini 2.5 Pro reasons over GraphMERT triple candidates:**
1. Analyze the Input:
Sequence: The text describes Chronic Kidney Disease (CKD) as a "renal disorder," indicating its primary location is the kidneys. It also explicitly mentions that in people with CKD, neuroimaging has found "abnormalities within the cerebellar gray matter."
. . .
2. Generate Potential Tails:
The most obvious and primary finding site for a "renal disorder" is the kidneys. The token list contains kidneys. This is a very strong candidate.

I can also form renal system from the tokens renal and system. This is also correct but is more general than kidneys. Specificity is preferred.

3. Verify and Filter Tails:
Candidate 1: kidneys

- Factual: Yes, CKD is a disease of the kidneys.
- Supported by sequence: Yes, the text calls CKD a "renal disorder."
- Valuable & Specific: Yes, it pinpoints the primary organ affected. It is specific and not vague.
- Uses only candidate tokens: Yes, "kidneys" is in the list.
- Conclusion: This is a high-quality, factual tail.

. . .
The text also mentions findings in the "cerebellar gray matter." I can look for tokens to construct this. Cerebrovascular is available, and structure is available. I could potentially form cerebrovascular structure. However, the cerebellum is not primarily a vascular structure, and this tail is very general and not as precise as the information in the text ("cerebellar gray matter"). More importantly, the key tokens cerebellar and gray matter are missing. Therefore,

---

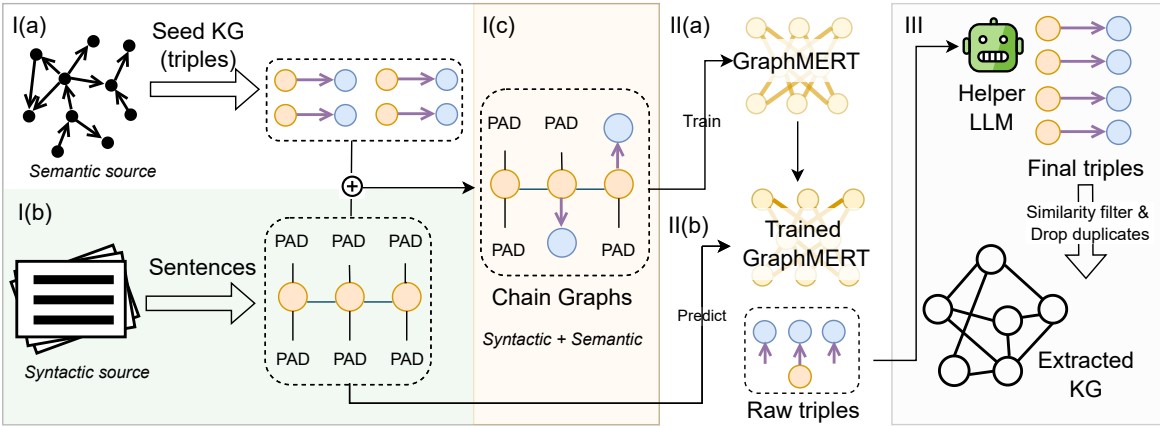

Figure 2: Overview of the GRAPHMERT framework. It is trained on the fusion of syntactic and semantic examples **(II)** and augments syntactic data with semantic tails **(I)**; an LLM helps determine the linguistic structure of tails proposed by GRAPHMERT **(III)**. **(I)**: Chain graph **(Ic)** combines syntactic knowledge from text corpora **(Ib)** with semantic examples and relations from a seed KG **(Ia)**: Roots hold syntactic knowledge (in orange), sparse leaves hold semantic examples (in blue), and edges encode semantic relations (purple arrows). **(II)**: GRAPHMERT is trained on chain graphs to align semantic examples with their syntactic context **(IIa)**. It then predicts novel semantic token completions for chain graphs without injections, using their syntactic information as context **(IIb)**. **(III)**: An LLM combines raw semantic token completions from GRAPHMERT into grammatically well-formed triple tails, producing complete triples. After filtering them by similarity to the source syntactic context and dropping duplicate triples, we obtain the final KG.

> I cannot accurately represent this secondary finding site.
>
> . . .
>
> 4. Final Decision: **The best and most directly supported tail I can construct from the given tokens is kidneys. While the text mentions other sites, the provided tokens do not allow for their accurate creation.**

As we noted above, there is an indirect correlation between CKD and cerebellar gray matter in the source sentence; however, it does not imply the triple suggested by most of the LLMs in their responses. This illustrative case reinforces recent evidence that LLMs regrettably answer based on word correlations in language rather than honing in on the semantic meaning of text and vernacular syntax, making them surprisingly brittle (Li et al., 2020). Recent works also show that, despite the high accuracy on various language tasks, when examined more closely, LLMs are only learning surface-level information, including word overlap, perplexity, sentence lengths, etc., and not the underlying task at hand (Durmus et al., 2022). Taken together, the example above and the additional results we present later make us skeptical about deploying these models in high-stakes use cases for constructing reliable, domain-specific KGs.

## 4 GraphMERT KG-extraction Framework

This section introduces the KG extraction pipeline and the GRAPHMERT architecture. It also introduces an LLM-based KG generation method that is used to obtain the baseline KG.

One of the earliest studies on relational factual knowledge extraction from pre-trained encoder-only models used cloze-style prompts, as discussed by Petroni et al. (2019). E.g., "*John Lennon plays* [MASK]" to complete ⟨John Lennon, PersonInstrument, ?⟩. Later work showed that such a prompt-based retrieval is heavily biased toward prompt syntax rather than factual content (Cao et al., 2021). For instance, García-Silva et al. (2023) report that BERT (Devlin et al., 2019) completes the above example with "*guitar, piano, drums, himself, harmonica,*" where syntactically plausible but non-factual predictions like "*drums*" rank high

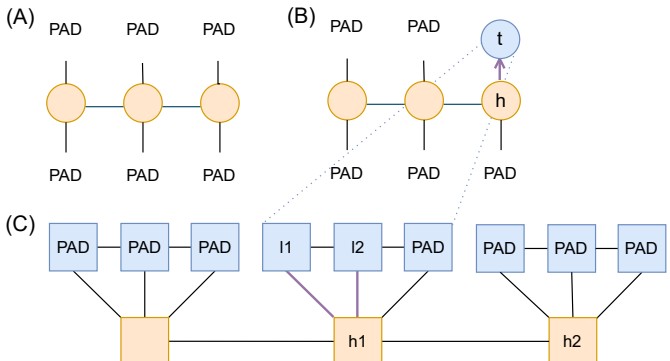

Figure 3: Chain graph. Roots are in orange, leaves are in blue. Conceptual representation (A, B): term level, each circle is a term. Actual representation in training (C): token level, each square is a token. Each term can be multi-token. (A) No injections, all leaves are empty. (B) One root node has a leaf term. (C) Token-level representation for the 3-leaf case. Here, the leaf in (B) is encoded with a maximum of three tokens and padded to the maximum length if needed. Root term comprises two tokens, and tail term also comprises two tokens that are connected to the first root token.

and, surprisingly, more specific prompts, e.g., "*John Lennon plays instrument [MASK],*" produce irrelevant outputs like "*here, there, too, himself, onstage.*"

These shortcomings stem from the syntactic form of the prompt overshadowing the factual knowledge of the model, which is mostly *semantic*. Our key innovation overcomes this obstacle by implanting relations into an encoder via graph attention and training relation embeddings in a dedicated semantic space under the unified MLM + MNM objectives. Our approach directly teaches the model relational knowledge abstracted away from prompt syntax. In parallel, GRAPHMERT learns the syntactic structure too and leverages syntactic information as a context for semantic knowledge.

Fig. 2 shows a high-level overview of the pipeline. GRAPHMERT is a multi-directional encoder-only trans-former. To complete a triple, it predicts a masked tail (a node in the KG, hence MNM). Just like typically text-based encoder-only models, GRAPHMERT also learns syntactic representations from text corpora via the MLM learning objective (Devlin et al., 2019). To enable an encoder-only extraction, we create a new textual data format that encapsulates semantic triples and engineer GRAPHMERT to work in this space.

## 4.1 Syntactic and Semantic Spaces: Merging Semantic Triples and Syntactic Text into a Unified Graphical Format

In essence, *GRAPHMERT performs syntactic-to-semantic knowledge conversion during prediction.* The sentences in the dataset represent the *syntactic space*. The KG triples represent the *semantic space* that includes *semantic relations*. To enable knowledge form conversion, we propose leafy chain graph encoding that unifies the semantic and syntactic representations into a joint representation (see Fig. 3) in which chain graph roots lie in the syntactic space and leaves, along with their relations, lie in the semantic space. The leafy chain graph lets the model process syntactic tokens and semantic leaves within a single attention graph, while still preserving the distinction between syntactic and semantic nodes during training and prediction. As we demonstrate later, leaves play a crucial role in training semantic relation embeddings.

Leafy chain graphs follow a regular structure, which enables sequential encoding of the graphical information. All chain graphs have a fixed number of root nodes; the number of leaves per root node is also fixed. Leaves of the same root are connected, introducing a shortest-path linkage between them. All edges are undirected (the directionality of relations is implied in the architecture).

To create a unified representation for the training data, we parse the dataset into chain graphs with `<pad>` tokens in all leaf positions, keeping only root nodes non-empty. Next, we populate the empty leaves with

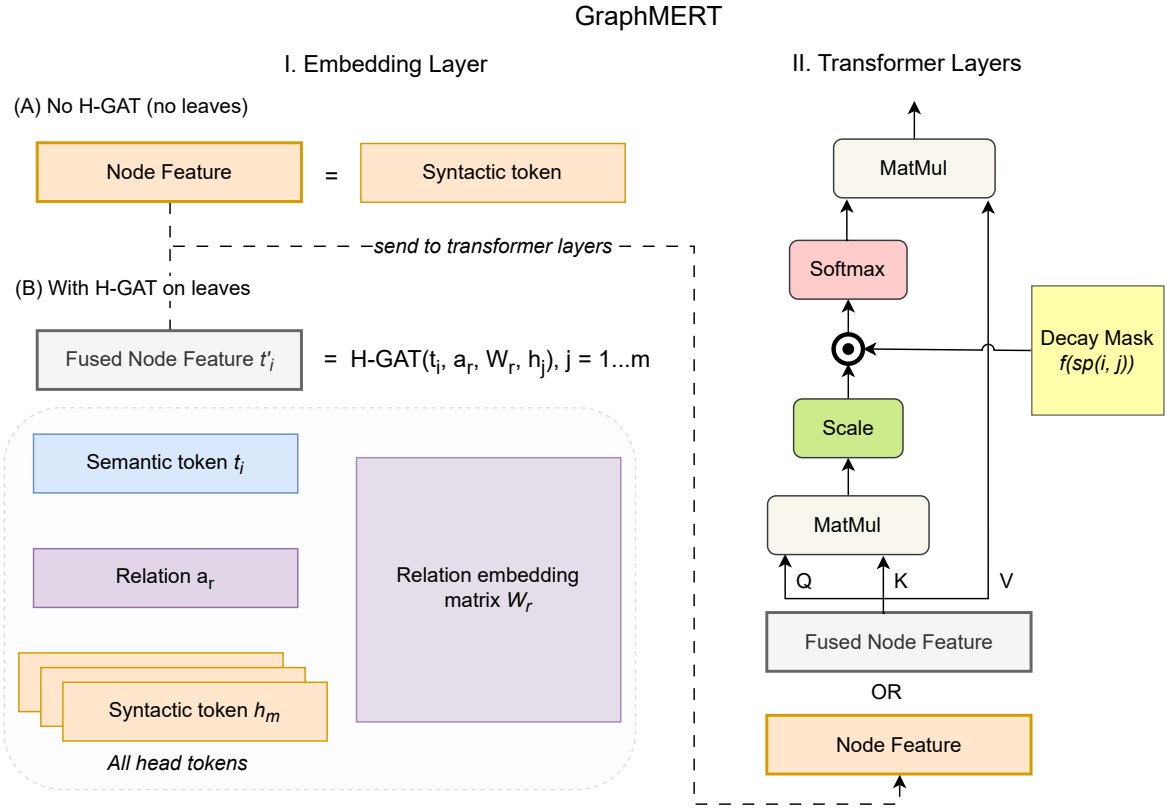

Figure 4: Main GRAPHMERT architectural components. GRAPHMERT is a RoBERTa transformer with two modifications. (I) In the embedding layer, H-GAT encodes semantic triples. (IA) There are no leaves connected to a root node; hence, the node feature is equal to the token embedding. (IB) There are leaves connected to a root node; H-GAT fuses leaves, relations, and head embeddings resulting in fused node feature. (II) In the attention layers, attention weights are multiplied by a function that exponentially decreases with pairwise distance. They encode graph relations and graph distance, respectively. The input is either a node feature or a fused node feature.

semantic nodes and their relations from the seed KG in a separate pipeline step, as detailed in Sec. 4.3. Thereafter, the dataset consists of leafy chain graphs with a *regular structure*. Most leaf nodes are pads, while some contain semantic tail tokens from the seed KG. This regularity in graphical input informs and simplifies the choice of graph encodings in GRAPHMERT, as discussed further in Sec. 4.2.2.

## 4.2 GraphMERT Architecture and Training

The core architectural challenge in graph transformer design lies in encoding graphs into a sequential input for attention-based learning, in order to enable robust graphical representations. We discuss this in the next subsections.

### 4.2.1 Hierarchical Graph Attention Networks

To incorporate semantic relations into GRAPHMERT, we combine a generic transformer architecture with a hierarchical graph attention network (H-GAT; Xu et al. 2021; Nathani et al. 2019). H-GAT fuses relation embeddings into semantic graph nodes before passing the graph sequences to the transformer layers. The original H-GAT architecture hierarchically combines intra-relation and inter-relation attention to derive

node embeddings by aggregating the embeddings of the neighbors of the graph node. To tailor H-GAT to our input (i.e., chain vocabulary graphs that we describe later), we discard the unnecessary inter-relation representations and use the simplified architecture with token embeddings instead of graph node embeddings.

In the GRAPHMERT implementation, given a triple $< h, r, t >$ — head, relation, tail, where head and tail are represented by $\{h_1, .., h_m\}$ and $\{t_1, .., t_n\}$ at the token level, for any given tail token $t_i$ we have:

$$e_{ij}^{(r)} = \text{LeakyReLU}\Big(a_r^\top \big[\, \mathbf{W}_r t_i \,\|\, \mathbf{W}_r h_j \,\big]\Big). \tag{1}$$

where $\mathbf{W}_r$ is a learnable relation embedding matrix, $a_r$ is a learnable relation embedding, and LeakyReLU is an activation function.

$$\alpha_{ij}^{(r)} = \text{softmax}_j\big(e_{ij}^{(r)}\big) = \frac{\exp\big(e_{ij}^{(r)}\big)}{\sum_{k=1}^m \exp\big(e_{ik}^{(r)}\big)}. \tag{2}$$

Then the final node embedding for the tail token is given by:

$$t_i' = \sum_{j=1}^m \alpha_{ij}^{(r)} \mathbf{W}_r h_j. \tag{3}$$

Thus, the new tail token $t_i'$ fuses the relation embedding via $W_r$ and $a_r$ with its initial tail token embedding $t_i$ and all head token embeddings $\{h_1, ..., h_m\}$.

### 4.2.2 GraphMERT Architecture with Graph Encodings

The proposed GRAPHMERT $\mathcal{F}(x, \theta)$, $x$: chain graph input, $\theta$: trainable parameters, is a RoBERTa-style (Liu et al., 2019) encoder-only transformer integrated with H-GAT, trained with the MLM + MNM objective. Fig. 4 illustrates the GRAPHMERT architecture. Our choice of graph encodings is informed by the regularity of the input graph: The input consists of chain graphs with a fixed number of root and leaf nodes. Therefore, to describe the input graph class, node encoding, semantic leaf relation encodings, and spatial distances between node pairs are sufficient. Though common in graph transformers, degree and sparsity encodings offer little value in our setup. The two core components of GRAPHMERT that encapsulate graph encoding are the input embedding layer and the attention decay mask.

The embedding layer processes input nodes, concretely, root nodes along with their leaves, and semantic relations for each injected leaf node. For every injected triple, its head (multi-token) lies in the root space and its tail (also multi-token) lies in the leaf space. The embedding block fuses every leaf token embedding with its relation and all its head tokens via H-GAT (see Sec. 4.2.1):

$$\begin{aligned} t_i' &= t_i + \text{H-GAT}(t_i, r, \{h_1, .., h_m\}), \\ dim(t_i') &= dim(t_i), \end{aligned} \tag{4}$$

where $t_i$ is the tail token embedding, $h_j$ is the head token embedding, and $dim$ is the embedding dimension. The derived embedding replaces the initial leaf embedding, effectively encoding the whole semantic triple into the leaf embedding space, as shown in Fig. 5. During training, masking leaf nodes enables the training of relation embeddings with backpropagation, as shown in Fig. 6.

The attention decay mask encodes the spatial distance between graph nodes. The core idea is that attention between two nodes should decrease with respect to their distance. Following Gradformer (Liu et al., 2024a), we use an exponential function with base $0 < \lambda < 1$ and the shortest path in the exponent. To adjust Gradformer exponential mask for vocabulary sequence graphs that experimentally need a smoother attention decay with respect to the shortest path, we introduce a square root in the exponent.

The shortest path for every node pair is calculated using the Floyd-Warshall algorithm:

$$sp(i, j) \in \mathbb{R}^{N \times N}, \ N = \text{number of input nodes} \tag{5}$$

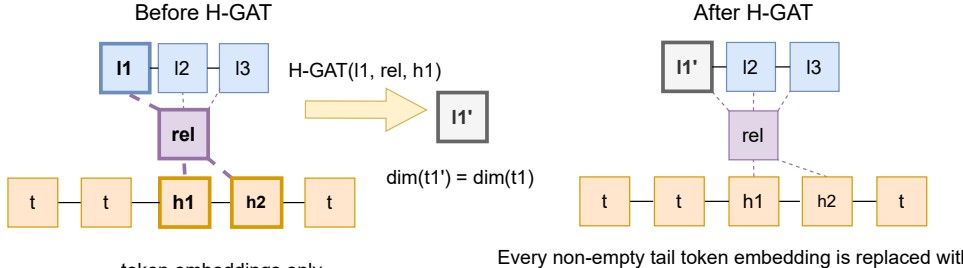

Figure 5: Semantic embedding derivation on leaves (only three leaves are shown). $h_i$: head token, $l_i$: leaf token, $t$: syntactic context token. For every injected triple, H-GAT fuses each leaf token with the relation and all the head tokens, yielding an embedding of the same dimension as the initial leaf token embedding. The derived embedding replaces the initial leaf embedding.

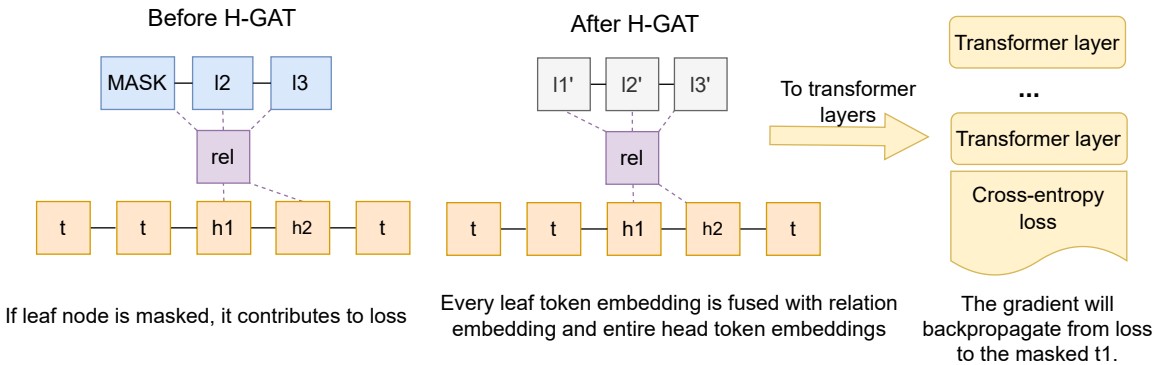

Figure 6: Relation embedding training. The sequence with updated leaf embeddings is passed to the transformer layers. The masked nodes (both roots and leaves) contribute to the loss calculation. For masked leaves, the gradient flows back to them through H-GAT, updating the relation embeddings.

The exponential decay mask is an $N \times N$ matrix defined as:

$$f(sp(i,j)) \triangleq \lambda^{GELU(\sqrt{sp(i,j)}-p)}, \ 0 < \lambda < 1 \tag{6}$$
$$GELU = x\Phi(x),$$
$$\Phi(x) \text{ — Gaussian cumulative distribution function,}$$

where $p$ is a learnable parameter and $\lambda$ is a hyperparameter. For $sp(i,j) \leq p$, the activation function, $GELU$ (Hendrycks & Gimpel, 2023), zeroes the exponent, making the mask close to zero for nodes with the shortest path less than or equal to the learned $p$. Finally, we multiply the attention weights $A \in \mathbb{R}^{N \times N}$ by the mask elementwise, effectively incorporating spatial distance in the attention mechanism.

$$\tilde{A} = A \odot f, \quad \tilde{A} \in \mathbb{R}^{N \times N} \tag{7}$$

The exponential mask is shared across all attention layers.

### 4.2.3 GraphMERT Training

As explained earlier, we represent each sentence as a leafy chain graph $G$ whose root nodes are text tokens, and graft semantic leaf nodes from a seed KG onto entity spans via relation edges. GRAPHMERT jointly

pretrains using MLM over syntactic tokens and MNM over semantic leaves: We randomly select spans in either space and train the model to reconstruct the missing words and/or KG leaves. This setup couples the transformer token encoder with the H-GAT relation encoder so that the surface form and KG semantics align during pretraining.

$$\mathcal{L}_{\mathrm{MLM}}(\theta) = - \sum_{t \in M_x} \Big( \log p_\theta\big(x_t \,\big|\, G_{\setminus M_g \cup M_x}\big) \;+\; \mathcal{L}_{\mathrm{SBO}}\big(x_t \,\big|\, G_{\setminus M_g \cup M_x}\big) \Big), \tag{8}$$

$$\mathcal{L}_{\mathrm{MNM}}(\theta) = - \sum_{\ell \in M_g} \log p_\theta\big(g_\ell \,\big|\, G_{\setminus M_g \cup M_x}\big), \tag{9}$$

$$\mathcal{L}(\theta) = \mathcal{L}_{\mathrm{MLM}}(\theta) \;+\; \mu\,\mathcal{L}_{\mathrm{MNM}}(\theta), \tag{10}$$

where $x$ denotes the input token sequence; $G$ is the text chain graph augmented with KG leaf nodes; $M_x$ and $M_g$ are the masked text and leaf spans, respectively; $x_t$ is a masked token target; $g_\ell$ is a masked semantic leaf target (as a leaf-span explained below); $\mathcal{L}_{\mathrm{SBO}}(x_t)$ is span boundary loss on $M_x$ (explained below); $\theta$ denotes all model parameters (transformer + H-GAT); and $\mu > 0$ balances the two losses (we use $\mu = 1$). Both objectives use span-wise masking.

In GRAPHMERT, H-GAT is responsible for training semantic relation embeddings. Dropout on relation embeddings prevents overfitting on scarce semantic examples. In parallel, transformer attention trains the remaining network parameters, attending jointly to tokens from both the syntactic and semantic spaces. GRAPHMERT is trained with a span-masking schema. Empirically, span masking tightens alignment among the top-$k$ tokens predicted within a single leaf. Later, better-aligned tokens result in more nuanced combinations when constructing complete tails. We discuss the top-$k$ predicted token combination stage in Sec. 4.4.

From the syntactic stream, we exactly follow the span masking implementation, which sums regular masked objective with span boundary loss of SpanBERT (Joshi et al., 2020). In the semantic space, however, we introduce a modification: Whenever a leaf span is selected (with a standard MLM/MNM probability of 0.15), we mask all the leaf tokens rather than sampling span length from the geometric distribution, which could mask a leaf subset for a given root. In other words, $M_g$ and $M_x$ follow different masking schemas. The rationale for this stems from backpropagation. Relation embeddings must receive gradients from the entire tail so that they can capture its full meaning, not just fragments from individual tokens. Since the semantic meaning of a relation manifests only across the complete tail expression, masking the whole leaf ensures that gradients reflect the whole semantic unit.

### 4.3 High-quality Text Sources and Dataset Preprocessing for GraphMERT Training

Our framework focuses on domains with stringent factuality requirements. The compact GRAPHMERT model makes it feasible to train exclusively on limited expert-verified open-source texts. This approach takes an essential step toward reliable KG extraction: Data cleaning reduces hallucinations and training on scientific corpora can substantially limit domain-specific errors (Li et al., 2024). Reliance on expert-verified data prevents importing spurious facts from vast, tainted corpora.

Data quality requirements apply equally to the seed KG. The seed KG is a set of domain-specific triples that serve as initial relation examples for the model. As the full training set in the semantic space, the seed KG defines the relation set for the extracted KG. Its quality, therefore, is of utmost significance in training of robust semantic relation embeddings in GRAPHMERT. A seed KG can be obtained from an external source, provided that it satisfies the following two conditions:

1. It contains clean, domain-specific data.

2. It has a sufficiently diverse vocabulary.

Condition (1) provides the foundation for learning accurate relation embeddings and Condition (2) prevents relation embedding overfitting on a small set of tokens. To obtain a domain-specific seed KG that meets

these requirements, we suggest either selecting a relevant, well-curated external KG or generating one (see Sec. 4.5), followed by thorough cleaning.

After obtaining a KG, we apply a similarity filter to it against the training data (see Sec. 4.3.1), which ensures alignment with the target domain [Condition (1)] and identification of the triples most relevant to the context (see Sec. 4.3.2). The algorithm for selecting the best matches, described in Sec. 4.3.3, addresses vocabulary diversity [Condition (2)]. This algorithm selects triples from the external KG to position them within the semantic space in the chain graphs, given the potential spots for triple heads within the syntactic space. For domain-specific head discovery in the dataset, we use a helper LLM.

### 4.3.1 Entity Linking

Although GRAPHMERT is applicable to any domain-specific text corpora, we leverage biomedical knowledge in the UMLS to develop a multi-stage process to link entities discovered in the text to concepts in UMLS. This process ensures that identified entities are mapped to standardized Concept Unique Identifiers, facilitating the subsequent retrieval of structured information from the UMLS KG. We combine both embedding similarity and string similarity.

**Stage 1: Embedding-based candidate retrieval:** The initial stage aims to rapidly identify a set of potential candidate concepts from the vast UMLS ontology. This is achieved by representing both the discovered source entities (queries) and the target UMLS entities in a shared high-dimensional vector space.

We use SapBERT (Lim & Kim, 2022), an encoder-only language model specifically pre-trained on biomedical knowledge from UMLS, which is able to capture the nuanced difference between biomedical terms. Each discovered head entity and every UMLS entity is processed through SapBERT to produce a unique vector embedding.

To efficiently search through millions of UMLS entity embeddings, we employ an Approximate Nearest Neighbor (ANN) algorithm. ANN constructs a pre-computed index of the UMLS embeddings, enabling highly efficient retrieval of the top-$k$ most similar vectors for a given query vector. For each discovered entity, we use this method to retrieve the top 10 UMLS candidates based on the cosine similarity of their embeddings.

**Stage 2: Fine-grained filtering with string matching:** The top candidates from embedding-based retrieval are then subjected to a more rigorous filtering process based on string similarity. We use character-level 3-grams (char-3grams) as a robust string comparison. We represent each entity name as a set of char-3grams, which can be easily compared with others using standard set-based similarity metrics. We compute Jaccard similarity between the 3-gram sets of the source entity and each of the 10 candidate entities:

$$J(A, B) = \frac{|A \cap B|}{|A| + |B| - |A \cap B|}$$

A candidate entity is confirmed as a valid link only if its Jaccard similarity score is greater than the threshold. The threshold is set to 0.5 based on manual inspection. The entities that successfully pass both stages are considered the final Linked UMLS Entities and are used for the following task.

### 4.3.2 Contextual Triple Selection

Following the entity linking stage, each input sequence is associated with a set of UMLS concepts. While these links grant access to the structured knowledge within UMLS, a single concept can be involved in hundreds of triples, many of which may be irrelevant to the specific context of the source text. Therefore, a crucial subsequent step is to identify and select only the most contextually relevant triples for each sequence.

We perform this selection using an embedding-based relevance-ranking procedure. For each sequence, we begin by retrieving the complete set of triples from the UMLS KG where any of the linked entities from that sequence appear as the head entity. We compute a semantic relevance score for each triple with respect to

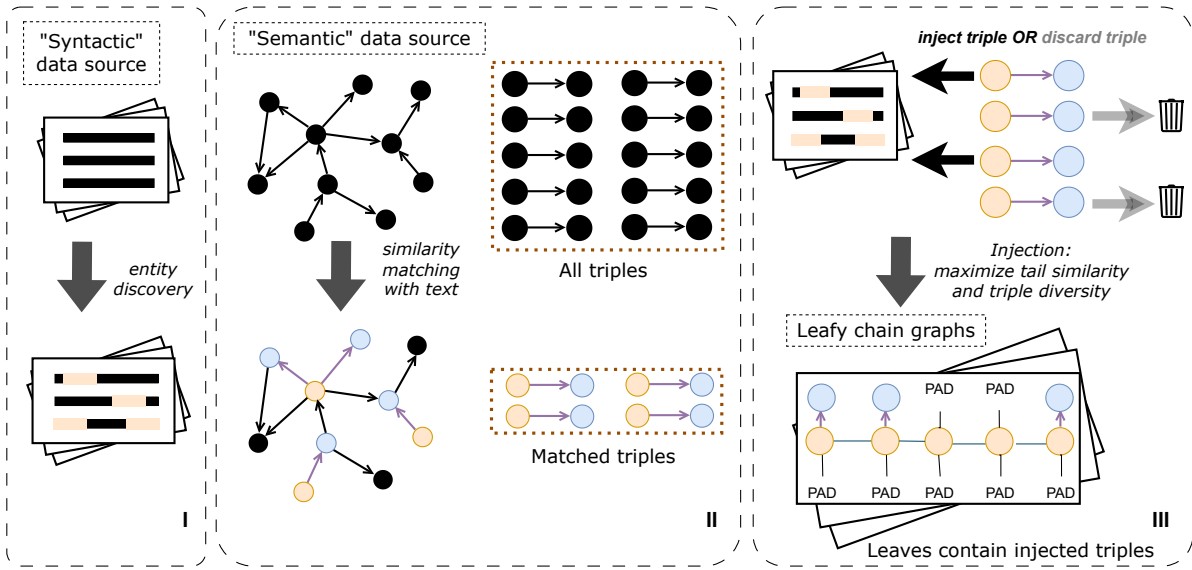

Figure 7: Data preparation for GRAPHMERT. To find the most relevant triples, we perform semantic similarity matching of triples to dataset sequences. The triple head should almost literally match one of the entities discovered in Step (I); from them, we pick the top triples whose tails are semantically close to the sequence. All matched triples are subject to the injection algorithm (III), which selects the top-scoring triples and limits the number of equivalent triples. The injected triples together comprise *a seed KG*.

the original input sequence. We exclude triples with undesired relations from the search: relations that are not useful to have in the KG (see Table C1 in Appendix A).

Specifically, each retrieved triple, consisting of a head, relation, and tail, is transformed into a linearized sentence by concatenating its components with spaces. We encode both the original input sequence and the sentence formed by each triple into high-dimensional vectors using the Gemini embedding model, text-embedding-004, and use cosine similarity as the semantic relevance score. For each linked entity, we rank its associated triples by their semantic relevance scores and retain the top 40 triples. The resulting contextually filtered set of triples is then used in the subsequent injection process.

### 4.3.3   Seed KG Injection

The KG injection algorithm prepares a GRAPHMERT-compatible dataset of leafy chain graphs by selecting relevant triples from an external KG source (potentially, limited in size) based on their similarity score to the input sequence, thresholded with a hyperparameter $\alpha$. At the same time, the algorithm maintains diversity in the injected relations and semantic vocabulary. All triples selected by the algorithm comprise the seed KG. In this process (see Fig. 7), triples are mapped to the chain graph semantic space: The head is placed at a root node and the tail at the root leaf node. Critically, the injected triples must be contextually relevant to the sequence. This aligns transformer attention with H-GAT during training on the chain graphs, as both attend to the semantic and syntactic spaces simultaneously; otherwise, the attention layers would receive a noisy signal from extraneous tokens. Furthermore, because both H-GAT and transformer attention jointly train GRAPHMERT relation embeddings, alignment between leaf and root tokens enables vocabulary transfer from the syntactic root space into the semantic leaf space. This process forms integrated representations that support the retrieval of novel tails from a shared embedding space during prediction.

**Why do we need an injection algorithm?**   A naive strategy would be to inject the top-scoring triple for each head out of all the matched ones. We now demonstrate why a triple with the best similarity to a sequence may be suboptimal for both (a) populating the semantic space and (b) GRAPHMERT training.

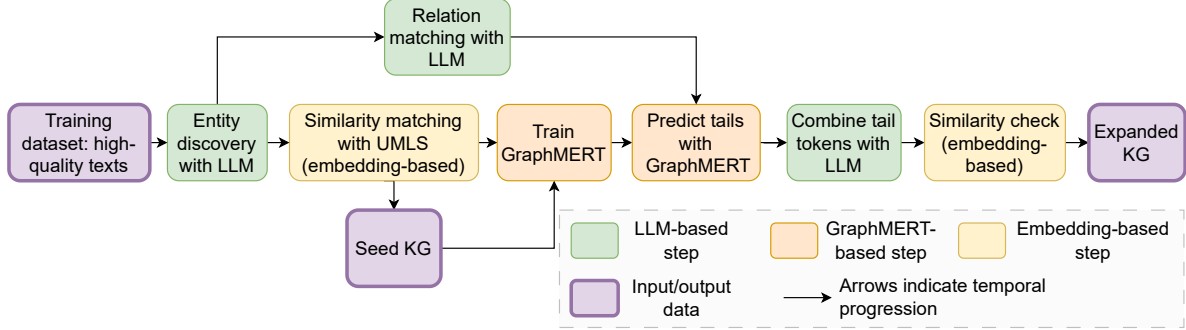

Figure 8: GRAPHMERT Pipeline flowchart with temporal execution ordering of the main components.

**(a) Limitations for the semantic space:** Similarity matching favors frequent terms in the dataset (Zhou et al., 2022). Hence, triples with common domain-specific keywords in tails score highly across many sequences. As a result, a small set of triples achieves high similarity scores across a large number of sequences. If only top matches were to be chosen, these ubiquitous tails would dominate the semantic space, suppressing rarer but semantically valuable tails that introduce novel terms into the semantic space. The scoring is also biased towards classificatory relations like `isa` or `inverse_isa`, since they often restate the head (e.g., ⟨*fibrosing interstitial lung diseases, isa, fibrosis of lung*⟩ in a biomedical KG). Such close textual matches contribute little new information.

**(b) Limitations for GraphMERT training:** Over-injecting frequent tokens leads to a skewed training distribution. If the semantic space vocabulary is dominated by a few tokens, relation embeddings overfit on them, causing GRAPHMERT to predict a narrow token subset. Likewise, certain relations, e.g., "isa" and "inverse_isa," would dominate the limited spots for injected triples, suppressing all other relations. Thus, GRAPHMERT will be undertrained on the other relations.

**Design goals:** To address these limitations, we design the injection algorithm around three goals:

1. Eliminate low-relevance matched triples.

2. Select one triple ("inject") per head out of all matched with the sequence.

3. Diversify injected relations by balancing examples across all relations.

Goal (1) can be satisfied by thresholding similarity scores. However, Goals (2) and (3) cannot be enforced independently: Selecting only the top match achieves (2) but undermines (3), amplifying certain top-scored relations, notably "isa". The algorithm must enforce (2) and (3) jointly, balancing contextual relevance with relation diversity. The proposed KG injection algorithm iteratively drops undesired triples from all matched triples in two interleaving phases: first, by maximizing the score; second, by maximizing the diversity of relations. The surviving triples are "injected" into the semantic space and comprise the seed KG. We defer the description and implementation details to Appendix D.

### 4.4 GraphMERT Pipeline for Knowledge Graph Extraction

We distill internal GRAPHMERT representations from the trained GRAPHMERT into explicit graph triples by adding leaf nodes. Using purely-MNM prediction, we distill semantic knowledge directly from GRAPHMERT weights, conditioned on a sequence from which we want to extract a triple. The role of a sequence in our framework is analogous to the role of a prompt in LLM-based KG generation, but our prediction is unambiguous and deterministic.

Fig. 8 shows the execution order of the framework components. Fig. 9 illustrates the first pipeline step. Starting with the syntactic corpus (no KG injections), we sample head ($h$) spans from the syntactic space

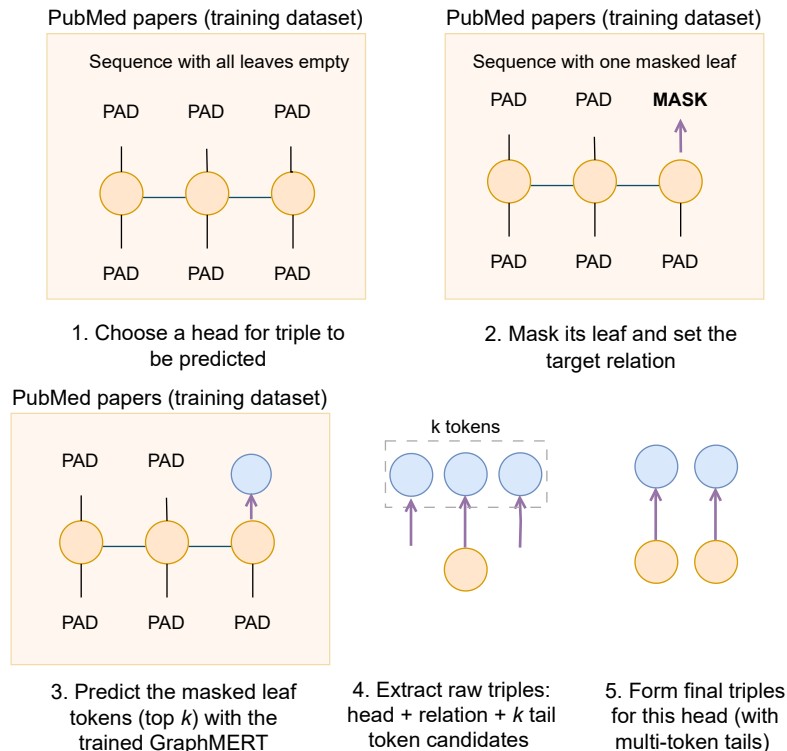

Figure 9: Prediction of triple tails. The trained GRAPHMERT predicts the top $k$ tokens for a masked leaf and the chosen relation, resulting in a set of raw triples with the same head.

(root nodes) and assign an outgoing relation ($r$). We then create the corresponding tail slot ($t$, leaf) for the chosen relation and initialize it with masks. We then ask the model to predict the masked tail (leaf) conditioned on the head, relation, and the rest of the sequence. The top-$k$ predictions for each masked leaf yield $k$ candidate tokens, which serve as building blocks for tails of each head–relation pair $\langle h, r \rangle$. Next, a helper LLM combines predicted tail tokens into coherent phrases, followed by a cleaning step. We further filter the generated triples by computing semantic similarity between each triple and its source sequence, discarding those with a score below the user-defined *similarity check threshold $\beta$*. GRAPHMERT may predict a tail that connects a head with a semantically related concept across the training corpus, and hyperparameter $\beta$ regulates the fraction of such triples in the output. A higher $\beta$ yields fewer but more sequence-specific triples, often explicitly included in the text. A lower $\beta$ enables broader, more general (yet semantically related) triples that may not be explicitly mentioned in the sequence. However, if $\beta$ is set too low, the output becomes flooded with triples that merely restate general truths, reflecting statistically dominant statements in the training dataset.

The surviving $\langle h, r, t \rangle$ triples expand the KG with novel facts. Importantly, each prediction is traceable to its source sequence, while at the same time accumulating knowledge distilled from the entire training corpus. This global knowledge accumulation contrasts with RAG methods, which remain local to retrieved documents. Further, RAG is a post-hoc data attribution mechanism, unlike ours.

**Role of the helper LLM:** The laborious (grammatical) part of the work is carried out with help from an LLM and the essential (triple extraction) part done with GRAPHMERT. In this pipeline, the helper LLM performs three auxiliary tasks: discovering head entities, selecting relations for subsequent GRAPHMERT prediction, and combining single-token predictions into meaningful, relation-aware tail phrases. Critically,

the LLM is constrained: It cannot invent new entities or relations, as heads must be present in the dataset, relations are restricted to the seed KG, and only GRAPHMERT-predicted tokens are allowed to be in tails.

**Why do we need a helper LLM for combining tokens?** An encoder-only model does not address span decoding. Masked span prediction would still be challenging for a small model, given the very limited number of semantic examples. Here, two factors come into play:

1. For a coherent span prediction, each token should be conditioned on other tokens. However, in an encoder-only model prediction, each masked token in the span is conditioned on the sequence independently of other span tokens.

2. In our experiments, training with span masking results in a prediction where tokens in the syntactic space are better aligned with each other within a span; however, the semantic space has orders of magnitude fewer examples ($10^4$ in our experiments). Given the limited scale of the semantic space, how to achieve the same effect on leaves remains an open question.

Thus, training on small corpora to some extent trades English proficiency for data quality. We further study the effect of the helper LLM size in an ablation in Appendix I. Exploring methods to augment GRAPHMERT and thereby remove the LLM-based combining-tokens step is an avenue for future work.

### 4.5 LLM-generated KG as a Baseline

We compare GraphMERT to LLMs for KG construction rather than fine-tuned BERT or T5 models because we focus on low-label domain regimes, where no large sentence–triple annotated corpus exists, and only a small seed KG is available.

To provide a fair and robust comparison for our framework, we construct a baseline LLM KG using a standard LLM-based pipeline that follows the GraphRAG indexing methodology, with a filtering step to align its schema with the GRAPHMERT KG.

The process begins by segmenting the source documents into smaller, manageable text chunks. An LLM is then prompted to perform open information extraction on each chunk, which involves identifying entities, extracting the relationships between them, and generating short, descriptive summaries for each entity. We retain only the relationships that are part of our pre-defined relation set, discarding all others. This ensures that the resulting LLM KG shares exactly the same relational schema as our framework, enabling a direct and equitable comparison of performance. After filtering, the extracted elements are subsequently aggregated and consolidated into a final graph structure. To handle multiple mentions of the same concept, exact string matching is used to resolve entities into unique nodes. Relationships that are generated multiple times between the same two entities are aggregated into a single edge.

### 4.6 KG Verification

We can subdivide KG verification methods into two categories: *graph-level* and *triple-level*.

#### 4.6.1 Graph-level Verification

This method evaluates the KG as a whole. It focuses on its logical coherence, internal consistency, comprehensiveness, coverage, important aspects of domain-relevant knowledge, and depth (considering whether it contains rich, insightful connections beyond surface-level facts). Graph-level approaches typically operate by retrieving relevant subgraphs and evaluating their quality with respect to these criteria.

**GraphRAG:** We employ GraphRAG to evaluate and benchmark KGs across various tasks. The KGs serve as the primary source of information in GraphRAG to answer medical questions, which enables us to compare their effectiveness directly. In our implementation, we use the Local Search method from GraphRAG and modify it to rely exclusively on the entities and relations in the querying stage. This process begins by identifying a set of entities within the KG that are semantically related to the user query. These entities

serve as entry points for retrieving connected entities and relationships. The retrieved data sources are then ranked and filtered to fit within a single predefined context window, which is used to generate a response to the user query.

### 4.6.2 Triple-level Verification

At the triple level, verification can enhance *factuality* and *validity*, as we describe next.

**FActScore:** The FActScore framework (Min et al., 2023) provides a fine-grained method for evaluating factual precision in long-form LLM outputs. Its principles transfer naturally to KG verification. FActScore evaluates atomic facts, i.e., short, self-contained statements, against a trusted text source that does not have any knowledge conflicts or overlaps. KG triples can be treated as atomic facts of equal importance. In our setting, each triple can be paired with a reliable text source: GRAPHMERT triples with sequences and LLM triples with short chunks, both drawn from the same trustworthy source. The short context length minimizes conflicts and overlaps.

**FActScore\*:** We follow the *Retrieve → LM* variant of automatic evaluation, in which an atomic fact is concatenated with the knowledge source and provided to the model. However, we strengthen triple evaluation with validity: In the prompt, we require verification of triple logical alignment in addition to context support, since the fact may appear in the text, yet the triple may still be malformed. Malformed triples should not be deemed reliable facts and would inflate the score. Because our prompt departs from the original FActScore prompt, we denote the modified version as FActScore\*. Formally, let $\mathcal{G}$ be a set of triples $\tau$, and $\mathcal{C}(\tau)$ the text source of $\tau$. Then FActScore\* for $\mathcal{G}$ is:

$$
\begin{aligned}
f(\tau) &= \mathbb{I}[\tau \text{ is supported by } \mathcal{C}(\tau)], \\
\text{FActScore*}(\mathcal{G}) &= \mathbb{E}_{\tau \in \mathcal{G}}[f(\tau)].
\end{aligned}
\tag{11}
$$

**ValidityScore:** Triples should follow semantic rules of a KG. For example, in UMLS (Bodenreider, 2004), the triple ⟨*beta-receptor, part_of, plasma membrane*⟩ is valid, since `part_of` denotes a meronymic (structural/spatial) relation: Every instance of the part must be a constituent of some instance of the whole. It follows, then, that ⟨*beta-receptor, part_of, adrenergic signaling*⟩ is invalid, because it links a physical structure to a biological process. Similarly, ⟨*beta-receptor, part_of, human*⟩ is factually plausible but still an illegitimate usage, since it violates granularity: The correct wholes are specific structures (e.g., plasma membrane), not entire organisms.

To quantitatively measure the validity of triples, we propose ValidityScore. It isolates ontological alignment of triples as an independent mode of evaluation. Concretely, we use a strong LLM judge to semantically validate a triple using the following prompt.

> **Prompt.** Evaluate if these medical KG triples are valid (yes/no/maybe) and give a very short reason why: ⟨list of triples⟩.

Then ValidityScore counts the number of "yes" responses.

## 5 Experimental Setup

To demonstrate the effectiveness of our framework, we extract a high-quality diabetes KG from a GRAPH-MERT-compatible diabetes training dataset obtained from expert-verified sources. This section provides an in-depth explanation of the proposed training, extraction, and evaluation pipeline.

### 5.1 GraphMERT Training and Extraction

Next, we discuss training data preparation and triple extraction.

Table 1: Dataset size

|            | Abstracts | Tokens | Sequences |
|------------|-----------|--------|-----------|
| **Training** | 350k | 124.7M | 989,666 |
| **Evaluation** | 39k | 13.9M | 110,297 |

Table 2: Seed KG statistics for $\alpha = 0.55$ (after Qwen3-32B). We obtain this seed KG based on the 1.1258E+06 matched UMLS triples with a similarity score greater than or equal to $\alpha$.

| Triples count | Similarity score mean | Similarity score median | Similarity score max | Number of relations |
|---------------|----------------------|-------------------------|----------------------|---------------------|
| 28533 | 0.613 | 0.605 | 0.848 | 28 |

### 5.1.1 Training Data Preparation

We showcase our framework in a sensitive medical domain, where concerns over trustworthiness of AI output remain the main barrier to wider AI adoption (Wang et al., 2025b; Mishra et al., 2024), despite the undeniable potential of AI in medical practice. As a case study, we set the goal of extracting a high-quality diabetes KG. It all starts with high-quality textual data. We build a GRAPHMERT-compatible diabetes training dataset (Table 1) from two main sources: (1) peer-reviewed medical abstracts from MEDLINE journals accessed via PubMed Central, and (2) a seed KG derived from the UMLS Metathesaurus (Bodenreider, 2004).

**Text corpus:** MEDLINE is the the National Library of Medicine bibliographic database and is accessible via PubMed. MEDLINE selects journals based on rigorous criteria and indexes them using MeSH terms. We retrieved diabetes-related papers from PubMed Central (see the query in Listing 1 in Appendix C), removed non-English records, parsed abstracts using the PubMed parser (Achakulvisut et al., 2020), lowercased all text, and filtered out opening boilerplate words including "Abstract," "Background," "Introduction," etc., with a regular expression. The resulting dataset contains 350k abstracts for training and 39k for evaluation, totaling 124.7M and 13.9M tokens, respectively.

**Seed KG:** From UMLS, we select SNOMED CT, US and GO (Gene Ontology) vocabularies because together they cover a broad range of biomedical concepts across clinical documentation, molecular biology, and data exchange. We exclude low-value relations (more details in Appendix C) and retrieve triples relevant to our dataset sequences based on semantic similarity matching (Sec. 4.3.1) with Gemini text-embedding-004. For matched triples, we use an injection algorithm (Sec. 4.3.3) with a similarity threshold $\alpha$ of 0.55, validated experimentally via grid search using GraphRAG evaluation (see Sec. 6.4.1). The injection algorithm uses 51 GB of peak RAM during the pre-processing stage and runs for approximately 0.5 hours on a single CPU. The resulting triples that are injected comprise the seed KG (Table 2). Statistics per relation are presented in Table D1 (Appendix D)

As a helper LLM, we employ Qwen3-32B-FP8 (an 8-bit quantized version of Qwen3-32B, further referred to as Qwen3-32B), an open-source, advanced, and lightweight LLM with "thinking mode" turned on. We always use "thinking mode," unless otherwise specified.

**Entity discovery and relation matching:** For head discovery, we prompt Qwen3-32B with each abstract sequence in the dataset with few-shot examples, asking it to search for medical entities that are relevant to diabetes and its comorbidities. From our observations, changing an example in the prompt significantly changes the number of discovered entities. The outputs are validated against sequences of origin to eliminate hallucinated or misspelled entities. For relation discovery, Qwen3-32B is few-shot prompted with a relation list from the corresponding training seed KG to match entities with all relations that make sense for a given entity in the context of the current sequence. Prompts and examples are presented in Appendix G.

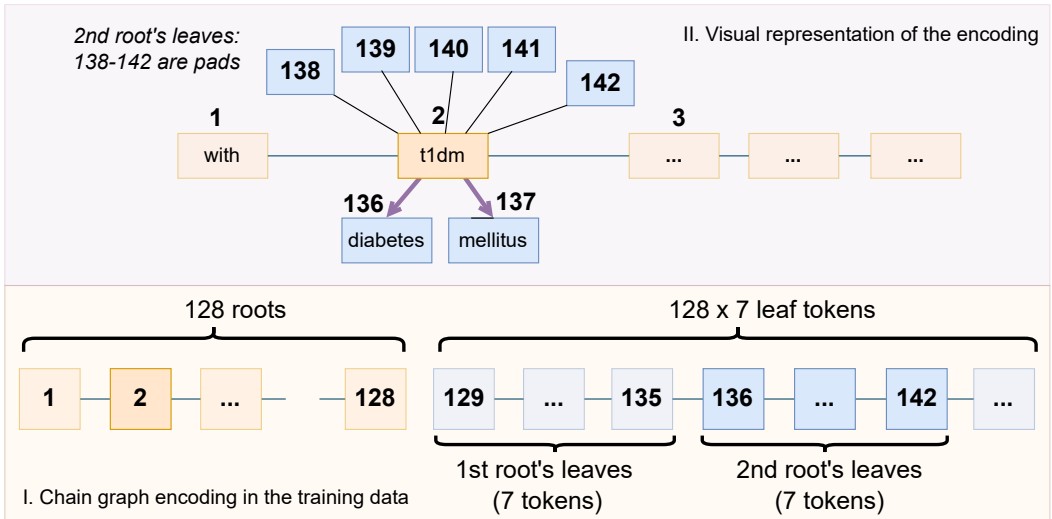

Figure 10: Leafy chain graph encoded sequentially: 7-leaf case. The sequence has a fixed length of 1024. The first 128 tokens are reserved for roots and leaves reside in the remaining tokens. The first group of seven leaves belongs to the first root, the second group of seven leaves belongs to the second root, and so on. Each leaf token group is padded to the maximum length of seven.

All Qwen3-32B runs are performed on one H100 GPU with 80 GB of memory using vLLM with the vendor-recommended sampling parameters: $temperature = 0.6$, $top\_p = 0.95$, $top\_k = 20$, and $min\_p = 0$. In addition, $max\_tokens$ and $max\_model\_len$ are set to 8192, which is sufficient for our generation length.

**Entity linking:** We implement approximate ANN search for entity linking using Faiss, running on CPU over a filtered subset of 3,647,914 UMLS entities. The search over 117,317 discovered head entities completes in 133 minutes, with approximately 48% of them successfully linked to a UMLS entity.

**Chain graphs for training:** We initialize chain graphs with 128 root nodes, each connected to seven leaves, leading to a 1024-token sequence (see Fig. 10). These numbers are chosen to fit GPU memory constraints while providing sufficient room for semantic tokens dedicated to leaves.

### 5.1.2 GraphMERT Training

We train GRAPHMERT with 12 hidden layers, eight attention heads, a hidden size of 512, and an intermediate size of a fully-connected layer of 2048, totaling 79.7M trainable parameters. We use the BioMedBERT tokenizer (Chakraborty et al., 2020), trained on a vast amount of medical vocabulary, to prevent frequent subword tokenization of common medical terms, which is particularly beneficial for the extraction stage. The tokenizer determines the vocabulary size, which is 30,522.

Training runs for 25 epochs on four H100 GPUs with 80 GB of memory each, with BF16 precision, totaling 90 GPU hours. We use an instantaneous batch size of 32 per GPU, achieving an effective batch size of 128 through gradient accumulation (steps = 2). We set dropout rates at 0.1 for regular, attention, and activation dropouts. In addition, we set an exponential mask with base $\lambda = 0.6$ and relation embedding dropout of 0.3. We train the model using the cosine learning rate scheduler with the maximum learning rate set to $4 \times 10^{-4}$. We use 500 steps for warm-up. The weight decay at each step is 0.01 times the learning rate at that step. We stop training when the learning rate reaches $1 \times 10^{-5}$. In the span masking training schema, we limit masked spans to a maximum length of seven, matching the number of leaf nodes connected to root nodes.

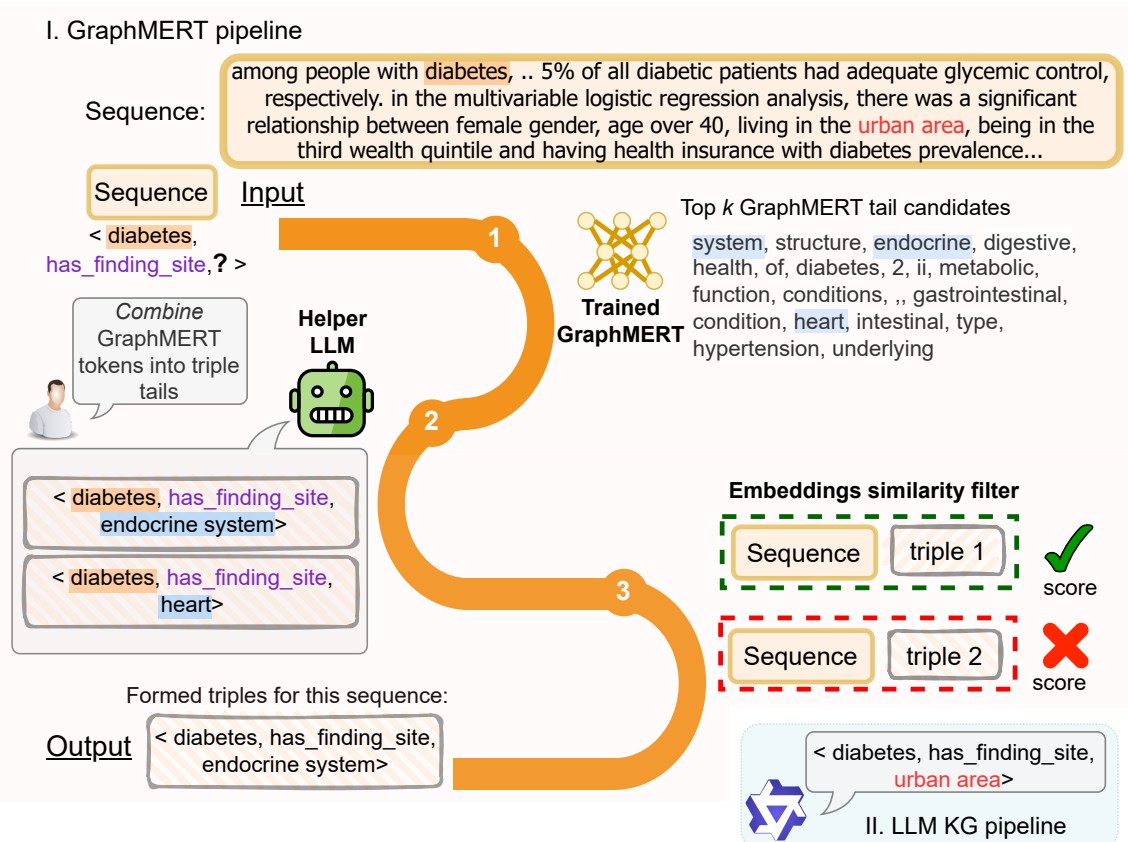

Figure 11: I. Forming triple tails for a given sequence with GRAPHMERT. (1) Given a sequence as a context, a triple head in the sequence, and a relation, GRAPHMERT predicts the tail token (we obtain the top 20). (2) The helper LLM attempts to combine the tokens into a complete, coherent medical term. It may output several or no tail candidates to complete the triple. (3) We evaluate the similarity score between triples from the previous step and the sequence of origin. Only triples with a score higher than a preset threshold pass. II. An example of a triple extracted with the LLM pipeline (Qwen3-32B) from the same context. Here, LLM misinterprets the "has_finding_site" relation, treating "site" as a location instead of an anatomical structure, which results in an invalid triple.

### 5.1.3 Triple Extraction

The triple extraction pipeline runs the following steps, as shown in Fig. 11. We begin with a leaf-masked prediction over the training dataset, given head entities and their relations. Running MNM prediction over the whole dataset with a batch size of 128 takes around 2.5h on 1 H100 GPU (80 GB RAM) with GPU utilization of 90-100%.

This produces a vocabulary distribution for each masked leaf. From this distribution, we select the top 20 tokens per leaf and use them to prompt the helper LLM, Qwen3-32B. Conditioned on the head, relation, and the originating sequence, the LLM combines these tokens into coherent, relevant, and medically meaningful multi-token tails (see Appendix G for the prompt specification). When no valid tail can be formed, the corresponding ⟨sequence, head, relation⟩ is skipped. Next, since the LLM may hallucinate tails outside the GRAPHMERT-predicted token space, we discard any output tails that contain out-of-scope tokens. The output of this stage is a set of completed candidate tails.

Table 3: Examples of GRAPHMERT-extracted triples (UMLS-style) from a single sequence. We pass the sequence with two marked heads together with relations for these heads to the trained GRAPHMERT. After GRAPHMERT makes a prediction in the semantic space, we use the top 20 predicted tokens to form complete triples with a helper LLM. "##" is a separator for subword tokens.

**Sequence 1:** ...##2 The Authors. The Journal of Pathology published by John Wiley & Sons Ltd on behalf of The Pathological Society of Great Britain and Ireland. **Non-alcoholic fatty liver disease** (NAFLD) is one of the main causes of chronic liver disease worldwide. Flavonoids, a group of natural compounds, have garnered a great deal of attention in the management of NAFLD because of their profitable effects on glucose and lipid metabolism, inflammation, and oxidative stress which are the pivotal pathophysiological pathways in NAFLD. **Naringenin** is a citrus-derived flavonoid with a broad spectrum of potential biological effects including anti-inflammatory and antioxidant properties, which may...

| Head, Relation | Top-20 GraphMERT-predicted tokens |
|---|---|
| naringenin, isa | flavonoid ##in flav nar ##idin ##arin ##flav compound ##ce hydrolase plant ##ingen ##ono protein polysaccharide polyphenol family ##anol - ##onin |
| naringenin, plays_role | therapeutic neuroprotective ##ingen antidepressant role medicinal ##arin flavonoid action baical inhibitory antibacterial bioactive medicine antimicrobial ##idin quercetin ##flav potential flavon |
| naringenin, has_disposition | nar flavonoid ##in hydrolase ##arin inhibitor - ##anol compound ##idin amy ##flav ##ingen derivative amide receptor acid family alkaloid product |
| non-alcoholic fatty liver disease, cause_of | liver disease alcoholic fatty ##osclerosis nafld steatosis disorder ##atitis ##ohep hypercholesterolemia hepatic - with syndrome mellitus myopathy fibrosis diseases hyperlipidemia |
| non-alcoholic fatty liver disease, associated_with | hyperlipidemia disorder alcoholic obesity - dyslipidemia myopathy liver hereditary hypercholesterolemia syndrome ##ament ##tr associated ##lip fatty disease ##oid mellitus related |

| Head | Relation | Tail (formed from the GraphMERT-predicted tokens) |
|---|---|---|
| naringenin | isa | flavonoid |
| naringenin | plays_role | therapeutic role |
| naringenin | has_disposition | flavonoid |
| non-alcoholic fatty liver disease | cause_of | fibrosis |
| non-alcoholic fatty liver disease | associated_with | obesity |

Next, each candidate triple is evaluated with a similarity matching algorithm (Sec. 4.3.1). Specifically, we compute the cosine similarity between the triple and its originating sequence using Gemini embeddings. All triples with a score below the similarity check threshold $\beta = 0.67$ (obtained through grid search, see Sec. 6.4.1) are discarded. The remaining set forms the final collection of extracted triples.

Table 3 illustrates the step-by-step triple extraction process for a representative sequence. Given a sequence with its head and relation, GRAPHMERT predicts a token in a masked tail. From the output distribution, we select the top 20 tokens as candidate tails, striking a balance between prediction quality and diversity, and providing a sufficient pool for subsequent token combination. We may construct zero, one, or several novel triples out of these. For more raw GRAPHMERT output examples, see Appendix I.

## 5.2 LLM-generated KG

Following the default GraphRAG indexing parameters, our diabetes corpus is split into 2,000-token chunks and processed by Qwen3-32B to extract entities and relationships. The thinking mode is enabled, and the parameters are set as $temperature = 0.6$, $top\_p = 0.95$, $top\_k = 20$, $max\_tokens = 8192$. The detailed prompt for extraction is shown in Appendix H1 and an example is shown in Appendix H2. After parsing

and cleaning, the final LLM-generated KG contains 272,346 triples. It serves as the baseline in all our experiments.

In addition, we extract a fixed set of 100,000 triples with Grok 4 Fast (grok-4-fast-reasoning), which is approximately a 1.7T-parameter model, and Qwen3-14B (thinking mode) to evaluate how model scale affects the results.

### 5.3 Evaluation with GraphRAG

Next, we provide details of GraphRAG evaluations.

#### 5.3.1 GraphRAG Settings

Our experimental setup uses Qwen3-14B as the backbone LLM in GraphRAG, with inference accelerated using the vLLM library. For all evaluations, we enable the thinking mode of the model and set $temperature = 0.6$, $top\_p = 0.95$, $top\_k = 20$, and $max\_tokens = 8192$. To ensure the reliability of our findings, each experiment is conducted three times with different random seeds (1, 2, and 3). We report the average accuracy across these runs in our final results.

The GraphRAG query process is configured with nomic-embed-text-v1 (Nussbaum et al., 2025) as the embedding model. To construct the context for each query, the system retrieves the top 30 entities and the top 10 relationships per entity, with the maximum context length capped at 12,000 tokens. Furthermore, we tailor the system prompts and simplify the output table structure to better suit our tasks. The complete prompt is given in Appendix H3. The modified table schema is also detailed in Appendix H4.

#### 5.3.2 Benchmark Evaluation

We further verify the quality of our extracted KGs on the diabetes subsets of medical benchmarks: ICD-Bench (Dedhia et al., 2025), MedMCQA, MedQA, and MMLU (medical). ICD-Bench is a targeted question-answering benchmark aligned with the International Classification of Diseases (ICD) taxonomy (World Health Organization, 1992), designed to evaluate domain-specific medical reasoning in language models across 15 medical sub-specialties. We primarily evaluate the extracted KGs on ICD-Bench, as it is the only benchmark that has a dedicated endocrinology subset; for this reason, we defer the results on other (public) benchmarks to Appendix F. For evaluation, we employ GraphRAG to answer benchmark questions.

To address the domain mismatch between our diabetes-specialized model, GraphMERT, and general medical benchmarks, we create domain-specific evaluation subsets. First, we synthesize our training corpus into an approximately 2000-word summary using Gemini 2.5 Pro. Subsequently, we use this summary to prompt a Qwen3-32B model to filter the question-answer pairs in each benchmark, retaining only questions relevant to the summary. All GraphRAG evaluations are then conducted on these filtered subsets to ensure a fair assessment.

#### 5.3.3 Traceability and User Verification

Allowing users to judge correctness is a simple but powerful principle. OpenAI WebGPT (Nakano et al., 2022) addresses this by browsing the web and citing sources, a strategy also adopted by Perplexity AI. In these systems, the user decides (1) whether the source is credible and (2) whether the output is factual. Our framework builds on this idea: Each triple is directly traceable to its originating sequence. This enables automatic cross-checking, rather than searching external sources. The system retrieves the sequence from which the triple is derived. Since this sequence originates from a verified paper abstract, users can further validate the fact by consulting the source publication if needed.

## 6 Experimental Results

Next, we provide evaluation results for GraphMERT and LLM KGs on the benchmarks.

Table 4: GRAPHMERT-extracted KG. Number of triples over extraction stages; similarity threshold $\beta = 0.67$. We show the number of triples after the helper LLM combines the tail tokens, next after discarding tails hallucinated by the LLM, then after similarity filtering, and finally after dropping triple repetitions.

| $\alpha$ value | Formed tails (non-unique) | Formed tails excluding LLM hallucinated (non-unique) | After $\beta$-filtering (non-unique) | Final: repetitions dropped (unique) |
|---|---|---|---|---|
| $\alpha = 0.55$ | 1,760,088 | 1,536,581 | 138,201 | **109,293** |

## 6.1 Description of the Extracted KG

Table 4 summarizes the statistics of the extracted KG. It inherits the 28 relations from the seed KG (mentioned in Table 2) but contains approximately four times as many triples; the comparison chart (Fig. E1) is presented in Appendix E. The extracted triples include vocabulary that extends beyond the seed KG. As expected, novel *heads* originate from the training dataset, which has a richer vocabulary than the seed KG. GRAPHMERT also generates novel *tails*, predicting tokens from the dataset vocabulary that are absent in the seed KG (see an example given in Table E1 in Appendix C).

## 6.2 Triple-level Evaluation

Next, we present results for triple-level evaluations.

### 6.2.1 FActScore* Results

We use two prompts to evaluate a triple based on:

1. context only,

2. context and model's internal knowledge of general truth.

The first prompt includes the text in black only, the second prompt includes the text in black and teal:

---

**Prompt.** You will evaluate the quality of triples for a medical knowledge graph on diabetes and its comorbidities. For each triple, you are given:

- A sequence providing context
- A head entity, a relation, and a tail entity

Your task: Accept the triple ("[yes]") or reject it ("[no]") based on:

- **Logical alignment**: the tail must logically align with the head and relation; relation must match entity types.
- **Context support**: the sequence should support the triple. Allow statements that are factual and general truth, even if not perfectly aligned with context, but still avoid contradictions. If the triple has no reliable support, reject the triple.
- **Knowledge value**: the triple must add new, medically meaningful information to the graph.

Output **only** "[yes]" or "[no]" as your final judgment. Wrap your reasoning in `<think>...</think>`.

---

Table 5 reports the FActScore* of GRAPHMERT KG versus the Qwen3-32B LLM KG (baseline) using Qwen3-32B and Gemini 2.0 Flash as the validators. In addition, we compare our KG with Grok 4 LLM KG and Qwen3-14B KG samples. Since Gemini 2.0 Flash does not respond with a definitive yes/no judgment for a significant number of queries, we include Table 6 with its FActScore* results breakdown.

We conduct two variants of the evaluation, differing only in an extra accept condition in the prompt: 1) based on *context only*, closely following the original FActScore, and 2) based on *context + model's internal*

Table 5: FActScore* KG evaluation (in percentage points).
**Bold**: better than Qwen3-32B baseline; **underlined bold**: best in column. Percentages are computed per KG type with respect to its own #triples.

| KG type | Model size | #triples | Qwen3-32B as a judge | | Gemini 2.0 Flash as a judge | |
|---|---|---|---|---|---|---|
| | | | Context only | Context + General truth | Context only | Context + General truth |
| Qwen3-32B (baseline) | 32B | 515,460 | 40.2 | 48.1 | 49.1 | 44.3 |
| GRAPHMERT | 80M | 138,201 | **69.8** | **72.2** | **79.7** | **84.0** |
| Reference LLM KGs | | | | | | |
| Grok 4 Fast | 1.7T | 100,000 | **79.4** | **79.7** | **73.3** | **67.9** |
| Qwen3-14B | 14B | 100,000 | **46.4** | 45.7 | **62.4** | **57.6** |

Table 6: FActScore* KG evaluation with Gemini 2.0 Flash: breakdown.
**Bold**: better than Qwen3-32B baseline; **underlined bold**: best in column. Percentages are computed per KG type with respect to its own #triples.

| KG type | Model size | #triples | Context only | | | Context + General truth | | |
|---|---|---|---|---|---|---|---|---|
| | | | yes | no | no verdict | yes | no | no verdict |
| Qwen3-32B (baseline) | 32B | 515,460 | 49.1 | 41.7 | 9.2 | 44.3 | 30.2 | 25.4 |
| GRAPHMERT | 80M | 138,201 | **79.7** | 18.3 | 2.0 | **84.0** | 12.8 | 3.2 |
| Reference LLM KGs | | | | | | | | |
| Grok 4 Fast | 1.7T | 100,000 | **73.3** | 8.0 | 18.7 | **67.9** | 6.6 | 25.5 |
| Qwen3-14B | 14B | 100,000 | **62.4** | 20.4 | 17.2 | **57.6** | 18.2 | 24.2 |

*knowledge*, which additionally accepts triples that express general truths even if not explicitly stated in the context. Variant (2) acknowledges cross-dataset concept linking, whereas (1) restricts evidence to the local context. Because the same triple can originate from multiple sequences, some triples are checked more than once against different texts. As a result, the reported number of triples exceeds the count in the final deduplicated KGs. We also score the seed KG with respect to the sequences into which its triples are injected. In this setting, FActScore primarily reflects the injection relevance.

**Why are Qwen3-32B LLM KG FActScores so low?**   Surprisingly, the Qwen3-32B LLM KG scores poorly even when validated by the same model, which flags many of its own errors. This underscores weak prompt steerability: Knowledge may exist in parameters, yet prompt-based generation fails to elicit correct, ontology-respecting triples. Our analysis of Qwen3-32B responses points to three recurring failure modes:

1. Relation misinterpretation: The model maps relations by lexical similarity rather than ontological meaning, drawing on its internal knowledge.

2. Systematic malformed repetition: The same ill-formed triples reappear across different text chunks.

3. Overlinking a head–tail pair: Multiple, largely invalid relations are assigned to the same entity pair.

**Illustrative errors**

- *Misinterpreted relation:* ⟨diabetes, `has_finding_site`, urban area⟩. Here, "site" is treated as a location (anatomical structure is expected). Likewise, ⟨telehealth intervention, `has_part`, diabetes⟩ misuses `has_part` to describe care; a correct variant: ⟨diabetes, `focus_of`, telehealth intervention⟩.

Table 7: ValidityScore evaluation (in percentage points).
**Bold**: better than Qwen3-32B baseline; **underlined bold**: best in column.
Percentages are computed per KG type with respect to its own #triples.

| KG type | Model size | #triples | Qwen3-32B as a judge | | | | Gemini 2.0 Flash as a judge | | | |
|---|---|---|---|---|---|---|---|---|---|---|
| | | | yes (Validity-Score) | maybe | no | miss-ing | yes (Validity-Score) | maybe | no | miss-ing |
| Qwen3-32B (baseline) | 32B | 515,460 | 43.0 | 24.1 | 31.4 | 1.5 | 66.0 | 33.6 | 0.3 | < 0.1 |
| UMLS Seed KG | N/A | 28,533 | **53.4** | 10.0 | 34.7 | 1.9 | **82.5** | 16.5 | 1.1 | < 0.1 |
| GRAPHMERT | 80M | 138,201 | **68.7** | 18.3 | 10.8 | 2.2 | **81.3** | 18.5 | 0.2 | < 0.1 |
| Reference LLM KGs | | | | | | | | | | |
| Grok 4 Fast | 1.7T | 100,000 | **56.9** | 21.9 | 21.2 | < 0.1 | **68.8** | 30.9 | 0.3 | < 0.1 |
| Qwen3-14B | 14B | 100,000 | 31.8 | 21.9 | 46.2 | 0.1 | 53.4 | 45.7 | 0.9 | < 0.1 |

- *Overlinking (redundant relations):* For $\langle \text{obesity}, \text{diabetes} \rangle$, the LLM proposed `is_modification_of`, `part_of`, `plays_role`, `cause_of`, `causative_agent_of`, `has_component`, `has_pathological_process`. However, only `associated_with` is valid.

- *Spurious co-occurrence link:* $\langle \text{diabetes}, \texttt{has\_method}, \text{mammography} \rangle$. The entities merely co-occurred in an abstract; mammography is not used for diabetes, yet the LLM infers a spurious relation.

*"Spurious co-occurrence link"* illustrates a disadvantage of local KG extraction methods vs. global methods that capture how concepts are distributed and connected across the full dataset. As a result, spurious links between concepts that rarely co-occur remain weak and are less likely to be predicted.

**Implication:** Broad, pre-trained LLMs tend to project general associations into domain-specific relations, yielding invalid triples. High-stakes domains benefit from compact, domain-tailored models and pipelines that enforce ontology and provenance.

We want to note that a higher FActScore* of Qwen3-14B in some columns does not reflect a stronger KG. Qualitatively, Qwen3-14B often produces short, generic tails (e.g., "disease," "complication") that the judge can accept as context-compatible while largely ignoring the head and relation, which is reflected in its lower ValidityScore. Qwen3-32B, in contrast, tends to generate longer, more specific tails that try to capture the full semantic relation; these are penalized more harshly by FActScore* but remain more ontology-consistent overall.

### 6.2.2 ValidityScore

**Validity check on the whole KG:** To assess the semantic quality of triples in GRAPHMERT- and LLM-extracted KGs, we conduct a validity check: Test the validity of each triple in the graph with Qwen3-32B and Gemini 2.0 Pro as the judge LLMs.

Table 7 summarizes the validity checks across KGs. GRAPHMERT attains a markedly higher "yes" rate, i.e., ValidityScore, than the LLM baseline (68.8% vs. 43.0% with Qwen3-32B), indicating substantially cleaner relation usage and predicate hygiene. Overall, the LLM KG is less conservative (more details ahead) and less valid. The typical malformed triples are shown in Tables 8 (GRAPHMERT) and 9 (LLM), which illustrate the characteristic failures of each model.

**Validity check on a KG subset with GPT-5:** We also perform a validity check on small KG subsets with GPT-5 (Thinking). GPT-5 is particularly strict in evaluating consistency of triples and whether the relation is used in the correct direction, aligning with our goal of obtaining dependable judgments. We

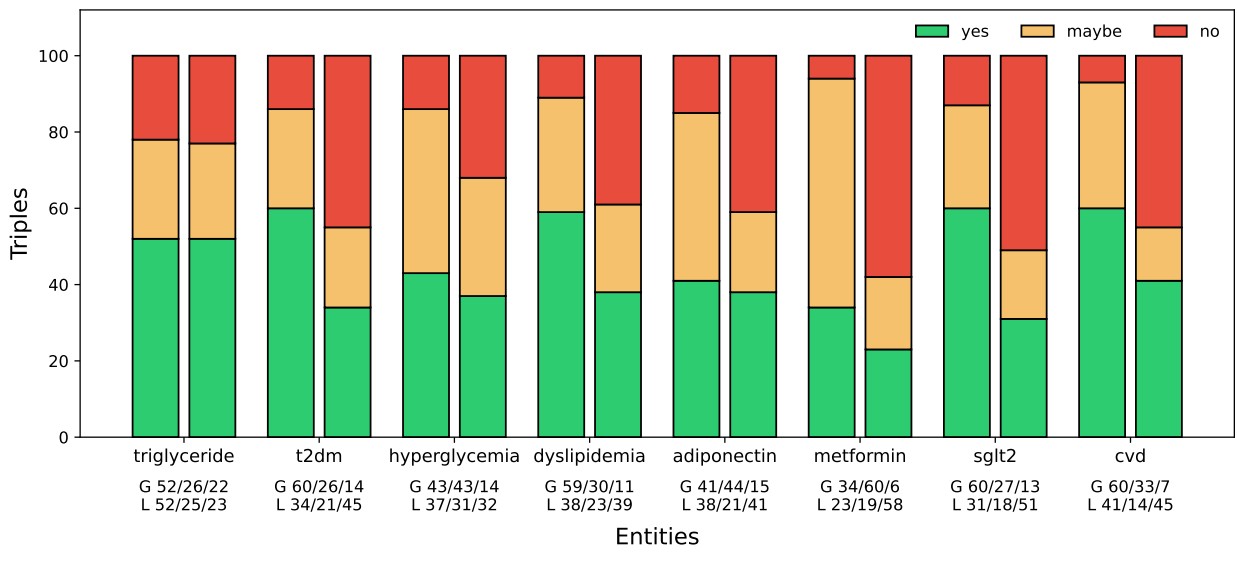

Figure 12: Validity check with GPT-5 Thinking, 100 random triples per keyword. The keywords are lined on the $x$-axis. Within each category, the left bars represent GRAPHMERT triples and right bars Qwen3-32B LLM triples. The captions below the labels show yes/maybe/no split counts for GLM and LLM.

Table 8: GRAPHMERT-extracted KG triples: Malformed triple examples

| Triples with an incomplete tail | Comment |
| --- | --- |
| CKD risk prediction model, associated_with, validated | adjectival tail |
| gestational diabetes mellitus, associated_with, twin | should be twin pregnancy |
| elastin, has_modification, carbam | should be carbamylation |
| **Triples with overstated causality** | **Comment** |
| insulin resistance in CKD, cause_of, metabolic syndrome | IR is a criterion |
| insulin resistance in CKD, cause_of, vascular disorder | vague tail |
| diabetes mellitus, cause_of, tuberculosis | cause is mycobacterium; DM is risk |
| **Triples with vague tails** | **Comment** |
| blind patients with DM, associated_with, diabetic retinopathy | head is a cohort |
| atherosclerotic cardiovascular disease, associated_with, lifestyle | vague tail |
| **Triples with predicate misuse** | **Comment** |
| atherosclerotic cardiovascular disease, has_causative_agent, apolipoprotein b | should be apolipoprotein B–containing lipoproteins |
| type 2 diabetes mellitus, associated_with, parasitic infection | weak/unspecific link |
| glucocorticoid-induced hyperglycaemia, has_finding_site, tissue | wrong site (should be blood) |

retrieve 100 random triples across seven diabetes-related keywords: *T2DM, hyperglycemia, dyslipidemia, adiponectin, metformin, SGLT2, and CVD*. Each triple is assigned a *yes/maybe/no* verdict (Fig. 12). The keywords are drawn from the 2000-word dataset-level summary generated with Gemini 2.5 Pro, representing some of the most frequent and clinically relevant terms related to diabetes and its comorbidities.

Overall, the GRAPHMERT KG consistently produces a higher proportion of valid (*yes*) triples and fewer incorrect (*no*) triples across all keywords, whereas the LLM KG shows more relation misuse and ontology violations, reflected in a greater share of *maybe* and *no* verdicts. This highlights the more conservative but domain-appropriate character of the GRAPHMERT KG compared to the noisier LLM KG. According to

Table 9: Qwen3-32B LLM-extracted KG triples: Malformed triple examples

| Triples with reversed relation | Comment |
|---|---|
| cryptogenic stroke, causative_agent_of, paradoxical embolism | reversed causality |
| ischemic stroke, cause_of, ps | PS — Protein S deficiency; reversed |
| glucocorticoids, has_part, steroidogenesis | steroidogenesis produces GCs |

| Triples with predicate misuse | Comment |
|---|---|
| ischemic stroke, cause_of, telomere length | biologically wrong |
| elastin, finding_site_of, vascular smooth muscle cell | cells aren't "found" in elastin |
| CKD awareness, has_component, race/ethnicity | CKD = chronic kidney disease; ontologically wrong |
| lockdown, associated_with, bone mineral density | mismatches ontological types |
| elastin, has_pathological_process, matrix metalloproteinase | metalloproteinase is an enzyme |

| Triples with ill-defined target | Comment |
|---|---|
| chronic kidney disease (CKD), cause_of, renal retinopathy | tail isn't a standard retinal disorder term |

Table 10: FActScore* after validity check with Qwen3-32B as a judge (in percentage points)

| KG type | #triples | Context only | Context and General truth |
|---|---|---|---|
| Qwen3-32B LLM (baseline) | 221,632 | 55.6 | 71.9 |
| GRAPHMERT | 96,020 | **76.9** | **80.2** |

verdicts *(a very short reason why)*, the LLM often violates ontology, confusing methods with diseases, and misuses relations, with the two errors reinforcing each other. Next, we analyze the main errors of each KG separately.

**GraphMERT:**  The main issues in GRAPHMERT triples are vagueness and incomplete tails, though they remain domain-appropriate. Tail incompleteness arises when the helper LLM accepts an incomplete token as a valid tail during token combination. We observe this effect across all helper LLMs we tested, including Gemini Flash 2.0 and 2.5. Tail vagueness occurs when GRAPHMERT does not rank the required tail tokens within its top-20 predictions; the helper LLM then stitches together a completion that is contextually acceptable but semantically weak. This also explains most of the cases of overstated causality and predicate misuse: missing key tokens required for high-quality tail completion, the LLM still attempts a plausible but semantically weak completion. A practical mitigation is to exclude low-information 128-token sequences in the prediction stage (e.g., segments dominated by measurements, dosages, or numbers). We currently do not filter these sequences, which may contaminate the extracted KG.

**LLM:**  In contrast, we observe that the LLM misinterprets UMLS biomedical relations and substitutes them with its broader internal knowledge, resulting in approximations that violate the ontology. Designing prompts to fully explain all relations is impractical. Our experiments show that even multiple examples fail to steer the model consistently: It defaults to its own internal semantics. We also observe systematic relation reversal, consistent with findings in (Berglund et al., 2024). More broadly, extracting well-formed triples requires capturing semantic rather than syntactic representations, which is challenging for LLMs trained primarily on surface text.

**Examples:**  To exemplify the difference between GRAPHMERT and LLM KGs explicitly, for the same prompt, we provide Tables E3, E4, E5, E6 with GPT-5 verdicts of triples sampled from GRAPHMERT-extracted KG and LLM-extracted KG in Appendix E.

**FactScore after validity check:**  We further assess the factuality of KGs after applying the validity check, where all invalid triples are removed before rerunning the FActScore evaluation. The results, summarized in Table 10, show that LLM-based cleaning, which primarily eliminates malformed triples, further improves

Table 11: GraphRAG KG evaluation on filtered ICD-Bench dataset, organized by difficulty levels

| Difficulty | Trivial | Very Easy | Easy | Medium | Hard | Very Hard | Average |
|---|---|---|---|---|---|---|---|
| # Questions | 20 | 26 | 8 | 6 | 2 | 7 | 69 |
| Qwen3-32B LLM KG (baseline) | 56.7 | 68.0 | 33.3 | 38.9 | 0.0 | 9.5 | 50.2 |
| Seed KG | 56.7 | 71.8 | 37.5 | 55.6 | 0.0 | 4.8 | 53.1 |
| GRAPHMERT | 66.7 | 79.5 | 50.0 | 50.0 | 0.0 | 0.0 | **59.4** |

factuality, with GRAPHMERT reaching 76.9%. The larger FActScore gain for the LLM KG in the *Context & General truth* setting compared to the *Context only* case indicates that the LLM relies heavily on its background knowledge when generating KGs.

### 6.3 Graph-level Evaluation

To evaluate the extracted KG, we apply GraphRAG to the filtered benchmarks and assess accuracy based on the success rate of the model in answering the questions. We compare the accuracy by using different KGs as the primary source of information in response generation.

We evaluate the GRAPHMERT KG against the baseline LLM KG on the filtered endocrinology subset of ICD-Bench, reporting the average question-answering accuracy across three runs. The evaluation is stratified by difficulty level, a label from the original benchmark. Our filtered test set includes 20 trivial questions (corresponding to 1-hop QA), and a total of 49 questions across difficulty levels ranging from very easy to very hard, based on the difficulty definitions from ICD-Bench. The detailed results are presented in Table 11. We also test the KGs on popular medical benchmarks, which are described in Appendix F. The findings demonstrate that the GRAPHMERT KG consistently outperforms the baseline. Across the whole filtered subset, our framework achieves an overall accuracy gain of 9.2% on ICD-Bench, and 1.7% to 3.7% gain on other medical benchmarks. This highlights the advantages of the GRAPHMERT KG for downstream medical question-answering tasks.

### 6.4 Ablations

To validate our design choices and understand the contribution of different components, we conduct a series of ablation studies. We analyze the sensitivity of GRAPHMERT to its core hyperparameters, its robustness to the density of the seed KG, and the effect of similarity and fact-checking.

#### 6.4.1 Training GraphMERT with Less-relevant Injections

Our framework relies on two key similarity thresholds: the injection threshold $\alpha$, which determines the relevance threshold of seed triples used for training, and the acceptance threshold $\beta$, which filters the final triples generated by the pipeline. To find the optimal configuration, we perform a grid search over a range of values for both parameters.

Hyperparameter $\alpha$ controls the trade-off between the quality and quantity of knowledge injected during training. A higher $\alpha$ imposes a stricter relevance filter, ensuring high-quality injections but limiting their volume and diversity. Hyperparameter $\beta$ serves as a grounding function for the final triples on the source sentence. The baseline LLM KG is filtered in each experiment to include only the relations available for the corresponding $\alpha$ value, ensuring an equitable evaluation.

The results are presented in Table 12 and Fig. 13. We observe that performance peaks at $\alpha = 0.55$ and $\beta = 0.67$. Lower $\alpha$'s likely introduce noisy, contextually irrelevant triples that degrade performance. Conversely, higher $\alpha$'s appear overly restrictive, preventing the model from leveraging a sufficient breadth of knowledge. In addition, $\beta = 0.67$ gives good results, underscoring the importance of a final quality check on generated triples. A higher optimal $\beta$ strengthens the importance of cross-document understanding for triple generation, which is not possible in LLM-generated KGs. The optimal configuration achieves a 9.2% improvement over the baseline LLM KG.

Table 12: GraphRAG accuracies for different $\alpha$ and $\beta$. Performance gain versus Qwen3-32B LLM KG is shown in parentheses.

| $\alpha$ value | Qwen3-32B LLM KG | Seed KG | $\beta = 0.62$ | $\beta = 0.65$ | $\beta = 0.67$ | $\beta = 0.69$ |
|---|---|---|---|---|---|---|
| 0.50 | 58.0 | 51.7 | 55.1 (-2.9) | 55.1 (-2.9) | 54.6 (-3.4) | - |
| 0.53 | 55.1 | 51.2 | 60.4 (+5.3) | 55.1 (+0.0) | 58.9 (+3.8) | - |
| 0.55 | 50.2 | 53.1 | 57.5 (+7.3) | 59.2 (+9.0) | 59.4 (**+9.2**) | 52.2 (+2.0) |
| 0.57 | 51.2 | 53.1 | 56.0 (+4.8) | 54.6 (+3.4) | 52.2 (+1.0) | - |
| 0.60 | 53.1 | 49.8 | 53.6 (+0.5) | 52.7 (-0.4) | 53.6 (+0.5) | - |

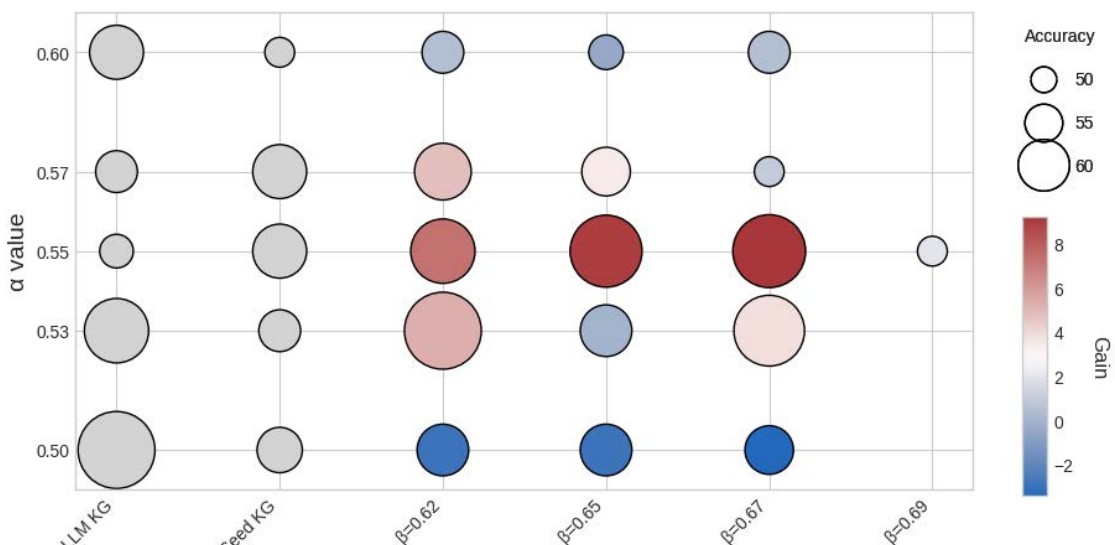

Figure 13: GraphRAG accuracies for different $\alpha$ and $\beta$. Bubble size corresponds to absolute accuracy and color indicates the accuracy gain relative to the Qwen3-32B LLM KG baseline (red denotes positive gain, blue denotes negative gain).

### 6.4.2 Training GraphMERT with a Smaller Seed KG

To evaluate the dependency of GRAPHMERT on the density of provided knowledge, we conduct an experiment to measure its performance with a sparser seed KG. A robust system should be able to function effectively even when the initial knowledge base is incomplete.

Using the optimal hyperparameters identified previously ($\alpha = 0.55$ and $\beta = 0.67$), we simulate varying levels of knowledge sparsity by randomly removing 25%, 50%, and 75% of the triples from the original seed KG before executing our pipeline.

The results, shown in Table 13, indicate that while performance generally decreases as the seed KG becomes sparser, our framework remains effective. Even with 75% of the seed knowledge removed, GRAPHMERT still outperforms the baseline LLM KG by 3.86%. This finding highlights the robustness of our approach, demonstrating that it can effectively leverage even a sparse set of seed triples to generate a high-quality KG.

### 6.4.3 Ablating GraphMERT Components

Next, we ablate architectural components and the training objective:

**No-span MLM/MNM:** Replace the span masking objective with a simple one-token MLM/MNM masking objective.

Table 13: GraphRAG accuracies upon dropping part of the seed KG. Performance gain versus Qwen3-32B LLM KG is shown in the last column.

| KG type | GraphRAG accuracy | Gap vs. Qwen3-32B LLM KG |
|---|---|---|
| GRAPHMERT | 59.4 | **+9.2** |
| GRAPHMERT (remove 25% of seed KG) | 54.6 | +4.4 |
| GRAPHMERT (remove 50% of seed KG) | 56.5 | +6.3 |
| GRAPHMERT (remove 75% of seed KG) | 54.1 | +3.9 |

Table 14: GraphRAG accuracies when ablating GRAPHMERT features

| KG type | Trivial | Very Easy | Easy | Medium | Hard | Very Hard | Average |
|---|---|---|---|---|---|---|---|
| GRAPHMERT | 66.7 | 79.5 | 50.0 | 50.0 | 0.0 | 0.0 | **59.4** |
| GRAPHMERT (no-span MLM/MNM) | 66.7 | 76.9 | 29.2 | 61.1 | 0.0 | 9.5 | 58.0 |
| GRAPHMERT (no H-GAT) | 65.0 | 68.0 | 33.3 | 55.6 | 0.0 | 0.0 | 53.1 |
| GRAPHMERT (no dropout) | 56.7 | 65.4 | 45.8 | 50.0 | 0.0 | 14.3 | 52.2 |

Table 15: FActScore* KG evaluation with Qwen3-32B (in percentage points) when ablating GRAPHMERT features

| KG type | #triples | Context only | Context and General truth |
|---|---|---|---|
| GRAPHMERT | 138,201 | **69.8** | **72.2** |
| GRAPHMERT (no-span MLM/MNM) | 185,768 | 69.1 | **72.2** |
| GRAPHMERT (no H-GAT) | 155,311 | 68.3 | 69.9 |
| GRAPHMERT (no dropout) | 138,363 | 69.0 | 70.9 |

**No H-GAT:** Switch off graph attention. This implies training without relation embeddings; the model predictions are purely syntactic.

**No dropout:** Switch off dropout on relation embeddings.

Before discussing the ablation results, we outline the main observations for each one:

- In the "no H-GAT" ablation, we observe a large number of irrelevant tokens in the top-$k$ predicted tokens, with commas and articles being primarily predicted in the top 3. This is supported by findings by García-Silva et al. (2023): They attempted to complete triples with top-$k$ tokens extracted from the BERT model in a purely syntactic manner, and had to rely on a stop word list to make the results usable.

- Disabling dropout leads to overfitting on the seed KG vocabulary; hence, less diverse tails.

- Training with a no-span MLM/MNM objective produces simpler tail completions (1–2 tokens long), because each individual candidate is not well aligned with the others.

Table 14 demonstrates that the full GRAPHMERT KG configuration achieves the highest performance. While the full model achieves the best results, the variant without span-masking performs only slightly worse. In contrast, the removal of either dropout or H-GAT leads to a substantial decrease in accuracy, underscoring their importance for robust performance.

Table 15 demonstrates that the full model FActScore* is the highest: removing dropout or H-GAT lowers acceptance and increases rejections. GRAPHMERT without span-masking reaches the same FActScore* in "Context and General truth" case.

Table 16: Validity check with Qwen3-32B (in percentage points) when ablating GRAPHMERT features

| KG type | #triples | yes | maybe | no |
|---|---|---|---|---|
| GRAPHMERT | 138,201 | 68.8 | 18.3 | 10.8 |
| GRAPHMERT (no-span MLM/MNM) | 185,768 | **69.3** | 18.2 | 10.3 |
| GRAPHMERT (no dropout) | 138,363 | 66.0 | 19.2 | 12.0 |
| GRAPHMERT (no H-GAT) | 155,311 | 67.5 | 17.5 | 12.3 |

Table 17: FActScore* KG evaluation (in percentage points) of GRAPHMERT Iteration 2 with Qwen3-32B as a judge

| KG type | #triples | Context only | Context and General truth |
|---|---|---|---|
| GRAPHMERT | 138,201 | 69.8 | 72.2 |
| GRAPHMERT with Qwen3-32B LLM KG instead of UMLS KG | 589,338 | 74.4 | 76.7 |
| GRAPHMERT with expanded KG instead of UMLS KG | 492,271 | **77.0** | **79.0** |

Table 18: ValidityScore KG evaluation (in percentage points) of GRAPHMERT Iteration 2 with Qwen3-32B as a judge

| KG type | #triples | yes (ValidityScore) | maybe | no |
|---|---|---|---|---|
| GRAPHMERT | 138,201 | **68.7** | 18.3 | 10.8 |
| GRAPHMERT with Qwen3-32B LLM KG instead of UMLS KG | 589,338 | 58.0 | 30.9 | 8.7 |
| GRAPHMERT with expanded KG instead of UMLS KG | 492,271 | 65.1 | 24.0 | 8.1 |

GRAPHMERT without span-masking achieves the best acceptance (69.3% yes, 10.3% no) (Table 16), but, according to GraphRAG, the KG remains less informative overall. In effect, the KG obtained from this variant tends to be populated with trivially correct tiples, i.e., ⟨diabetes, is_a, disease⟩. This leads to a higher rate of successful tail completion: 185k against 138k with span masking. Such short, obvious facts pass a validity check, manifesting in a higher ValidityScore. However, this simplicity comes at a cost: No-span-masked completions lack the nuance and granularity provided by span masking, resulting in poorer coverage and a loss of fine-grained domain details. This trade-off stresses the importance of evaluating KGs both at the graph and triple levels. We advise employing token-level MLM/MNM when the simplicity of triples is not a limitation.

### 6.4.4 GraphMERT Iteration 2: Retraining

Next, we address two questions:

1. What if a seed KG for the domain of interest is not available?

2. Can we further improve and expand the GRAPHMERT KG through iterative retraining?

**Retraining with the LLM KG replacing the seed KG.** A key limitation of the pipeline described so far is its reliance on a seed KG, which may not be available in some domains. A practical alternative is to generate a seed KG using an LLM and apply GRAPHMERT to it, rather than construct the entire KG using an LLM. To test this approach, we replace the gold-standard UMLS seed KG with the Qwen3-32B LLM-generated KG and rerun the GRAPHMERT pipeline.

**Retraining with the expanded KG replacing the seed KG.** After running the pipeline once, GRAPH-MERT extracts ontology-consistent triples grounded in the source text. We can then augment the UMLS

seed KG with these extracted triples and retrain the model on this expanded KG. This provides additional examples and may, in principle, yield further performance gains.

The FActScore* and ValidityScore results for these two experiments are shown in Tables 17 and 18. When using the Qwen3-32B LLM-generated KG instead of the UMLS seed KG, the GRAPHMERT pipeline attains a higher FActScore*, surpassing that obtained with the UMLS seed KG. However, the ValidityScore drops notably by 10.1%, which is expected since the LLM-generated KG does not strictly follow the ontology, resulting in lower-quality examples. Ultimately, although, when available, using a gold-standard seed KG is preferable, generating a seed KG with an LLM from the training dataset and subsequently training GRAPHMERT remains a practical and effective alternative when a seed KG is not available.

When retrained with the expanded KG rather than the UMLS KG, GRAPHMERT achieves the highest FActScore*, while the ValidityScore decreases slightly (by 3.6%). Notably, both alternative "seed KGs" (the LLM-generated and expanded versions) also lead to a larger extracted KG size, which may be attributed to their larger size.

## 7  Discussion, Limitations, and Future work

**Discussion:**  Our results show that GRAPHMERT yields higher factuality and validity of triples than LLM-based KG extraction while preserving ontology fidelity. We observe a consistent difference between GRAPHMERT and LLM KGs in relation usage and predicate hygiene. GRAPHMERT generally employs relations correctly, aligns tails with heads, and preserves biomedical categories (diseases, syndromes, complications). The LLM KG often misuses or reverses the direction of relations, mixes categories in ontology-violating ways (e.g., socio-economic with biomedical), and produces inverse/ill-typed statements. The GRAPHMERT KG has far fewer ontology violations and hews closer to UMLS. The GRAPHMERT KG vocabulary is more conservative: The conservativeness can be explained based on the limited scope of the seed KG vocabulary and its tendency to restate head tokens (mimicking UMLS tautological triples). While generally more accurate, GRAPHMERT may still produce incomplete or vague triples.

We attribute the difference in factuality and validity of triples to the usage of semantic relation embeddings, which move predictions toward ontology-aligned ones. Triple-level error analysis shows GRAPHMERT may sometimes extract incomplete and vague but often domain-appropriate tails with factual relations, whereas LLMs usually produce diverse tails but frequently misuse or reverse the direction of relations. Our graph-level evaluation may conflate the KG signal with backbone model knowledge under GraphRAG; future work will include graph-level metrics that isolate the contribution of KGs as well as relation-aware retrieval.

**Limitations:**  The main limitation of GRAPHMERT is its reliance on the seed KG. First, running the framework requires a high-quality seed with 100-1000 examples per relation. Second, once training is complete, the relation set is fixed. Hence, adding new relations requires retraining. A further limitation is its dependence on a helper LLM for tail combination, which introduces occasional incompleteness in the extracted triple tails. If the helper LLM lacks sufficient expertise in the target domain, it may struggle to construct accurate long multi-token completions, which can limit the reliability of the extracted KG. It may also reintroduce a risk of hallucination or non-factual tail combinations, even though the triple candidates are constrained by the GRAPHMERT model. This can be mitigated by using domain-specialized helper models, when available, or applying simple post-filters (e.g., ontology checks, part-of-speech filters) to discard low-confidence tails. Alternatively, one can select only triples that have passed ValidityCheck. As a neural model, GRAPHMERT also tends to prioritize frequent entities, potentially overlooking rare but meaningful ones. Further, our experiments are based on a small 125M-token domain-specific text corpus. In principle, GraphMERT architecture (Transformer + graph injection) should scale to larger text corpora and more relation types, but billion-token training regimes remain to be studied systematically. In addition, we did not evaluate the model's ability to predict or handle numerical tokens. Finally, its robustness to unseen entities remains untested; extracting entirely novel concepts from new texts would likely require fine-tuning, limiting adaptability in fast-changing domains.

**Future work:** We plan to improve the KG quality by unifying entity spellings, replacing the fixed top-$k$ limit with high-probability token selection. We aim to extend GRAPHMERT to direct multi-token span prediction in the semantic space, reducing reliance on the helper LLM for tail token combining. Further, we plan to conduct more rigorous graph-level evaluations, since GraphRAG alone often blends KG information with model knowledge, and does not guarantee retrieval of the most relevant subgraph, as its retrieval is entity-guided and relations play only a secondary role. It may also be valuable to investigate how the GRAPHMERT evaluation loss correlates with the quality of the extracted KGs. We also plan to extract and evaluate KGs for other domains and employ them in various downstream applications.

## 8 Conclusion

We proposed a new framework for automatically extracting domain-specific KGs from unstructured, sentence-level text. Central to this work is our idea of encoding both semantic and syntactic information into textual chain graphs, a new representation we introduced. To operate in this space, we presented GRAPHMERT, a transformer-based model that unifies an encoder-only architecture with graph attention. Together, these contributions enable the distillation of explicit semantic structures from trained neural networks, bridging neural and symbolic representations and advancing interpretable, reliable KG construction.

We also outlined KG-powered applications, underscoring that future progress hinges on reliable, factual KGs and other high-level explicit abstractions that embody collective knowledge while remaining compatible with AI inference and downstream tasks. We argue that neural-KG integration is a key step toward domain-specific superintelligence. Looking ahead, we foresee the research community embracing the neurosymbolic paradigm, where explicit, auditable, and evolving KGs complement neural inference that is approximate, efficient, and capable of handling ambiguity.

### Acknowledgments

This work was supported by NSF under Grant No. CNS-2216746. The authors are pleased to acknowledge that the work reported on in this paper was substantially performed using the Princeton Research Computing resources at Princeton University. Princeton Research Computing is a consortium of groups including the Princeton Institute for Computational Science and Engineering (PICSciE) and Research Computing at Princeton University.

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

# Appendices

## A Extended background

### A.1 KG with Neural Network: Unifying the Representations

In many tasks, the solution is not to pit symbolic and neural approaches against each other but to combine them in hybrid, modular systems. Neural networks discover statistical patterns through gradient descent; symbolic layers then manipulate extracted structure efficiently (Garcez & Lamb, 2023). Within such a neurosymbolic framework, KGs can naturally serve as symbolic memory and rule repositories. Coupled with neural networks, they provide modularity and cross-representation translation. By decoupling learning (implicit, neural) from reasoning (explicit, symbolic), KGs address interpretability, verifiability, and factuality gaps in modern AI systems that are dominated by neural approaches (Besold et al., 2017).

### A.2 KG Applications in Neurosymbolic Frameworks

KGs with high-quality expert knowledge enable a range of downstream applications.

**Explicit reasoning and knowledge transfer:** Symbolic inference over KGs is interpretable (Cheng et al., 2024), and explicit rules can also be transferred across related domains. For instance, retrieving all drugs ($d$) that (1) target a protein ($p$) associated with Type 2 Diabetes and (2) are approved by the FDA, expressed as the conjunctive query:

$$\exists\, p, d\ \big(\text{associated\_with}(p, \text{Type\_2\_Diabetes}) \,\wedge\, \text{targets}(d, p) \,\wedge\, \text{approved\_by}(d, \text{FDA})\big) \tag{12}$$

**Discoveries:** KGs uncover hidden insights by linking apparently unconnected concepts through explicit graph paths. This connectivity underpins applications, notably recommendations, where relevant entities are suggested based on graph neighborhoods, and analogical discovery, where cross-domain parallels emerge through reasoning over paths. In biomedicine, for example, a path like drug $\rightarrow$ protein target $\rightarrow$ pathway $\rightarrow$ disease can reveal repurposing opportunities, e.g., a hypertension drug acting on pathways implicated in diabetes.

**KGs for Domain-specific Superintelligence:** A natural scarcity of high-quality text sources may impede the scalability of ever-larger LLMs, necessitating a paradigm shift that favors smaller language models with domain-specific expertise. This would shift focus from general-purpose artificial general intelligence to domain-specific superintelligence (Bostrom, 2014; Szocik et al., 2020). A recent study by Dedhia et al. (2025) illustrates one path toward this goal by fine-tuning a small language model on multi-hop KG paths

translated into composite natural language statements. They show that longer KG paths yield stronger domain reasoning. From this perspective, conventional text datasets resemble one-hop knowledge that often leads to surface-level understanding. In this context, a high-quality, domain-specific KG that explicitly encodes knowledge through concise and unambiguous semantic relationships can provide a useful inductive bias toward deeper reasoning.

## B    Extended Examples of LLM responses

---

**Claude Sonnet 4.5:** Based on the provided sequence, the medical KG triple should be completed as: (pediatric chronic kidney disease (pckd), has_finding_site, kidney)

Reasoning:
The primary and defining finding site for chronic kidney disease is the kidney itself, as explicitly stated in the passage: "Chronic kidney disease (CKD) is a renal disorder." While the passage does discuss secondary neuroimaging findings in the cerebellar gray matter, these are described as:

- Subtle and group-level abnormalities
- Heterogeneous observations
- Not diagnostic
- Modest and regional alterations

In contrast, the kidney is the primary pathological site where CKD occurs by definition. The cerebellar changes represent associated findings or potential complications, but they are not the primary finding site of the disease entity itself. For medical knowledge graphs, the has_finding_site relation typically indicates the primary anatomical location where the disease process occurs, which for CKD is definitively the kidney. If you needed to represent the cerebellar findings, that would be better captured with a different relation such as has_associated_finding or may_affect, rather than has_finding_site.

---

**GPT-5:** Based on the medical knowledge graph triple format and the provided sequence, the completed triple would be: (chronic kidney disease, has_finding_site, cerebellar gray matter)

This is based on the text where chronic kidney disease (CKD) is associated with observable findings in the cerebellar gray matter, specifically mentioning reductions in cerebellar gray-matter volume, cortical thickness, and diffusion metrics.

---

**Grok 4:** Based on the medical knowledge graph triple format and the provided sequence, the completed triple would be: (chronic kidney disease, has_finding_site, cerebellar gray matter)

The triple (chronic kidney disease, has_finding_site, cerebellar gray matter) is completed based on the provided sequence, which states that neuroimaging studies of chronic kidney disease (CKD) patients have identified subtle abnormalities in the cerebellar gray matter, such as reductions in volume, cortical thickness, and diffusion metrics. The relation "has_finding_site" indicates the anatomical location where these abnormalities are observed, making "cerebellar gray matter" the appropriate tail entity.

---

## C    Training Data

We exclude some relations from the UMLS KG that add little semantic value, i.e.: (i) custom-defined mappings of outdated-to-new UMLS relations for backward compatibility, which cannot be inferred from external data; (ii) relations used only for cross-vocabulary mappings; (iii) relations where tails are largely redundant with respect to heads; and (iv) relations with very few possible tails. For example, relation has_associated_finding is a redundant relation:

- family history of diabetes, `has_associated_finding`, diabetes mellitus

Table C1: Excluded UMLS relations

| | | |
|---|---|---|
| acted_on_by_process | has_scale_type | regulated_by |
| active_ingredient_of | has_specimen | replaced_by |
| associated_procedure_of | has_subject_relationship_context | replaces |
| basis_of_strength_substance_of | has_temporal_context | was_a |
| component_of | inverse_was_a | has_intent |
| consider_from | mapped_from | referred_to_by |
| direct_device_of | mapped_to | refers_to |
| direct_substance_of | moved_to | characterizes |
| has_associated_finding | negatively_regulated_by | substance_used_by |
| has_finding_context | positively_regulated_by | specimen_source_topography_of |
| has_interpretation | possibly_replaces | specimen_substance_of |
| has_laterality | precise_active_ingredient_of | has_active_ingredient |
| has_realization | realization_of | has_property |

- family history of glaucoma, `has_associated_finding`, glaucoma
- parental history of diabetes, `has_associated_finding`, diabetes mellitus
- family history of hypertension, `has_associated_finding`, hypertensive disorder
- family history of cvd, `has_associated_finding`, congenital heart disease

In these cases, the tail subject is the same as the head subject. `has_laterality` is an example of a relation with very few possible tails, with almost all tails being "side".

```
(
    (
        diabetes[Abstract] OR diabetes[Body - All Words] OR diabetes[Body - Key
            Terms]
    )
  OR (
    ''diabetes mellitus''[MeSH Terms] OR ''diabetes insipidus''[MeSH Terms]
  )
)
OR diabetes[Title]
NOT (
  ''sars-cov-2''[MeSH Terms] OR ''covid-19''[MeSH Terms]
)
AND (
  medline[sb] AND ''2019/04/01''[PubDate] : ''2025/04/01''[PubDate]
)
```

Listing 1: PubMed search query

## D    KG Injection Algorithm

Input: Sequences with heads: Each head $h$ may have multiple triples $T = \langle h, r, t \rangle = T(seq)$ in sequence $seq$ (we cap this number at 40 triples to ensure enough diversity and leaving room for further dropping); triple embedding similarity score with respect to the sequence: $score(T)$; similarity matching threshold $\alpha$ (a hyperparameter).

Output: Each head has one injection $T(seq)$ or does not have any, in all sequences.

**Preprocessing**

1. Drop all triples with a score less than threshold $\alpha$.

2. Make all triples unique: If a triple matches multiple sequences, retain the triple $\tilde{T}$ with the highest score, i.e., in the sequence to which the triple is most relevant:

$$\tilde{T} = \underset{seq}{\arg\max}\, score(T(seq))$$

The second preprocessing step prevents overfitting in the semantic space on common triples.

**Triple selection for each head:**  To balance contextual relevance with relation diversity: 1st priority: maximize injection score, 2nd priority: maintain relation diversity. Relation diversity is measured by the number of unique triples that contain the relation.

Maximize diversity

1. Split relations into relation buckets based on the number of unique triples at step $k$ and assume that within each bucket all relations are equally diverse (e.g., $k = 20$ implies that relations with #triples 100-120, 120-140.., are treated as equally diverse).

2. Within each relation bucket, sort all triples by score regardless of relation.

3. Start with the lowest-numbered bucket (rarest relations). Within it, start with the triple with the highest score and retain only it for its head, removing all other matched triples, which may have a higher score but may be in a higher relation bucket. As a result, one of the rarest possible relations in the dataset would survive for this head, increasing relation diversity overall.

Maximize score then diversity

1. Order triples by score.

2. Split into score buckets: Assume that within each score bucket, triples are equally good.

3. Then, within each score bucket, apply Maximize diversity.

Altogether, we group triples by how "low" the score is (higher scores are assigned to lower-bucket IDs). Then, within each score bucket, we favor relation types that are less frequent. Finally, we choose the highest-scoring triple for each head.

The algorithm is implemented using the Pandas framework and presented in Algorithm 1.

In our experiments, we use $score\_bucket\_size = 0.01$, $relation\_bucket\_size = 100$. The bucket size and relation bucket size are chosen under the assumption that, within each bucket, score and relation diversity remain approximately balanced.

---

**Algorithm 1:** Maximize Score then Diversity

---

**Input** : *df* (a Pandas DataFrame where each row comprises a triple $(h, r, t)$, unique
      *matched_head_id*, and an associated *score*), *score_bucket_size*, *relation_bucket_size*

**Output:** Filtered DataFrame *result*

---

$max\_s \leftarrow \max(df.score)$;

**foreach** *row* **in** *df* **do**
   $row.score\_bucket \leftarrow \lfloor (max\_s - row.score)/score\_bucket\_size \rfloor$;

$rel\_counts \leftarrow$ count occurrences of each relation_type $r$ in $df$;

**foreach** *row* **in** *df* **do**
   $rel\_count \leftarrow rel\_counts[row.r]$;
   $row.rel\_bucket \leftarrow \lfloor rel\_count/relation\_bucket\_size \rfloor$;

Sort *df* by ascending (*score_bucket*, *rel_bucket*) and by descending *score*;
$result \leftarrow [\,]$;
$seen\_heads \leftarrow \emptyset$;

**foreach** *row* **in** *sorted df* **do**
   **if** $row.matched\_head\_id \notin seen\_heads$ **then**
      append row to *result*;
      $seen\_heads \leftarrow seen\_heads \cup \{row.matched\_head\_id\}$;

`// Equivalent to df.drop_duplicates(subset="matched_head_id", keep="first")`
**return** *result*;

---

Table D1: Seed KG: Relation statistics for $\alpha = 0.55$ in the training split (Qwen3-32B)

| # | Relation | # injections | # | Relation | # injections |
|---|---|---|---|---|---|
| 1 | isa | 8627 | 15 | possibly_equivalent_to | 446 |
| 2 | inverse_isa | 5512 | 16 | has_component | 433 |
| 3 | cause_of | 1440 | 17 | due_to | 365 |
| 4 | interprets | 1268 | 18 | has_part | 350 |
| 5 | associated_finding_of | 1145 | 19 | has_modification | 310 |
| 6 | has_disposition | 1084 | 20 | associated_with | 254 |
| 7 | focus_of | 1038 | 21 | part_of | 211 |
| 8 | is_interpreted_by | 962 | 22 | plays_role | 194 |
| 9 | has_associated_morphology | 863 | 23 | occurs_before | 187 |
| 10 | causative_agent_of | 809 | 24 | has_clinical_course | 144 |
| 11 | finding_site_of | 741 | 25 | occurs_in | 138 |
| 12 | associated_morphology_of | 598 | 26 | same_as | 134 |
| 13 | has_method | 515 | 27 | has_causative_agent | 127 |
| 14 | has_finding_site | 477 | 28 | has_focus | 118 |

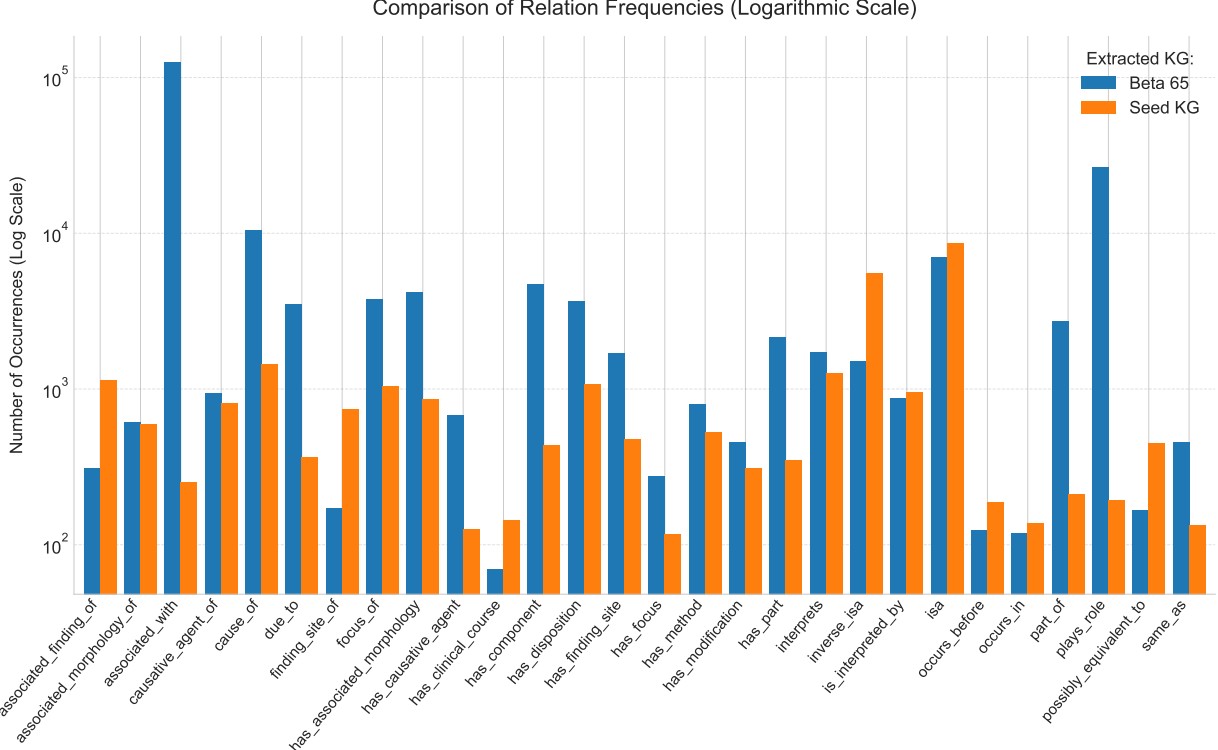

Figure E1: Relation distribution in the GRAPHMERT-extracted KG vs. seed KG. The shapes differ: While "isa" prevails in the seed KG, the GRAPHMERT KG is heavily skewed towards "associated_with." This reflects inclination of the helper LLM to select "associated_with" during relation matching as the most appropriate, given a sequence.

# E  Extracted KGs

### E.1   Relation Distribution: GraphMERT-extracted KG vs. Seed KG

Fig. E1 shows the relation distribution on a logarithmic scale. While "isa" is the most represented relation in the training data, in the to-be-extracted KG, the helper LLM tends to select "associated_with" most frequently during relation matching.

### E.2   Sanity Check

We ran a lightweight screening with GPT-5 Thinking on small, comparable samples from each KG. This screening should be viewed as complementary to benchmark-based verification, providing useful diagnostic signals but not replacing factual evaluation.

**Setup:**   For each KG, we retrieved all triples whose head contains the keywords "insulin-like growth factor 1 (IGF-1)" and "glucocorticoid receptor (GR)." These terms are highly relevant to diabetes, yield comparable sample sizes per KG (22–29), and remain small enough for human inspection.

> **Prompt.** Evaluate if these medical KG triples are valid (yes/no/maybe) and give a very short reason why: ⟨list of triples⟩.

Table E1: Example of GRAPHMERT-extracted triple with novel tail vocabulary from a 128-token sequence. The seed KG does not include the token "nlrp3" — it was learned and extracted from the text. "Pathway" is a token that GRAPHMERT learned from the training corpus. Here, "pathway" is implicitly supported: The phrase "specific inhibitors of NLRP3 inflammasome activation" links the activation to NLRP3-mediated signaling (i.e., the NLRP3 pathway).

**Sequence:** . . . inflammasome activation and regulation is highlighted, including its putative roles in adipose tissue dysfunction and insulin resistance. Specific inhibitors of NLRP3 inflammasome activation which can potentially be used to treat metabolic disorders are also discussed. Identifying a quantitative biomarker of neuropsychiatric dysfunction in people with HIV (PWH) remains a significant challenge in the neuroHIV field. The strongest evidence to date implicates the role of monocytes in central nervous system (CNS) dysfunction in HIV, yet no study has examined monocyte subsets in blood as a correlate and/or predictor of neuropsychiatric function in virally suppressed PWH. METHODS In 2 independent cohorts of virologically suppressed women with HIV (vsWWH. . .

| Head | Relation | Tail |
|---|---|---|
| inflammasome activation | associated_with | nlrp3 pathway |

Table E2: GPT-5 Thinking screening (counts with proportions)

| Keyword | KG | Yes | Maybe | No | Total |
|---|---|---|---|---|---|
| IGF–1 | LLM | 8 (0.28) | 3 (0.10) | 18 (0.62) | 29 |
| IGF–1 | GRAPHMERT | 15 (0.68) | 0 (0.00) | 7 (0.32) | 22 |
| GR | LLM | 4 (0.17) | 12 (0.50) | 8 (0.33) | 24 |
| GR | GRAPHMERT | 15 (0.54) | 7 (0.25) | 6 (0.21) | 28 |

Table E2 summarizes screening results. Tables E3, E4, E5, E6 provide GPT-5 Thinking responses for each triple in the GRAPHMERT and LLM KG samples.

Table E3: GRAPHMERT KG. Sanity check with GPT-5 Thinking. We select all KG triples that contain "insulin-like growth factor-1," 22 triples in total, *yes/no/maybe: 15/0/7.*

| Triple | Verdict | GPT-5 explanation |
|---|---|---|
| insulin-like growth factor 1, isa, peptide hormone | **yes** | 70-aa peptide hormone. |
| insulin-like growth factor 1, isa, growth factor | **yes** | Canonical growth factor. |
| insulin-like growth factor 1, associated_with, insulin resistance | **yes** | Correlates with IR. |
| insulin-like growth factor 1 receptor (IGF1R), plays_role, target | **maybe** | "Target" is vague. |
| insulin-like growth factor 1 (IGF-1), associated_with, diabetes mellitus | **yes** | Linked with DM. |
| insulin-like growth factor 1 (IGF-1), associated_with, biomarker | **maybe** | Biomarker is a role. |
| insulin-like growth factor 1 (IGF-1), associated_with, insulin | **yes** | Strong crosstalk. |
| insulin-like growth factor 1 (IGF-1), associated_with, type 2 diabetes | **yes** | Epidemiologic link. |
| insulin-like growth factor 1 (IGF-1), associated_with, insulin resistance | **yes** | Well supported. |
| insulin-like growth factor 1, plays_role, growth factor | **yes** | Acts as growth factor. |
| insulin-like growth factor 1 (IGF-1), plays_role, biomarkers | **maybe** | Plural/ontology mismatch. |
| insulin/insulin-like growth factor 1 (IGF-1) signaling pathway, has_component, IGF-1 receptor | **yes** | IGF-1R is in pathway. |
| insulin-like growth factor 1 (IGF-1), associated_with, insulin | **yes** | Duplicate; supported. |
| insulin-like growth factor 1, plays_role, growth hormone action | **yes** | Mediates GH effects. |
| insulin-like growth factor 1 receptor (IGF1R), plays_role, signaling receptor | **yes** | RTK signaling receptor. |
| insulin-like growth factor 1 receptor, plays_role, signaling | **yes** | Mediates signaling. |
| insulin-like growth factor 1 (IGF1), associated_with, androgen | **maybe** | Context-specific crosstalk. |
| insulin-like growth factor 1 (IGF1), associated_with, transcription | **maybe** | Indirect gene-expression effects. |
| insulin-like growth factor 1 (IGF-1), associated_with, hyperglycemia | **maybe** | Context dependent. |
| insulin-like growth factor 1 level, associated_with, growth hormone treatment | **yes** | GH raises IGF-1. |
| insulin/insulin-like growth factor 1 (IGF-1) signaling pathway, has_component, insulin receptor | **yes** | IR is in pathway. |
| insulin-like growth factor 1 (IGF1), plays_role, downstream target | **maybe** | Phrasing vague (GH downstream). |

Table E4: (Part 1 of 2). LLM KG (Qwen3-32B). Sanity check with GPT-5 Thinking. We select all KG triples that contain "insulin-like growth factor-1," 29 triples in total, *yes/no/maybe: 8/18/3*.

| Triple | Verdict | GPT-5 explanation |
|---|---|---|
| insulin-like growth factor 1, associated_with, diabetes | **yes** | Linked with DM. |
| insulin-like growth factor 1 (IGF-1), has_direct_substance, type 2 diabetes | **no** | Disease $\neq$ substance. |
| insulin-like growth factor 1, associated_with, insulin resistance | **yes** | Strong physiologic link. |
| insulin-like growth factor 1, associated_with, metabolic syndrome | **yes** | Levels track MetS. |
| insulin-like growth factor 1 (IGF-1), associated_with, chronic kidney disease | **yes** | GH/IGF axis altered. |
| insulin-like growth factor 1 (IGF-1), has_direct_substance, gestational diabetes mellitus (GDM) | **no** | Disease $\neq$ substance. |
| insulin-like growth factor 1 (IGF1), has_direct_substance, angiogenesis | **no** | Process, not substance. |
| insulin-like growth factor 1 (IGF-1), has_direct_substance, short-chain fatty acids (SCFAs) | **no** | Unrelated metabolites. |
| insulin-like growth factor 1 (IGF-1), has_component, insulin signaling pathway | **no** | Pathway $\neq$ component of ligand. |
| insulin-like growth factor 1 (IGF-1), plays_role, cardiovascular health | **maybe** | Broad/context-dependent. |
| insulin-like growth factor 1 (IGF-1), associated_with, bone metabolism | **yes** | Anabolic for bone. |
| insulin-like growth factor 1, has_part, growth hormone | **no** | GH regulates; not part. |
| insulin-like growth factor 1 (IGF-1), cause_of, prostate cancer (PCA) | **no** | Association $\neq$ causation. |
| insulin-like growth factor 1 (IGF-1), has_modification, left ventricular global longitudinal strain (LVGLS) | **no** | Clinical metric, not modification. |
| insulin-like growth factor 1 receptor, cause_of, epithelial-mesenchymal transition | **maybe** | Signaling can induce EMT. |
| insulin-like growth factor 1 (IGF-1), has_modification, HAAT-MSCs | **no** | Not a molecular modification. |
| insulin-like growth factor 1, has_direct_substance, testis | **no** | Organ produces IGF-1. |
| insulin-like growth factor 1, plays_role, endocrine-related cancers | **maybe** | Vague class-level claim. |

Table E4: (Part 2 of 2). LLM KG (Qwen3-32B). (continued)

| Triple | Verdict | GPT-5 explanation |
|---|---|---|
| insulin-like growth factor 1, associated_with, oocyte cohort quality | yes | Follicular IGF-1 correlates. |
| insulin-like growth factor 1 (IGF1), has_component, phosphoinositide 3-kinase (PI3K) | no | Downstream effector, not component. |
| insulin-like growth factor 1 receptor, isa, receptor | yes | Canonical RTK. |
| insulin-like growth factor 1 receptor, has_direct_substance, lactate dehydrogenase a | no | Not a receptor substance. |
| insulin-like growth factor 1 (IGF1), has_pathological_process, core body temperature | no | Not a pathology. |
| insulin-like growth factor 1 (IGF1), has_component, IGF-binding protein 3 (IGFBP3) | no | Binding partner, not component. |
| insulin-like growth factor 1 receptor, has_method, JAKS2/STAT3 pathway | no | Wrong relation/atypical pathway. |
| insulin-like growth factor 1 (IGF1), has_component, ZYG11A | no | Unrelated protein. |
| insulin-like growth factor 1 receptor (IGF-1R), has_part, nositide 3-kinase (PI3K) | no | PI3K is recruited, not part. |
| IGF-1 (insulin-like growth factor 1), has_direct_substance, PPAR$\gamma$ (peroxisome proliferator-activated receptor gamma) | no | transcription factor $\neq$ substance of IGF-1. |
| insulin-like growth factor 1, plays_role, cardiac development | yes | Supports cardiac growth. |

Table E5: GRAPHMERT KG. Sanity check with GPT-5 Thinking. We select all KG triples that contain "glucocorticoid receptor," 28 triples in total, *yes/no/maybe: 15/6/7*.

| Triple | Verdict | GPT-5 explanation |
|---|---|---|
| endothelial glucocorticoid receptor, plays_role, therapeutic role | maybe | Role is vague/context-dependent. |
| endothelial glucocorticoid receptor, plays_role, glucocorticoid signaling | yes | Mediates GC signaling in endothelium. |
| endothelial glucocorticoid receptor, plays_role, protective role | maybe | Protective effects reported but not universal. |
| glucocorticoid receptor (GR), plays_role, glucocorticoid signaling | yes | Canonical function. |
| glucocorticoid receptor, associated_with, insulin signaling | yes | Well-known pathway crosstalk. |
| glucocorticoid receptor, has_part, ligand-binding | maybe | Should be "ligand-binding domain." |
| endothelial glucocorticoid receptor, has_disposition, signaling | no | Wrong relation; participates in signaling. |
| glucocorticoid receptor, plays_role, glucocorticoid receptor signaling | yes | Tautologically true. |
| glucocorticoid receptor (GR), plays_role, hypothalamic signaling | maybe | HPA feedback involvement but broad. |
| endothelial glucocorticoid receptor, has_disposition, immunomodulator | no | "Immunomodulator" is an agent role, not a disposition. |
| glucocorticoid receptor (GR), plays_role, steroid signaling | yes | Nuclear steroid receptor pathway. |
| glucocorticoid receptor (GR), associated_with, glucocorticoids | yes | Binds GC ligands. |
| glucocorticoid receptor agonists, plays_role, therapeutic | yes | Used clinically. |
| glucocorticoid receptor (GR), plays_role, steroid hormone receptor | no | This is an is_a identity, not a role. |
| glucocorticoid receptor gene (NR3C1), associated_with, hormone receptor | maybe | Gene encodes one; relation imprecise. |
| glucocorticoid receptor, plays_role, signaling | yes | Ligand-activated signaling/TF activity. |
| glucocorticoid receptor (GR), associated_with, steroid | yes | Glucocorticoids are steroids. |
| selective glucocorticoid receptor agonists, plays_role, therapeutic role | maybe | Class intended for therapy; phrasing vague. |
| endothelial glucocorticoid receptor, cause_of, renal fibrosis | no | GR generally anti-fibrotic; causation unsupported. |
| selective glucocorticoid receptor modulators, plays_role, pharmacological modulator | no | Category is is_a, not a role. |
| glucocorticoid receptor, associated_with, glucocorticoids | yes | Duplicate; correct. |
| glucocorticoid receptor (GR), plays_role, glucocorticoid signaling | yes | Duplicate; correct. |
| glucocorticoid receptor $\alpha$, plays_role, steroid receptor | no | $\alpha$ isoform is_a steroid receptor, not a role. |
| glucocorticoid receptor (GR), associated_with, transcription | yes | Ligand-activated transcription factor. |
| glucocorticoid receptor, plays_role, signaling | yes | Duplicate; correct. |
| glucocorticoid receptor (GR), associated_with, hormone | yes | Responds to hormones (GCs). |
| glucocorticoid receptor (GR), associated_with, signaling | yes | Broad but true. |
| selective glucocorticoid receptor modulators, plays_role, pharmacological modulator | no | Duplicate; is_a, not a role. |

Table E6: LLM KG (Qwen3-32B). Sanity check with GPT-5 Thinking. We select all KG triples that contain "glucocorticoid receptor," 24 triples in total, *yes/no/maybe: 4/18/2.*

| Triple | Verdict | GPT-5 explanation |
|---|---|---|
| glucocorticoid receptor haploinsufficiency, cause_of, hypertension | yes | GR resistance drives mineralocorticoid excess → hypertension. |
| glucocorticoid receptor, associated_with, insulin resistance | yes | GC/GR signaling induces IR. |
| glucocorticoid receptor locus (GRL) polymorphisms, associated_with, type 2 diabetes (T2D) | yes | NR3C1 variants linked to T2D risk. |
| podocyte-specific glucocorticoid receptor knockout (GR pKO) mice, has_pathological_process, diabetic nephropathy | yes | Podocyte GR loss worsens DN phenotype. |
| glucocorticoid receptor, has_component, autophagy | no | Autophagy is a process, not a receptor component. |
| glucocorticoid receptor (GR), finding_site_of, liver | no | Relation reversed; GR is located in liver. |
| glucocorticoid receptor, has_component, diabetic complications | no | Diseases aren't receptor components. |
| endothelial glucocorticoid receptor (GR), has_causative_agent, renal fibrosis | no | Relation misuse; fibrosis doesn't "cause" the receptor. |
| glucocorticoid receptor, has_disposition, cortisol | no | Cortisol is a ligand, not a disposition. |
| glucocorticoid receptor alpha, has_component, coronavirus disease 2019 | no | COVID-19 isn't a component. |
| glucocorticoid receptor, has_component, estrogen | no | Hormone ≠ receptor component. |
| endothelial glucocorticoid receptor (GR), has_pathological_process, WNT signaling | no | WNT signaling isn't inherently pathological. |
| glucocorticoid receptor, has_pathological_process, osteoporosis (OP) | maybe | Excess GR signaling leads to GC-induced OP; relation loose. |
| glucocorticoid receptor, has_direct_substance, PEPCK | no | GR regulates PEPCK expression; not a substance of GR. |
| glucocorticoid receptor (GR), has_direct_substance, Kupffer cells | no | Cells aren't receptor substances. |
| glucocorticoid receptor (GR), has_component, Leydig cells | no | Tissues/cells aren't components of a receptor. |
| glucocorticoid receptor, has_direct_substance, stress response | no | Process ≠ substance. |
| glucocorticoid receptor, associated_with, miR-32-5p | maybe | Limited, context-specific miRNA linkage. |
| glucocorticoid receptor, has_direct_substance, G6P | no | Metabolite isn't a receptor substance. |
| glucocorticoid receptor, has_component, SETD1A/COMPASS complex | no | Possible cofactor interaction, not a component. |
| glucocorticoid receptor (GR), has_direct_substance, STAT6 | no | TF interactor, not a substance of GR. |
| glucocorticoid receptor (GR), has_direct_substance, adipose tissue macrophage (ATM) | no | Cells ≠ substances. |
| glucocorticoid receptor (GR), has_component, peripheral sensory neurons | no | Neurons aren't receptor components. |
| glucocorticoid receptor $\beta$ (GR$\beta$), has_direct_substance, rs948820149 | no | SNP pertains to gene, not isoform "substance." |

# F GraphRAG evaluation on Medical Benchmarks

Table F1: GraphRAG KG evaluation on public benchmarks

|  | MedMCQA | MedQA | MMLU (medical) |
|---|---|---|---|
| **# Questions** | 61 | 75 | 62 |
| LLM KG (baseline) | 72.1 | 85.3 | 71.0 |
| Seed KG | **76.5** | 81.3 | 73.1 |
| GRAPHMERT | 73.8 | **88.0** | **74.7** |

In this evaluation (Table F1), we select questions related to diabetes and its comorbidities from popular medical benchmarks and run GraphRAG evaluation on the selected questions. We first filter the benchmarks with the Qwen3-32B model, as stated in Section 5.3.2, and then manually review and remove some questions that are irrelevant to diabetes. The full list of selected questions can be downloaded from GitHub.

## G Helper LLM prompts

We include only one example for all few-shot settings. For each prompt, all examples are mined from the dataset. Then, we prompt GPT-o3 with the zero-shot prompt, edit the reply if needed, and include it as a few-shot example.

All sequences in our datasets are lower-case; here, we use normal case to improve readability.

---

### Entity Discovery Prompt

You are a medical-domain extractor building a diabetes KG of ⟨head, relation, tail⟩. You possess advanced medical academic knowledge.

Given an input sequence, identify entities specifically relevant to diabetes, its complications, comorbidities, therapeutics, and related biomedical entities that help to clarify or contextualize them. Output a Python list of up to 6-word entity "heads" following these rules:

1. Select a precise and medically-specific span (e.g., "myocardial infarction," not "infarction"). Avoid generic terms like "disease," "condition," "patients," and "comorbidity" without a specific context. When encountering vague descriptors like "complication," "symptom," or "effect," always prefer explicitly named conditions or symptoms directly linked to diabetes pathology or diabetes comorbidities.

2. Keep original spelling, casing, and abbreviations from the sequence.

3. Choose only entities that add meaningful medical knowledge to the diabetes KG. Do not include COVID-related terms. Do not include head entities that describe findings in animal models (mice, rats, etc.).

4. A few examples of low-value entities you **should not** include:
   - '≥ 10 % weight reduction' (too context-dependent).
   - 'nhanes 2015 - 2018' (dataset/survey, not a medical entity).
   - 'semaglutide 2.4 mg' (includes a dosage, which can vary).
   - '60+ women' ("60+" is too context-dependent).
   - "anxiety," "home births," "pregnant women," "neonatal deaths," "general practitioners" (not specific enough to diabetes; only include if explicitly related to diabetes).

5. If it is not clear whether a term adds diabetes-specific knowledge, look at the context. If the text explicitly links the term to a diabetes-specific concept, include it. Otherwise, exclude it when mentioned only in a generic context. Include such terms when the sequence clearly links them to a diabetes-relevant gene, pathway, cell type, or therapeutic effect.

You will be provided with incorrect output examples beginning with "Output (Incorrect)." Use them to avoid the common mistakes. Wrap your intermediate reasoning steps clearly within ⟨think⟩ ... ⟨/think⟩ tags. Be strict and discard any entity about which you are uncertain and that is not relevant to diabetes. After generating, verify your output.

**Steps:**

1. Identify candidate spans.

2. Filter by medical precision and relevance rules.

3. Confirm the entity's relevance and contribution to the diabetes KG; discard low-value entities.

**Input format:** sequence
**Output format:** ['head1', 'head2', ...]. If none, output [].

---

---

### Few-shot Example for Entity Discovery Prompt

Input:

> sequence: . . . , its upstream regulator has the opposite effect (Han et al., 2013). Previous studies suggest that CHOP deteriorates ER stress and accelerates cell death via promoting protein synthesis and oxidative stress (Han et al., 2013). In addition, ER stress damages $\beta$-cells, possibly through altering $Ca^{2+}$ homeostasis. It has been indicated that ER stress interferes with the function of RyR located in the membrane of the ER and causes leakage of ER $Ca^{2+}$ (Yamamoto et al., 2019). The destruction of $\beta$-cell ER $Ca^{2+}$ homeostasis results in impaired insulin secretion and further promotion of $\beta$-cell death . . .

Output:
```
[‘‘chop,’’ ‘‘er stress,’’ ‘‘ryr,’’ ‘‘$\beta$-cells,’’ ‘‘impaired insulin secretion’’]
```

The rationale behind the output:
Candidates: CHOP; ER stress; $\beta$-cells; RyR; impaired insulin secretion; oxidative stress; protein synthesis. Keep precise diabetes-relevant entities: CHOP (ER stress regulator), ER stress, $\beta$-cells, RyR, impaired insulin secretion.

---

### Relation Matching Prompt

You are a medical intelligence with academic knowledge in diabetes and comorbidities. We are building a diabetes knowledge graph of triples ⟨head, relation, tail⟩.

Given:

- a sequence including biomedical context,
- a list of heads,

return, for each head, all relations chosen from the list below that could form a plausible KG triple and are supported by the sequence.

Allowed relations:
**###**

| | | |
|---|---|---|
| associated_finding_of | has_associated_morphology | inverse_isa |
| associated_morphology_of | has_component | is_interpreted_by |
| associated_with | has_disposition | isa |
| causative_agent_of | has_finding_site | occurs_before |
| cause_of | has_method | part_of |
| due_to | has_modification | plays_role |
| finding_site_of | has_part | possibly_equivalent_to |
| focus_of | interprets | |

**###**

The relations are taken from UMLS and have the same meaning as in UMLS.
Examples:
-------------

- carotid artery stenosis | associated_finding_of | history of carotid artery stenosis
- fibrosis | associated_morphology_of | endomyocardial fibrosis
- cancer | associated_with | anemia in malignant neoplastic disease
- Mycobacterium tuberculosis | causative_agent_of | Tuberculosis
- diabetes mellitus | cause_of | diabetic foot
- hypoglycemic alcoholic ketoacidosis | due_to | acute alcohol intoxication
- adipose tissue | finding_site_of | lipoatrophy
- renal failure | focus_of | emergency hemofiltration

- hepatitis A | has_associated_morphology | Hepatocellular necrosis
- fasting triglyceride | has_component | triacylglycerol
- tumor necrosis factor | has_disposition | immunomodulator
- melanoma | has_finding_site | skin
- bariatric surgery | has_method | surgical action
- glucagon | has_modification | glucagon hydrochloride
- nephron | has_part | glomerulus
- overweight | interprets | body weight measure
- adiponectin | inverse_isa | high molecular weight adiponectin
- blood eosinophil counts | is_interpreted_by | asthmatic pulmonary eosinophilia
- empagliflozin | isa | sodium glucose cotransporter subtype 2 inhibitor
- cardiac amyloidosis | occurs_in | old age
- coronary syndrome | possibly_equivalent_to | preinfarction syndrome
- MI | same_as | Myocardial infarction
-------------

Note the meaning of some relations in UMLS: *isa* and *inverse_isa* are exact inverses of each other.

- *isa* — points up the hierarchy: "Diabetic retinopathy" isa "Retinal disease." (specific → general)
- *inverse_isa* — points down the hierarchy: "Retinal disease" inverse_isa "Diabetic retinopathy."
- *cause_of* — directional link where the source concept is understood to directly or indirectly produce, trigger, or give rise to the target concept.
- *due_to* — causal link: the subject condition, finding, or situation results from the object. Inverse: *cause_of*.
- *associated_with* — non-directional link indicating that two concepts are statistically or clinically linked without asserting a clear cause-and-effect direction.
- *has_associated_morphology* — links a pathological or clinical entity (typically a disease, syndrome, or injury) to the characteristic structural change ("morphology") it produces. Concretely: source = disorder concept; target = morphologic abnormality (e.g., "Necrosis," "Hyperplasia," "Fibrosis"). Inverse: *associated_morphology_of*.
- *associated_finding_of* — reads as: "X associated_finding_of Y" "Finding X is the clinical finding for which procedure Y is performed."
-------------

Input format:

```
sequence
heads: [head1, head2, ...]
```

Output format:

```
{
    ''head 1'': [''relation 1,'' ''relation 2,'' ...],
    ''head 2'': [...],
}
```

Steps:

1. Understand Input
   - Clearly understand the biomedical context from the sequence.
   - For each head, find explicit mentions in the text.
   - Check if each head is explicitly linked to other concepts or relations.
2. Use the list of allowed relations. Evaluate each head individually. Do not overuse the relation `associated_with` — apply it only when appropriate.
3. For each head, list only plausible and supported relations. Return `[]` if none apply.

Think concisely within ⟨think⟩ ... ⟨/think⟩. Immediately after, output JSON.

---

### Few-shot Example for the Relation Matching Prompt

Input:

> . . . interleukin-1 R6, and receptor activator of nuclear factor kappa-B (RANK). Together, proteomic data suggest the targeting of several key regulators of inflammation, bone, and adipose turnover, via transforming growth factor-beta/SMAD, and Wingless-related integration site/be-catenin signaling pathways. To the best of the knowledge, this is first evidence of an intervention that drives against bone loss via RANK. Metatranscriptomic analyses of the gut microbiota show P7C3 increased Porphyromonadaceae bacterium, Candidatus Melainabacteria, and Ruminococcaceae bacterium abundance, potentially contributing to the favorable inflammatory. . .

```
heads: ['interleukin-1 r6,' 'receptor activator of nuclear factor kappa-b,'
'transforming growth factor-beta']
```

Output:

```
{
    ''interleukin-1 r6'': [''associated_with''],
    ''receptor activator of nuclear factor kappa-b'': [''cause_of''],
    ''transforming growth factor-beta'': [''part_of''],
}
```

The rationale behind this output:

interleukin-1 r6 $\rightarrow$ `associated_with` $\rightarrow$ Named as a "key regulator of inflammation," which links it to the inflammatory process without stating direction or hierarchy, so the non-causal `associated_with` relation fits best.

receptor activator of nuclear factor $\kappa$-B (RANK) $\rightarrow$ `cause_of` $\rightarrow$ The text says the intervention prevents bone loss via RANK, implying that RANK signalling produces or drives bone loss; therefore `cause_of` is appropriate.

transforming growth factor-beta $\rightarrow$ `part_of` $\rightarrow$ Explicitly mentioned within the "TGF-$\beta$/Smad signalling pathway," so it is a constituent component (`part_of`) of that pathway.

No additional relations are warranted.

---

### Combining GRAPHMERT-predicted Top Tokens Prompt

You are completing triples for a medical knowledge graph on diabetes and its comorbidities. For each sample, you're given a sequence, a head entity in that sequence, a relation, and a list of candidate tokens. The relations are from UMLS and have the same meaning.

Your task is to output a filtered list of high-quality and factual tails in the format: `[''tail 1,'' ''tail 2,'' ...]` or `[]`.

**To form the list of candidates:**
Step 1: Analyze the sequence to understand the context and identify the head entity and relation.
Step 2: Choose candidate tails. You can combine tokens from the candidate list to get the most precise, relevant, and meaningful tails in the context of the head and relation. Combine subword tokens, too.
Step 3: Verify each candidate.

**Verification.** Each tail must:

- Be causally and factually related to the head via the specified relation. Make sure the relation direction is correct: the head implies the tail given the relation. Note that `isa` is a subclass→class relation, and `inverse_isa` is a class→subclass relation.

- Be supported by the sequence, but you can rely on well-established medical knowledge even if the sequence does not spell it out verbatim. If no reliable support exists, reject the tail.

- Add valuable medical knowledge to the graph. Tails must be non-redundant. When all tails are factual, prefer specific tails over general and vague (e.g., "proliferative diabetic retinopathy" over "retinopathy"). Terms that include "level," "disease," "disorder," or "complication" are too vague and rarely add useful knowledge to the KG.

- Include only tokens from the list of candidates.

Reason step by step within `<think>...</think>`.

You will see incorrect outputs labeled "Output (Incorrect)." Avoid similar errors.

**Before finalizing:**
– Ensure all output constraints are met.
– Validate that each tail is **logically, contextually, and factually aligned** with the head and relation.
– Confirm that each triple adds **meaningful** knowledge to the graph.

Note the meaning of some UMLS relations you may encounter in the input:

- *isa* and *inverse_isa* are exact inverses of each other.
  - *isa* – points up the hierarchy: "Diabetic retinopathy" *isa* "Retinal disease." (specific → general)
  - *inverse_isa* – points down the hierarchy: "Retinal disease" *inverse_isa* "Diabetic retinopathy."
- *cause_of* – directional link where the source concept directly or indirectly produces, triggers, or gives rise to the target concept.
- *due_to* – causal link: the subject condition, finding, or situation results from the object. Inverse: *cause_of*.
- *associated_with* – non-directional link indicating that two concepts are statistically or clinically linked without asserting a clear cause–effect direction.
- *has_associated_morphology* – links a pathological or clinical entity (typically a disease, syndrome, or injury) to the characteristic structural change ("morphology") it produces.
  Concretely: source = a disorder concept; target = a *Morphologic Abnormality* concept (e.g., "Necrosis," "Hyperplasia," "Fibrosis"). Inverse: *associated_morphology_of*.
- *associated_finding_of* – reads as:
  "X *associated_finding_of* Y" ⇒ "Finding X is the clinical finding for which procedure Y is performed."

---

## Few-Shot Example for Combining GRAPHMERT-predicted Top Tokens Prompt

Input:

> sequence: . . . 2+ binding to S100A1 EF-hand motifs, the conformation of S100A1 changes and promotes interactions with target proteins. RAGE consists of three domains: the cytoplasmic, transmembrane, and extracellular domains. The extracellular domain consists of C1, C2, and V domains. V domains are the primary receptors for the S100 protein. It was reported several years ago that S100A1 and RAGE V domains interact in a pathway involving S100A1-RAGE signaling, whereby S100A1 binds to the V domain, resulting in RAGE dimerization. The autophosphorylation of the cytoplasmic domain initiates a signaling cascade that regulates cell proliferation, cell growth, and tumor formation. In this study. . .

head: [''s100a1'']
relation: associated_with
predictions: protein receptor hydrolase structure process pathway factor complex glycoprotein s100a family domain oxidoreductase proteinase ligand extracellular signaling calcium apolipoprotein s100

Output:

---

[``calcium signaling pathway'']

The rationale behind this output:
Available tokens let us build terms such as:

- "calcium signaling pathway" (calcium + signaling + pathway)
- "extracellular signaling" (extracellular + signaling) – too broad
- "s100a family" – classification, not an association

Other tokens ("glycoprotein," "oxidoreductase," etc.) do not represent well-known processes linked to S100A1.

**Preference & verification:**
"S100A1 modulates intracellular $Ca^{2+}$ handling and is well-documented to participate in the calcium signaling pathway, especially in cardiac and skeletal muscle." This is a concrete biological pathway association that adds value to the graph.

---

## H  GraphRAG Prompts

---

**I1:  GraphRAG Index Prompt**

-Role-

You are an AI assistant specialized in extracting structured information from biomedical texts to build a knowledge graph about diabetes.

-Goal-

Given some medical paper abstracts, a predefined list of entity types, and a predefined list of relations, identify all entities of those types and the medically meaningful relationships explicitly described among the identified entities within the abstract. You should only extract entities that are relevant to diabetes, its complications, and comorbidites.

-Entitiy Types-

You should extract entities from the following 5 entity types:

Organism, Anatomical Structure, Manufactured Object, Substance, Conceptual Entity.

Use the subcategories listed below SOLELY as guidance to help you determine the correct main entity type. Only use the 5 main entity types in your output.

1.  Organism: Plant; Fungus; Virus; Bacterium; Archaeon; Eukaryote; Vertebrate; Amphibian; Bird; Fish; Reptile; Mammal; Human

2.  Anatomical Structure: Embryonic Structure; Anatomical Abnormality; Congenital Abnormality; Acquired Abnormality; Fully Formed Anatomical Structure; Body Part, Organ, or Organ Component; Tissue; Cell; Cell Component; Gene or Genome

3. Manufactured Object: Medical Device; Drug Delivery Device; Research Device; Clinical Drug

4.  Substance: Chemical; Pharmacologic Substance; Antibiotic; Biomedical or Dental Material; Biologically Active Substance; Hormone; Enzyme; Vitamin; Immunologic Factor; Receptor; Indicator, Reagent, or Diagnostic Aid; Organic Chemical; Nucleic Acid, Nucleoside, or Nucleotide; Amino Acid, Peptide, or Protein; Inorganic Chemical; Element, Ion, or Isotope; Body Substance; Food

5.  Conceptual Entity: Idea or Concept; Body System; Body Space or Junction; Body Location or Region; Molecular Sequence; Nucleotide Sequence; Amino Acid Sequence; Carbohydrate Sequence; Geographic Area; Finding; Laboratory or Test Result; Sign or Symptom; Organism Attribute; Clinical Attribute; Intellectual Product; Occupation or Discipline; Organization; Group

-Relation Types-

Please only identify the following 35 relations:  ['associated_finding_of,' 'associated_morphology_of,' 'associated_with,' 'causative_agent_of,' 'cause_of,' 'direct_procedure_site_of,' 'due_to,' 'finding_site_of,' 'focus_of,' 'has_associated_morphology,' 'has_causative_agent,' 'has_clinical_course,' 'has_component,' 'has_direct_procedure_site,' 'has_direct_substance,' 'has_disposition,' 'has_entire_anatomy_structure,' 'has_finding_site,' 'has_focus,' 'has_method,' 'has_modification,' 'has_part,' 'has_pathological_process,' 'interprets,' 'inverse_isa,' 'is_interpreted_by,' 'is_modification_of,' 'isa,' 'method_of,' 'occurs_before', 'occurs_in,' 'part_of,' 'plays_role,' 'possibly_equivalent_to,' 'same_as']

The following provides one example for each type of relation, formatted as 'head, relation, tail':

fetal growth restriction (fgr), associated_finding_of, history of fetal growth retardation

tumors, associated_morphology_of, neoplastic disease

neutropenia, associated_with, neutropenic sepsis

s. epidermidis, causative_agent_of, staphylococcus epidermidis ventriculitis

chronic kidney disease (ckd), cause_of, renal retinopathy

gastric fundus, direct_procedure_site_of, laparoscopic fundoplication

diabetic cardiomyopathy (dbcm), due_to, diabetes mellitus

endocrine pancreas, finding_site_of, extreme insulin resistance type a

gait abnormalities, focus_of, prosthetic gait training

pyoderma gangrenosum, has_associated_morphology, neutrophilic infiltration

chronic chagas disease cardiomyopathy, has_causative_agent, trypanosoma cruzi

membranous nephropathy, has_clinical_course, chronic

---

serum creatinine level, has_component, creatinine
fmt, has_direct_procedure_site, gastrointestinal tract structure
high-intensity statins, has_direct_substance, hmg-coa reductase inhibitor
resveratrol (res), has_disposition, platelet aggregation inhibitor
middle occipital gyrus, has_entire_anatomy_structure, entire lateral occipital gyrus
diabetes retinopathy, has_finding_site, retinal structure
on-line hemodiafiltration, has_focus, renal failure syndrome
islet cell transplant, has_method, surgical transplantation
uric acid, has_modification, calcium urate
anaerobic glycolysis, has_part, pyruvate kinase activity
evans syndrome, has_pathological_process, autoimmune process
endocrine hypertension, interprets, blood pressure
adaptive thermogenesis, inverse_isa, diet induced thermogenesis
serum triglyceride, is_interpreted_by, serum triglyceride levels
tau, is_modification_of, uridine
t cell receptors, isa, antigen receptor
amputation, method_of, cineplastic amputation
renal transplantation, occurs_before, accelerated rejection of renal transplant
paediatric obesity, occurs_in, childhood
bone resorption, part_of, bone remodeling
everolimus (eve), plays_role, antineoplastic therapeutic role
non-alcoholic fatty liver disease, possibly_equivalent_to, fatty liver
retinal cotton wool spots, same_as, retinal exudates

-Steps-
1. Identify all entities corresponding to one of the 5 main entity types and relevant to diabetes, using the subcategory examples as guidance for classification. For each identified entity, extract the following information:
- entity_name: Name of the entity, lowercase
- entity_type: One of the following types: Organism, Anatomical Structure, Manufactured Object, Substance, Conceptual Entity.
- entity_description: concise description of the entity's attributes and activities. Format each entity as
("entity"<|><entity_name><|><entity_type><|><entity_description>)

2. From the entities identified in step 1, identify all pairs of (source_entity, target_entity) that are clearly related to each other according to the given text, and are medically meaningful. Only use the 35 relationships that are in the predefined list.

Avoid relationships that are attached to entities that are too general, for example: patients, bodily functions, parameters, management, optimization. Only keep the relationships that state facts, represent the main idea in the text, or other important relationships that are in the predefined list. It's acceptable if some entities identified in the previous step are not used.

For each pair of related entities, extract the following information: - source_entity: name of the source entity, as identified in step 1
- target_entity: name of the target entity, as identified in step 1
- relationship: one relation that is in the predefined list, according to the given text
- relationship_strength: a numeric score out of 10 indicating the strength of the relationship between the source entity and target entity
Format each relationship as
("relationship"<|><source_entity><|><target_entity><|><relationship_description><|>
<relationship_strength>)

3. Return output in English as a single list of all the entities and relationships identified in steps 1 and 2. Use ## as the list delimiter.

4. When finished, output <|COMPLETE|>

- Constraints and Guidelines
- Strict Textual Grounding: Base all extractions only on the provided medical abstract. Do not use external knowledge or make assumptions beyond what is written.
- Entity Filtering: Only extract the entities whose type is present in the provided 5 Entity Type, and only extract entities that are relevant to diabetes, its complications, and comorbidites.
- Relationship Filtering: Extract only the 35 relationships as defined. Exclude all other relationships.
- Delimiter Usage: Strictly adhere to the specified tuple_delimiter within tuples and record_delimiter between records.

## I2: GraphRAG Index Example

Input:

```
#######################
Entity_types: Organism, Anatomical Structure, Manufactured Object, Substance, Conceptual Entity
Text:
CMR imaging enables comprehensive detection of myocardial remodeling in patients undergoing TAVR.
Regression of LV matrix volume as a surrogate for reversible diffuse myocardial fibrosis is accompanied by
increase of myocardial function and recovery from heart failure. Further data are required to define the
value of these parameters as therapeutic targets for optimized management of TAVR patients.
#######################
Output:
```

Output:
```
("entity"<|>cmr imaging<|>Conceptual Entity<|>A medical imaging technology that allows for the
comprehensive detection of myocardial remodeling.)
##
("entity"<|>myocardial remodeling<|>Conceptual Entity<|>A pathological process involving changes
in the heart muscle, observed in patients undergoing TAVR.)
##
("entity"<|>tavr<|>Conceptual Entity<|>Transcatheter Aortic Valve Replacement, a medical
procedure undergone by patients with myocardial remodeling.)
##
("entity"<|>regression of LV matrix volume<|>Conceptual Entity<|>The reduction in the volume of
the left ventricular matrix.)
##
("entity"<|>reversible diffuse myocardial fibrosis<|>Conceptual Entity<|>A condition of the
heart muscle for which the regression of LV matrix volume acts as a surrogate.)
##
("entity"<|>myocardial function<|>Conceptual Entity<|>The physiological performance of the heart
muscle.)
##
("entity"<|>heart failure<|>Conceptual Entity<|>A clinical condition from which recovery is
observed along with the regression of LV matrix volume.)
##

("relationship"<|>myocardial remodeling<|>cmr imaging<|>has_method<|>9)
##
("relationship"<|>regression of LV matrix volume<|>reversible diffuse myocardial
fibrosis<|>is_interpreted_by<|>9)
##
("relationship"<|>regression of LV matrix volume<|>myocardial function<|>associated_with <|>10)
##
("relationship"<|>regression of LV matrix volume<|>heart failure<|>associated_with<|>10)
##
("relationship"<|>tavr<|>myocardial remodeling<|>occurs_before<|>8)
```

```
<|COMPLETE|>
```

## I3: GraphRAG Query Prompt

—**Role**—
You are a helpful assistant responding to questions about data in the tables provided.

—**Data tables**—
```
{context_data}
```

—**Goal**—
Generate a response in the specified target format that answers the user's question using only the information provided in the tables. Do not use any external knowledge. If any prior knowledge conflicts with the tables, always rely on the tables as the sole source of truth.
If you cannot get the answer based on the tables, just say so. Do not make anything up.
Points supported by data should list their data references as follows:

> "This is an example sentence supported by multiple data references `[Data: <dataset name> (record ids); <dataset name> (record ids)]`."

Do not list more than 5 record ids in a single reference. Instead, list the top 5 most relevant record ids and add "+more" to indicate that there are more.

For example:

> "Person X is the owner of Company Y and subject to many allegations of wrongdoing `[Data: Entities (5, 7); Relationships (2, 7, 34, 46, 64, +more)]`."

where 5, 7, 2, 34, 46, and 64 represent the id (not the index) of the relevant data record.
Do not include information where the supporting evidence for it is not provided.

—**Target response format**—
Response type: multiple paragraphs.
Provide a concise answer using `\boxed{}`, select only the correct letter from A, B, C, D. (e.g., `\boxed{C}`)

Reference data points that support your answer using the given format (e.g., `[Data: Relationships (2, 3, 4); Entities (35, 36, 37, 39, 55, +more)]`). If no relevant information from the table supports your answer, leave the reference empty (e.g., `[]`).

## I4: GraphRAG Context Example

—**Relationships**—
```
source - (relation) -> target
chemotherapy - (cause_of) -> peripheral neuropathy due to and following chemotherapy
chemotherapy - induced peripheral neuropathy - (due_to) -> administration of antineoplastic
agent
antineoplastic drugs - (associated_with) -> resistance to antineoplastic drug
chemotherapy - induced peripheral neuropathies - (due_to) -> administration of antineoplastic
agent
hematologic malignancies - (isa) -> neoplastic disease
lymphoid leukaemia - (associated_finding_of) -> history of lymphoid leukemia
leukemia - (associated_finding_of) -> history of leukemia
cancer - (focus_of) -> oral chemotherapy for malignant neoplasm
cancer - (associated_with) -> restrictive cardiomyopathy secondary to malignancy
cancer - (associated_with) -> cancer - related fatigue
imatinib - (plays_role) -> antineoplastic therapeutic role
imatinib - (isa) -> antineoplastic agent
vorinostat - (plays_role) -> antineoplastic therapeutic role
nivolumab - (plays_role) -> antineoplastic therapeutic role
rucaparib - (plays_role) -> antineoplastic therapeutic role
```

```
nivolumab - (isa) -> antineoplastic agent
rucaparib - (isa) -> antineoplastic agent
antineoplastic agents - (inverse_isa) -> vorinostat
......
```

Table I1: FActScore* KG evaluation with Qwen3-32B as a judge: helper LLM ablation. **Bold**: better than Qwen3-32B baseline; **underlined bold**: best in column.

| KG type | Model size | #triples | Context only | Context + General truth |
|---|---|---|---|---|
| Qwen3-32B (baseline) | 32B | 515,460 | 40.2 | 48.1 |
| GRAPHMERT with Qwen3-32B as helper LLM | 80M | 138,201 | **69.8** | **72.2** |
| GRAPHMERT with Qwen3-4B as helper LLM | 80M | 9,000 | **61.5** | **62.0** |

Table I2: ValidityScore evaluation with Qwen3-32B as a judge: helper LLM ablation. **Bold**: better than Qwen3-32B baseline; **underlined bold**: best in column.

| KG type | Model size | #triples | yes (Validity-Score) | maybe | no | missing |
|---|---|---|---|---|---|---|
| Qwen3-32B (baseline) | 32B | 515,460 | 43.0 | 24.1 | 31.4 | 1.5 |
| GRAPHMERT with Qwen3-32B as helper LLM | 80M | 138,201 | **68.7** | 18.3 | 10.8 | 2.2 |
| GRAPHMERT with Qwen3-4B as helper LLM | 80M | 9,000 | **63.8** | 18.5 | 15.1 | 2.7 |

## I Helper LLM ablation

In this section, we investigate how much our gains in FActScore and ValidityScore come from GraphMERT versus a helper LLM that converts its token predictions into text tails. We provide an evaluation of 9,000 triples combined with a smaller Qwen3-4B LLM.

Tables I1 and I2 compare GRAPHMERT KG using Qwen3-32B as a helper LLM vs. GRAPHMERT KG using a smaller Qwen3-4B helper. The result shows that with a small helper, GRAPHMERT still clearly outperforms the baseline, indicating that a substantial part of the improvement comes from the own predictions of GRAPHMERT rather than from the capacity of the helper. At the same time, scaling the helper LLM up reduces nonsensical completions and helps form more nuanced, semantically precise tails, and yields further gains.

Although the small Qwen3-4B helper LLM is prompted to merge token-level predictions into fluent English tails, in practice, it sometimes produces ungrammatical or clause-like spans (e.g., "hybrid closed loop technology, has_part, hybrid closed insulin delivery loop pump systems," "ldl-c / apo b ratio associated_with high"), which we do not want to accept as tails. To filter out such malformed completions while preserving genuine biomedical phrases, we use a small set of syntactic heuristics. First, we normalize the tail span and use a domain lexicon (keywords and medical suffixes, e.g., "-emia," "-itis," "-and kinase") to whitelist clearly biomedical strings, guaranteeing they will not be misclassified as non-nouns or filtered out. We then reject a tail if it is very long ($> 40$ characters) or dominated by stopwords (stopword ratio $> 0.3$ for spans with $\geq 4$ tokens). For relations other than a small set of "adjacency/role" relations (e.g., `plays_role`, `has_associated_morphology`), we also require a nominal structure: Multi-token tails must contain at least one noun/proper noun and have a nominal root, and spans containing a finite verb are rejected as clause-like rather than entity-like.

Further, to let readers directly inspect the GRAPHMERT prediction quality, we also include raw GRAPHMERT predictions for four sequences prior to filtering by a helper LLM. Examples A-D display the full input text chunk and the raw top-20 GRAPHMERT token predictions. Tokens like `##oscler` are subword token outputs. We include outputs from standard GRAPHMERT trained with MNM and GRAPHMERT trained with a simple MLM objective. The former is used throughout the article as it creates a basis for nuanced multi-token tails, increasing coverage based on fine-grained tails. This is why the output of GRAPHMERT trained with the MNM objective includes many subword tokens. GRAPHMERT trained with the MLM objective leans on trivial words, which may boost accuracy on some test tasks, but neglect fine-grained details.

## Raw GraphMERT Output: Example A

**Input Context Sequence:**
. . . To examine direct and indirect costs, early retirement, cardiovascular events and mortality over 5 years in people with atherosclerotic cardiovascular disease (ASCVD) and matched controls in Sweden. Methods Individuals aged ≥~16 years living in Sweden on 01 January 2012 were identified in an existing database. Individuals with ASCVD were propensity score matched to controls without ASCVD by age, sex and educational status. We compared direct healthcare costs (inpatient, outpatient and drug costs), indirect costs (resulting from work absence) and the risk of stroke, myocardial infarction (MI) and early retirement. Results After matching, there were 231,417 individuals in each cohort. Total . . .

**Query 1:** (atherosclerotic cardiovascular disease, cause_of, ?)

GRAPHMERT with MNM objective:

[disease, cardiovascular, atherosclerotic, with, vascular, atherosclerosis, ##oscler, due, ##vd, cerebrovascular, arteri, chronic, ischemic, risk, coronary, of, disorder, dementia, hypercholesterolemia, ##osclerosis]

GRAPHMERT with MLM objective:

[atherosclerosis, ischemic, disease, cardiovascular, atherosclerotic, coronary, artery, ather, stroke, vascular, infarction, carotid, cerebrovascular, ischaemic, plaque, angina, arteri, chronic, arterial, vessel]

**Query 2:** (myocardial infarction, due_to, ?)

GRAPHMERT with MNM objective:

[ischemia, due, ischemic, injury, ischaemia, circulatory, revascularization, caused, myocardial, infarction, cerebrovascular, to, disease, heart, vascular, cardiovascular, complication, of, bypass, stroke]

GRAPHMERT with MLM objective:

[accident, poisoning, ischemia, injury, diabetes, due, cardiovascular, infarction, ischaemia, hypertension, care, emergency, burn, cardiac, accidents, revascularization, medical, finding, myocardial, accidental]

**Query 3:** (stroke, due_to, ?)

GRAPHMERT with MNM objective:

[ischemia, artery, cerebrovascular, vascular, ischemic, stroke, disease, revascularization, fracture, ischaemia, of, bleeding, injury, due, neuropathy, complications, venous, cardiovascular, complication, to]

GRAPHMERT with MLM objective:

[ischemia, accident, ischemic, ischaemia, ischaemic, diabetes, due, cerebrovascular, emergency, coronary, -, of, poisoning, adverse, myocardial, care, hypoglycemia, angina, acute, heart]

---

**Raw GraphMERT Output: Example B**

**Input Context Sequence:**
...widespread, fatal, infectious disease. Several antivirals against rabies virus (RABV) infection have been reported, but no approved, RABV-specific antiviral drugs that inhibit RABV infection in the clinic after symptom onset are available. Therefore, more effective drugs to reduce rabies fatalities are urgently needed. Bardoxolone methyl (CDDO-Me), an FDA-approved compound that has long been known as an antioxidant inflammatory modulator and one of the most potent nuclear factor erythroid-derived 2-likerythroid-derived 2-likects myelin, axons, and CNS neurons by Nrf2 activation. Therefore, we investigated the potency of its ...

---

**Query 1:** (bardoxolone methyl, has_disposition, ?)

GRAPHMERT with MNM objective:

[inhibitor, oxidoreductase, -, agonist, receptor, ##olone, reductase, ##ane, ##ulator, proliferator, aryl, histone, alkaloid, methyltransferase, peroxisome, hydrolase, ##ine, acid, protease, antine]

GRAPHMERT with MLM objective:

[globulin, inhibitor, ##olone, steroid, compound, protein, ligase, agonist, polysaccharide, flavonoid, tetracycline, alkylating, glucoside, hydrolase, peptide, reductase, lyase, protease, antibiotic, isothiocyanate]

**Query 2:** (bardoxolone methyl, plays_role, ?)

GRAPHMERT with MNM objective:

[therapeutic, ##oplastic, antine, antibacterial, antimalarial, antifungal, ##olone, ##ora, medicinal, role, anth, ##ano, anti, antipar, oral, pharmacological, antimicrobial, ##oside, drug, ##azine]

GRAPHMERT with MLM objective:

[antimalarial, antiviral, antibacterial, quercetin, therapeutic, antine, drug, antibiotic, compound, chloroquine, doxycycline, azithromycin, tetracycline, role, antioxidant, protease, medicine, flavonoid, antifungal, penicillin]

## Raw GraphMERT Output: Example C

**Input Context Sequence:**
. . . CVS-11 and CTN) in N2a cells. We also examined whether CDDO-Me activates the Nrf2-associated pathway upon infection with RABV strains of differing virulence. Nrf2, phosphorylated sequestosome (SQSTM1), SQSTM1, hemoglobin oxygenase (HO-1) and NAD(P)H dehydrogenase quinone 1 (NQO1) expression in N2a cells increased to varying degrees with CDDO-Me treatment, accompanied by Kelch-like ECH-associated protein 1 (Keap1) dissociation, upon infection with SC16, CVS-11 or CTN. The activation of SQSTM1 phosphorylation was significantly . . .

**Query 1:** (nrf2, part_of, ?)

GRAPHMERT with MNM objective:

```
[-, response, cellular, pathway, /, signaling, defense, process, stress, keap1, to, intracellular,
protein, nrf2, of, activation, degradation, oxidative, related, inflammatory]
```

GRAPHMERT with MLM objective:

```
[nrf2, oxidoreductase, protein, antioxidant, signaling, nuclear, kinase, pathway, transcription, response,
chaperone, signal, isomerase, cytoprotective, keap1, activator, stress, apoptosis, autophagy, factor]
```

**Query 2:** (sqstm1, part_of, ?)

GRAPHMERT with MNM objective:

```
[degradation, process, of, response, pathway, complex, -, oxidoreductase, biosynthetic, reticulum,
catabolic, seques, autophagy, assembly, protein, sq, acid, synthesis, phase, ##osomal]
```

GRAPHMERT with MLM objective:

```
[protein, autophagy, kinase, mitochondrial, transcription, p62, oxidoreductase, degradation,
phosphorylated, mitophagy, ubiquitin, ribosomal, reticulum, nuclear, dehydrogenase, response, seques,
gene, transcript, lysosomal]
```

**Query 3:** (cddo - me, cause_of, ?)

GRAPHMERT with MNM objective:

```
[##do, -, to, ##o, ##in, 2, o, ##on, ##oside, epi, cd, of, toxic, ##e, protein, disease, :, product, ##y,
selen]
```

GRAPHMERT with MLM objective:

```
[fk, heme, protein, stress, oxidoreductase, endoplasmic, autophagy, zinc, chaperone, serine, pigment,
keap1, cysteine, compound, er, apoptosis, mitochondrial, reactive, oxidative, neutrophil]
```

## Raw GraphMERT Output: Example D

**Input Context Sequence:**
... interleukin-1 R6, and receptor activator of nuclear factor kappa-B (RANK). Together, proteomic data suggest the targeting of several key regulators of inflammation, bone, and adipose turnover, via transforming growth factor-beta/SMAD, and Wingless-related integration site/be-catenin signaling pathways. To the best of the knowledge, this is first evidence of an intervention that drives against bone loss via RANK. Metatranscriptomic analyses of the gut microbiota show P7C3 increased Porphyromonadaceae bacterium, Candidatus Melainabacteria, and Ruminococcaceae bacterium abundance, potentially contributing to the favorable inflammatory ...

**Query 1:** (interleukin - 1 r6, plays_role, ?)

GRAPHMERT with MNM objective:
```
[, like, and, -, chemoattractant, of, (, /, necrosis, receptor, 1, cytokine, cytokines, chemotactic, by,
on, including, inducible, in, )]
```

GRAPHMERT with MLM objective:
```
[interleukin, cytokine, response, chemokine, „ receptor, inflammatory, metalloproteinase, cytokines,
matrix, signaling, pathway, protein, process, factor, mediator, immunomod, signalling, system, secretion]
```

**Query 2:** (receptor activator of nuclear factor kappa - b, cause_of, ?)

GRAPHMERT with MNM objective:
```
[nuclear, activator, and, the, tnf, -, /, transient, a, activation, p38, activating, phosphorylated, 2,
factor, an, adiponectin, of, „ receptor]
```

GRAPHMERT with MLM objective:
```
[receptor, ligand, transcription, protein, nuclear, interleukin, „ signaling, cytokine, inhibitor,
activator, chemokine, activity, activated, transducer, kinase, ##ulator, pathway, binding, activation]
```

**Query 3:** (transforming growth factor - beta / smad, part_of, ?)

GRAPHMERT with MNM objective:
```
[growth, differentiation, factor, fibroblast, process, signaling, signalling, proliferation, /, epidermal,
##ogenesis, wnt, morphogenetic, cell, extracellular, matrix, tgf, resorption, development, regeneration]
```

GRAPHMERT with MLM objective:
```
[receptor, signaling, differentiation, protein, hormone, factor, matrix, cell, gene, transcription,
growth, collagen, cytokine, response, adipogenic, fibroblast, secretory, signalling, wnt, pathway]
```

**Query 4:** (wingless - related integration site / be - catenin, part_of, ?)

GRAPHMERT with MNM objective:
```
[wing, wnt, morphogenetic, morphogenesis, differentiation, regeneration, process, /, cell, signaling,
development, matrix, signalling, of, phosphatidylinositol, growth, ##omyosin, kinase, pathway, nuclear]
```

GRAPHMERT with MLM objective:
```
[wnt, signaling, transcription, receptor, hedgehog, differentiation, transition, protein, factor,
homeobox, enhancer, pathway, notch, kinase, cell, adipogenic, matrix, signalling, hippo, gene]
```

