# OpenReview forum: "GraphMERT: Efficient and Scalable Distillation of Reliable Knowledge Graphs from Unstructured Data"
_TMLR — Accepted by TMLR_

### Review · Reviewer_FEBJ · 2025-11-18

**Summary Of Contributions:**

This paper proposes GraphMERT, an encoder-only transformer augmented with hierarchical graph attention and a combined MLM + MNM objective, designed to extract reliable domain-specific knowledge graphs (KGs) from small high-quality text corpora. The authors argue that existing LLM-based KG generation methods are unreliable due to hallucinations, prompt sensitivity, and poor domain alignment.

Strength:

1.Clear motivation and problem framing: The authors thoroughly articulate the shortcomings of current LLM-based KG generation—hallucinations, prompt brittleness, absence of provenance, and domain-misaligned relations. The motivational example using UMLS triples effectively demonstrates the brittle behavior of frontier LLMs.

2.  Novel graph-based encoder architecture: GraphMERT introduces a meaningful architectural innovation:
        1.  a leafy chain graph representation that merges syntactic (roots) and semantic (leaves) information;
        2.   H-GAT fusion that injects relation-aware semantic representations;
        3.  a distance-sensitive attention decay mask, integrating structural information.

Weakness:

1. FActScore* and ValidityScore rely on Qwen3-32B as the judge, even though Qwen3-32B also creates the baseline KG. This setup introduces risks: The judge may be more forgiving or harsher toward its own patterns.

2. Limited discussion of computational cost: Although the model is compact, the pipeline includes heavy KG injection, H-GAT processing, span masking, entity linking with ANN search. There is no discussion present on the computational cost in the paper.

3. The seed KG heavily defines relation vocabulary and tail semantics. This is briefly mentioned in limitations but deserves more rigorous treatment. However there should be a discussion what happens the seed KG is noisy or incomplete.

**Audience:**

Yes

**Audience Explanation:**

I think the research problem addressed in this paper is interesting and of interest to TMLR.

**Claims And Evidence:**

Yes

**Claims Explanation:**

The major claims of the submission—regarding improved factuality, validity, provenance, and reliability are well-supported by clear, quantitative evidence and reinforced with diagnostic analysis and ablations

**Requested Changes:**

Requested Changes
1. The authors should provide clarity on training/inference time, GPU hours, and memory requirements.
2. The paper provides detailed LLM failure examples but does not present. It would also be interesting to include failure modes of GraphMERT.


Optional Changes:
If possible expand the evaluation beyond the medical domain to show the generalisability of the approach.

---

> ### Author Response · Authors · 2025-12-19
> **Response 1/n**
>
> We sincerely thank the reviewer for the positive and detailed feedback, as well as for recognizing the key strengths of our work, particularly well-defined problem framing and novel graph-based encoder architecture. We greatly appreciate the remarks and the time invested in evaluating our submission.
>
> W1
> > The judge may be more forgiving or harsher toward its own patterns.
>
> We additionally ran the FActScore & ValidityScore evaluation of the main results with proprietary Gemini 2.0 Flash and added the following results to the paper.
>
>
> Table 5: FActScore* KG evaluation (in percentage points).
> **Bold**: better than Qwen3-32B baseline; ⭐: best in column. Percentages are computed per KG type with repsect to its own #triples.
>
> | KG type              | Model size | #triples | Qwen3-32B as a judge: Context only | Qwen3-32B as a judge: Context + General truth | Gemini 2.0 Flash as a judge: Context only | Gemini 2.0 Flash as a judge: Context + General truth |
> |---|---:|---:|---:|---:|---:|---:|
> | Qwen3-32B (baseline) | 32B        | 515,460  | 40.2 | 48.1 | 49.1 | 44.3 |
> | GraphMERT (ours) | 80M        | 138,201  | **69.8** | **72.2** | **79.7** ⭐ | **84.0** ⭐ |
> | Grok 4 Fast      | 1.7T       | 100,000  | **79.4** ⭐ | **79.7** ⭐ | **73.3** | **67.9** |
> | Qwen3-14B        | 14B        | 100,000  | **46.4** | 45.7 | **62.4** | **57.6** |
>
>
> Table 7: ValidityScore evaluation (in percentage points). **Bold**: better than Qwen3-32B baseline; ⭐: best in column. Percentages are computed per KG type with respect to its own #triples.
>
> | KG type              | Model size | #triples | Qwen3-32B as a judge: yes (Validity-Score) | Qwen3-32B as a judge: maybe | Qwen3-32B as a judge: no | Qwen3-32B as a judge: missing | Gemini 2.0 Flash as a judge: yes (Validity-Score) | Gemini 2.0 Flash as a judge: maybe | Gemini 2.0 Flash as a judge: no | Gemini 2.0 Flash as a judge: missing |
> |---|---:|---:|---:|---:|---:|---:|---:|---:|---:|---:|
> | Qwen3-32B (baseline) | 32B        | 515,460  | 43.0 | 24.1 | 31.4 | 1.5 | 66.0 | 33.6 | 0.3 | < 0.1 |
> | UMLS Seed KG         | N/A        | 28,533   | **53.4** | 10.0 | 34.7 | 1.9 | **82.5** ⭐ | 16.5 | 1.1 | < 0.1 |
> | GraphMERT (ours) | 80M        | 138,201  | **68.7** ⭐ | 18.3 | 10.8 | 2.2 | **81.3** | 18.5 | 0.2 | < 0.1 |
> | Grok 4 Fast          | 1.7T       | 100,000  | **56.9** | 21.9 | 21.2 | < 0.1 | **68.8** | 30.9 | 0.3 | < 0.1 |
> | Qwen3-14B            | 14B        | 100,000  | 31.8 | 21.9 | 46.2 | 0.1 | 53.4 | 45.7 | 0.9 | < 0.1 |

---

> ### Author Response · Authors · 2025-12-19
> **2/n**
>
> W2
> >Limited discussion of computational cost: Although the model is compact, the pipeline includes heavy KG injection, H-GAT processing, span masking, entity linking with ANN search. There is no discussion present on the computational cost in the paper.
>
> Please see the response to C1 below.
>
> W3
>
> > There should be a discussion of what happens if the seed KG is noisy or incomplete
>
> We acknowledge the reviewer’s observation concerning the robustness of GraphMERT to noise and incompleteness in the seed KG. Ablations in Sections 6.4.1 (Training GraphMERT with Less-relevant Injections) and 6.4.2 (Training GraphMERT with a Smaller Seed KG) provide insights into this question. Specifically, Table 11 presents GraphRAG performance across different alpha values, where alpha regulates injection relevance. Lower alpha values can be interpreted as simulating noisier KGs (less relevant injections).
>
> Section 6.4.4 (Retraining with the LLM KG replacing the seed KG) further discusses this scenario. The LLM-generated KG is larger but noisier than UMLS, and we observe a 10.1% drop in ValidityScore, consistent with reduced ontology alignment and lower example quality.
>
> Regarding incomplete KGs, the ablation in Section 6.4.2 demonstrates that training with a smaller (sparser) seed KG results in only a small performance degradation, indicating that GraphMERT remains relatively robust to this scenario.

---

> ### Author Response · Authors · 2025-12-19
> **3/n**
>
> C1
> > The authors should provide clarity on training/inference time, GPU hours, and memory requirements.
>
>
> We appreciate the reviewer’s suggestion to clarify the computational aspects of our method. We have added detailed information on the training and inference setup to clarify the computational cost of our approach.
>
> GraphMERT Training.
>
> We already had the following information about the training GraphMERT architecture:
> “Training runs for 25 epochs on four H100 GPUs with BF16 precision, totaling 90 GPU hours. We use an instantaneous batch size of 32 per GPU, achieving an effective batch size of 128 through gradient accumulation (steps = 2).”
>
> This includes H-GAT processing as an integral part of GraphMERT and span masking cost during training. We have clarified that H100 GPUs have 80GB of memory each.
> KG injection includes similarity matching with triples of an external KG and the injection algorithm itself.
>
> GraphMERT prediction.
>
> The model was run in bf16. In our experiment, it takes around 2.5h on 1 H100 GPU (80 GB RAM) to run MNM prediction over the whole dataset with a batch size of 128, with GPU utilization close to 90–100%.  For example, 1,407,076 masked node predictions took 2h 34min.
>
> Injection algorithm.
>
> It uses 51 GB of peak RAM during the preprocessing stage and runs for approximately 0.5 hours on a single CPU. The pipeline is executed in a Jupyter notebook.
>
> Most of the computational cost arises from converting raw dataset records into Pandas tables, decoding text and JSON, expanding entries with multiple head matches, and finally sorting all items by score. This step could be further optimized, for example, by replacing Pandas with direct Arrow processing or by parallelizing operations for larger datasets. However, since this time is negligible compared to that of other pipeline stages, we retained Pandas and Jupyter notebook for their convenience and transparency in inspecting intermediate results.
>
> Entity linking with ANN search.
>
> We have added the computational cost for ANN search in Section 5.1.1. It takes 133 CPU minutes and can be parallelized.

---

> ### Author Response · Authors · 2025-12-19
> **4/n**
>
> C2
> > The paper provides detailed LLM failure examples but does not present. It would also be interesting to include failure modes of GraphMERT.
>
> We agree that discussing GraphMERT’s own failure modes is important. Table 8 in Section 6.2.2 already includes examples of malformed or low-quality triples extracted by GraphMERT, grouped by error type (e.g., incomplete tails, overstated causality, vague tails, and predicate misuse). These examples illustrate the main ways in which GraphMERT can fail.
>
> Specifically, as described in the text following Table 8, tail incompleteness often arises when the helper LLM accepts partial tokens, while vagueness and predicate misuse occur when GraphMERT does not rank key tail tokens highly enough for accurate completion. We also discuss these effects in comparison with LLM-generated KGs on triple sets of comparable size for the same keywords (Appendix E, Tables E2–E6), showing that GraphMERT’s errors are generally limited to semantic underspecification rather than ontology violations or relation reversals typical of LLM KGs.
>
> We clarify this in the revised version by explicitly referring to Table 8 as the analysis of GraphMERT failure cases, and by adding a sentence in the Discussion section noting incomplete predictions. We also already stated the following in the Discussion section:
> “A further limitation is its dependence on a helper LLM for tail combination, which introduces occasional incompleteness in the extracted triple tails.”

---

> ### Author Response · Authors · 2025-12-19
> **5/n**
>
> Optional Changes
> > If possible expand the evaluation beyond the medical domain to show the generalisability of the approach.
>
> We appreciate this suggestion. Extending the evaluation to additional domains would indeed strengthen the generalizability analysis; however, it would require significant effort to find a new seed KG and construct a domain-specific dataset. We therefore leave this expansion to future work.

---

### Review · Reviewer_94oo · 2025-12-04

**Summary Of Contributions:**

The paper introduces GraphMERT, a relatively small (80M parameters) encoder-only Transformer model designed to distill reliable, domain-specific Knowledge Graphs (KGs) from unstructured text. Unlike current trends that rely on large Generative LLMs for KG construction, GraphMERT operates by jointly encoding syntactic information (text) and semantic information (KG triples) into a unified data structure called "Leafy Chain Graphs." Key contributions are below:

1.	Leafy Chain Graph Encoding: A novel input representation that grafts semantic relation edges and nodes (leaves) onto syntactic token sequences (roots), allowing the model to learn joint representations.

2.	Architecture: An integration of a RoBERTa-style encoder with Hierarchical Graph Attention Networks (H-GAT) and a geodesic distance-based attention mask to handle the structural inputs.

3.	Efficiency and Reliability: The authors demonstrate that this compact model, when trained with a high-quality seed KG, outperforms significantly larger LLMs (e.g., Qwen-32B) in terms of ontology compliance (ValidityScore) and factuality (FActScore) within the medical domain (specifically Diabetes), while reducing hallucinations common in generative approaches.

**Audience:**

Yes

**Audience Explanation:**

The paper sits at the intersection of Neurosymbolic AI, Knowledge Graph construction, and Efficient NLP. Given the current saturation of LLM-based research, TMLR readers will likely appreciate a method that advocates for smaller, structured models for high-stakes domains (like healthcare) where interpretability and strict ontology adherence are more critical than open-ended generation.

**Broader Impact Concerns:**

The authors have briefly touched upon reliability in medicine. However, a specific concern arises from the "Helper LLM" component. If the Helper LLM is a black-box commercial model or a large open model, it reintroduces the very "hallucination" and "lack of provenance" risks the paper argues against, just at a different stage of the pipeline (token combination rather than triple generation).
I suggest adding a sentence in the Limitations or Broader Impact section acknowledging that while GraphMERT reduces hallucinations, the final assembly step still relies on a probabilistic model (the Helper LLM), which technically breaks the "guaranteed" chain of provenance, even if the risk is minimized. No specific Broader Impact Statement is missing, but this nuance is ethically relevant for medical AI.

**Claims And Evidence:**

Yes

**Claims Explanation:**

The submission provides a strong empirical evaluation to support its primary claim: that a specialized, smaller model can outperform general-purpose LLMs in extracting strictly ontology-compliant KGs.

•	Metrics: The use of FActScore and ValidityScore is appropriate for measuring the two key failure modes of LLMs: hallucinations and ontology violations.

•	Comparison: The comparison against Qwen3-32B (a strong open-weights model) is fair and highlights the efficiency-performance trade-off. The results showing GraphMERT achieving ~69% FActScore vs. LLM's ~40% are compelling evidence of the method's effectiveness in constraining output to valid facts.

•	Ablation Studies: Section 6.4 effectively isolates the contributions of the H-GAT module, span masking, and dropout, justifying the architectural choices.

However, there is a nuance regarding the "Helper LLM" used in the inference pipeline (discussed in Requested Changes), which complicates the attribution of success solely to GraphMERT, though the evidence broadly supports the utility of the proposed framework.

**Requested Changes:**

I find this work to be a solid contribution to the field of KG extraction. The methodology is distinct from the dominant "prompt-the-LLM" paradigm. However, to fully secure a recommendation for acceptance and clarify the technical contribution, I request the following adjustments:

1. In Section 4.4 and 5.1.3, the pipeline uses a "Helper LLM" (Qwen3-32B) to combine GraphMERT-predicted tokens into coherent tails. To what extent is the FActScore/ValidityScore attributable to Qwen3's filtering capabilities rather than GraphMERT's predictions? If GraphMERT predicts a bag of tokens and Qwen3 reconstructs the meaning, Qwen3 is doing significant semantic heavy lifting. So please include an analysis or a baseline where a simpler, non-LLM method (e.g., a simple heuristic or a smaller language model) is used to combine tokens. Alternatively, provide qualitative examples of "raw" GraphMERT token outputs before the Helper LLM processes them, so readers can judge the raw signal quality.

2. The paper relies heavily on "LLM-as-a-judge" (Qwen3-32B and GPT-5) for evaluating FActScore and ValidityScore. While this is standard practice, the paper targets the medical domain under the banner of "Reliability." LLMs are known to have biases as judges. Please conduct a small-scale human evaluation (e.g., on 50-100 randomly sampled triples) to calibrate the automated metrics. Does human judgment align with the GPT-5/Qwen verdicts? This is crucial for a paper claiming "reliable" medical AI.

3. The primary baseline is an LLM (Qwen) prompted to generate KGs. It would strengthen the paper to verify how GraphMERT compares to a traditional, fine-tuned BERT-based Relation Extraction (RE) model or a standard sequence-to-sequence extraction model (like T5-base fine-tuned on the triples). GraphMERT is essentially a specialized fine-tuning method. Readers might wonder if a vanilla fine-tuned RoBERTa or T5 (trained on the same text-triple pairs) would achieve similar results without the complex "Leafy Chain Graph" structure. Showing that GraphMERT outperforms a standard fine-tuned small model would solidly validate the architectural contribution (H-GAT + Unified Graphs).

4. Section 4.1 and Figure 3 introduce the data structure of "Leafy Chain Graph". The description is technically precise but dense. Please add a brief intuition on why this structure helps. Is the main benefit that it forces the model to attend to the "semantic" leaves using the same mechanism it uses for "syntactic" text? A sentence or two clarifying the intuition behind the "joint representation" would help.

5. The training data is ~125M tokens (Table 1). This is quite small by modern standards. Please explicitly discuss in the Limitations section whether this approach scales well to larger corpora or if it is specifically designed for "low-resource, high-quality" scenarios.

---

> ### Author Response · Authors · 2025-12-19
> **1/n**
>
> Thank you for the thoughtful review and for the concrete suggestions on how to strengthen our results and improve our paper for readers. Below, we describe the corresponding changes we made to the paper.
>
> C1
> > To what extent is the FActScore/ValidityScore attributable to Qwen3's filtering capabilities rather than GraphMERT's predictions?
> Include an analysis or a baseline where a simpler, non-LLM method (e.g., a simple heuristic or a smaller language model) is used to combine tokens.
>  Alternatively, provide qualitative examples of "raw" GraphMERT token outputs before the Helper LLM processes them, so readers can judge the raw signal quality.
>
> We evaluated 9,000 triples combined with a smaller Qwen3-4B LLM. In addition, we obtained raw GraphMERT output examples. We have added both to Appendix I “Helper LLM ablation”.
>
> When a smaller model, Qwen3-4B, is used as a helper LLM, it makes some trivial English mistakes when combining tokens. To fix that, we post-process the Qwen3-4B output with simple heuristics before evaluation (described in the appendix):
> - Too many stopwords (> 30% in a long enough tail)
> - Too long a tail (> 40 symbols)
> - No noun/pronoun in the tail for relations where a noun/pronoun is expected (using spaCy nlp tool).
>
> We got the following results:
>
> Table I1: FActScore* KG evaluation with Qwen3-32B as a judge: helper LLM ablation.
> **Bold**: better than Qwen3-32B baseline; ⭐: best in column.
>
> | KG type                               | Model size | #triples | Context only | Context + General truth |
> |---|---:|---:|---:|---:|
> | Qwen3-32B (baseline)                   | 32B        | 515,460  | 40.2 | 48.1 |
> | GraphMERT with Qwen3-32B as helper LLM | 80M        | 138,201  | **69.8** ⭐ | **72.2** ⭐ |
> | GraphMERT with Qwen3-4B as helper LLM  | 80M        | 9,000    | **61.5** | **62.0** |
>
>
> Table I2: ValidityScore evaluation with Qwen3-32B as a judge: helper LLM ablation. **Bold**: better than Qwen3-32B baseline; ⭐: best in column.
>
> | KG type                               | Model size | #triples | yes (ValidityScore) | maybe | no | missing |
> |---|---:|---:|---:|---:|---:|---:|
> | Qwen3-32B (baseline)                   | 32B        | 515,460  | 43.0 | 24.1 | 31.4 | 1.5 |
> | GraphMERT with Qwen3-32B as helper LLM | 80M        | 138,201  | **68.7** ⭐ | 18.3 | 10.8 | 2.2 |
> | GraphMERT with Qwen3-4B as helper LLM  | 80M        | 9,000    | **63.8** | 18.5 | 15.1 | 2.7 |
>
>
> While with the smaller model, scores drop, they are still much above those obtained by the Qwen3-32B LLM KG. We can attribute the drop to the better ability of Qwen3-32B to correctly construct longer sophisticated tails, whereas Qwen3-4B occasionally constructs non-grammatical long tails, e.g., *“fitbit - and of to fit the with three vs in two + versus min , compared using or mixed”*.
>
> For raw GraphMERT prediction examples, we include a comparison with GraphMERT trained with the MLM objective for illustration and include a small discussion.
>
> We hypothesize that if Qwen3-32B filtering was the main driving force, we’d expect LLM KG, which is purely based on Qwen3-32B extraction, to be on par with GrapMERT-extracted KG. In fact, Qwen3-32B KG scores lower in all our experiments.
>
> We also mention at the very end of Section 4.4 “GraphMERT Pipeline for Knowledge Graph Extraction” that removing the LLM-based combining-tokens step is a direction for future work.

---

> ### Author Response · Authors · 2025-12-19
> **2/n**
>
> C2
> > Please conduct a small-scale human evaluation (e.g., on 50-100 randomly sampled triples). Does human judgment align with the GPT-5/Qwen verdicts?
>
> We sent samples to five endocrinologists for evaluation. We included:
> 1. Sanity check with GPT-5 Thinking. Table E5: GraphMERT KG and Table C6: LLM KG (Qwen3-32B). That is 52 triples in all.
> 2. FActScore Verification with Qwen3-32B: 10 examples.
>
> Unfortunately, they informed us that it was difficult for them to comprehend the fine triple-level descriptions because these descriptions require extensive recall and reasoning. Thus, we would like to request the reviewer to allow us to defer this task to future work.

---

> ### Author Response · Authors · 2025-12-19
> **3/n**
>
> C3
> > Verify how GraphMERT compares to a traditional, fine-tuned BERT-based Relation Extraction (RE) model or a standard sequence-to-sequence extraction model (like T5-base fine-tuned on the triples).
>
> Our work explicitly targets low-label regimes for domains where large labeled datasets do not exist. The central design goal is to avoid constructing a large expert-annotated corpus in these domains by leveraging supervision with an existing small seed KG.
>
> However, both standard BERT RE and T5 triple-generation models would need a substantial data annotation effort in each new domain to function in the "raw domain-specific text + small seed KG" scenario.
>
> BERT-style RE is typically trained on sentences that contain mentions of both the head and tail entities, with explicit sentence-triple labels. We do not construct sentence-triple pairs. GraphMERT requires only the head to be present in the sequence, and it is enough for the training triple (including tail) to be semantically close to it. This means that the triple’s embedding is sufficiently close to the whole sequence embedding, regulated by the hyperparameter alpha (0.55 in our setting).
> Similarly, using T5 for multi-relation completion of the same head would require designing and training a multi-label decoding scheme over sentence-triple pairs. GraphMERT instead learns relation-specific embeddings and scores candidate tails for each relation independently, which helps us alleviate both the annotation and decoding burdens.
>
> Supervised BERT and T5 models are also sensitive to long-tailed label distributions in multi-relation extraction. In domain-specific texts, relation distributions are typically very imbalanced. Hence, training a robust T5/BERT model for our 28 relations would additionally require targeted oversampling and engineering for rare relations. In contrast, GraphMERT learns independent relation-specific embeddings. With that, it can exploit whatever data are available per relation without assuming a balanced label distribution, while it benefits from more data per relation when available.
>
> We precisely aim to minimize labeling a dataset in a new domain. For this reason, as the most appropriate baselines in our setting, we target LLM-based KG extraction methods, which can operate with minimal labeled supervision (through few-shot prompts).
>
> We appreciate the reviewer’s suggestion and now clarify the baseline choice in the revised version of the paper in Section 4.5. “LLM-generated KG” and rename this section as “LLM-generated KG as baseline”. We also add a clarification for the data requirement for BERT and sequence-to-sequence methods in Section 2.4.1 “Task-specific NLP Methods Scale Badly”.

---

> ### Author Response · Authors · 2025-12-19
> **4/n**
>
> > "Leafy Chain Graph".  Add a brief intuition on why this structure helps. Is the main benefit that it forces the model to attend to the "semantic" leaves using the same mechanism it uses for "syntactic" text? A sentence or two clarifying the intuition behind the "joint representation" would help.
>
> Yes, GraphMERT uses an integrated "Leafy Chain Graph" encoding that fuses syntax and semantics. It also allows training a separate embedding for each relation. We have added a clarification to Section 4.1 “Syntactic and Semantic Spaces: Merging Semantic Triples and Syntactic Text into a Unified Graphical Format”.
>
> Another relevant point is mentioned in Section 6.4.3 “Ablating GraphMERT Components”: “No H-GAT” ablation:
> In the “No H-GAT” ablation, we observe a large number of irrelevant tokens in the top-k predicted tokens, with commas and articles being primarily predicted in the top 3.

---

> ### Author Response · Authors · 2025-12-19
> **5/n**
>
> C5
> > The training data size is ~125M tokens (Table 1). This is quite small by modern standards. Please explicitly discuss in the Limitations section whether this approach scales well to larger corpora or if it is specifically designed for "low-resource, high-quality" scenarios.
>
> We intentionally start the process from the available amount of high-quality data and tailor the model size to that data, rather than target massive web-scale corpora. As shown in the paper, GraphMERT already works well on a relatively small training corpus. We primarily target scenarios where no large labeled training text corpus exists. Theoretically, GraphMERT can be scaled to larger corpora; verifying its scalability would require more GPU-hours than available to us and thus would be interesting future work. We now make this explicit in the Limitations section.

---

> ### Author Response · Authors · 2025-12-19
> **6/n**
>
> Broader Impact Concerns:
> > The authors have briefly touched upon reliability in medicine. However, a specific concern arises from the "Helper LLM" component. If the Helper LLM is a black-box commercial model or a large open model, it reintroduces the very "hallucination" and "lack of provenance" risks the paper argues against, just at a different stage of the pipeline (token combination rather than triple generation). I suggest adding a sentence in the Limitations or Broader Impact section acknowledging that while GraphMERT reduces hallucinations, the final assembly step still relies on a probabilistic model (the Helper LLM), which technically breaks the "guaranteed" chain of provenance, even if the risk is minimized.
>
> We have mentioned the risk of incomplete tails of the helper LLM in the Limitations section, and added the risk of hallucination introduced by the helper LLM.

---

### Review · Reviewer_vaak · 2025-12-06

**Summary Of Contributions:**

This paper introduces GraphMERT: a small encoder model that generates a Knowledge Graph (KG) from unstructured text. The KG is then used for performing various AI tasks. The paper then experiments on datasets such as FActScore in each case showing that use of KG outperforms the accuracy of base LLM. The paper also performs improvements on the KG starting from the original one obtained from the encoder model reporting its resulting accuracies. It proceeds to improve the KG over a few steps, at each step, reporting further improvements in the resulting accuracy. Based on these experiments, paper claims that GraphMERT leads to efficient, scalable, transparent, attributable, accountable, editable, auditable, and continually improvable state-of-the-art neurosymbolic AI.

**Audience:**

No

**Audience Explanation:**

No for this current version of the paper, in my view. But possibly yes, for a revised version of the paper.

The idea to include a set of useful keywords in the prompt to an LLM is not novel for the TMLR's audience.

Much of the paper is reviewing and explaining well known concepts such as Knowledge Graphs, Attention Networks, etc. Then it goes on to report the output of several models on a case study about kidney which might not be a clear example for many readers.

The output from Gemini Pro 2.5 without using the KG is quite brief and it is not clear whether a more detailed answer could have been obtained from this model without using the KG. At the end, it is not clear to me if there is any specific contribution in the method presented in the paper. If someone is specifically interested in a KG or a set of keywords, they can already include those keywords in their prompts to guide the LLMs. The encoder model itself is a simple method and there are already plenty of methods available in the literature for extracting KGs from unstructured text. Whether the KG leads to better accuracies, requires experiments on more advanced models, in my view.

Given the 29 pages preceding the experimental results in section 6, I was expecting a much stronger set of experiments. However, experiments are on only one dataset and one LLM (Qwen3-32B).

**Claims And Evidence:**

No

**Claims Explanation:**

### Weak experiments:

In my view, the paper’s method is interesting, yet, its experiments are weak, and it over claims its contributions.

The long list of adjectives that paper uses to describe the contributions of its model seems to me quite overblown (efficient, scalable, transparent, attributable, accountable, editable, auditable, and continually improvable). Many of these adjectives apply to LLMs, too. First and foremost paper needs to compare its contributions to the SoTA models.

For its motivation, paper reports experimenting with powerful models such as Grok on a topic related to kidney disease and it reports that those LLMs are not quite accurate in finding the relationships between the concepts related to kidney disease. However, when it comes to its own main experiments on FActScore, the paper does not compare its results to the accuracy of models such as Grok. In the best case, the paper achieves an accuracy of 76.9% on FActScore while it does not report that Grok 4.0 achieves an accuracy of 90% and Grok 4.1 achieves an accuracy of 97% on FActScore.


-------

### Weak literature review:

Extracting knowledge/representations from data and then leveraging that knowledge/representation is used all across the AI literature, but paper does not mention this clear enough. I give just two examples from a large number of papers that I am aware of.

Chen, C., Li, O., Tao, D., Barnett, A., Rudin, C. and Su, J.K., 2019. This looks like that: deep learning for interpretable image recognition. Advances in neural information processing systems, 32. – Extracts prototypes from the datasets and then processes the inputs by comparing the inputs against the prototypes.

Kadkhodaie, Z., Guth, F., Simoncelli, E.P. and Mallat, S., Generalization in diffusion models arises from geometry-adaptive harmonic representations. In The Twelfth International Conference on Learning Representations.


Similarly, extracting knowledge graphs from unstructured text is not a novel approach, neither leveraging the KG for boosting LLM accuracy. Below are only two papers that this paper has not cited/acknowledged.

Deng, Y., Ye, C., Huang, Z., Ma, M.D., Kou, Y. and Wang, W., 2024. Graphvis: Boosting llms with visual knowledge graph integration. Advances in Neural Information Processing Systems, 37, pp.67511-67534.

Liu, W., Zhou, P., Zhao, Z., Wang, Z., Ju, Q., Deng, H. and Wang, P., 2020, April. K-bert: Enabling language representation with knowledge graph. In Proceedings of the AAAI conference on artificial intelligence (Vol. 34, No. 03, pp. 2901-2908).


I suggest adding a new section in the paper to review the above two lines of research in the literature and thoroughly citing the prior work, not just the two example that I gave.


-------

The paper presents a case study about kidney which is too brief in my view. If paper has coauthors that are experts in kidney disease, that’s fine, otherwise, it would be best for a medical expert to be invited to review those claims and verify their correctness.


-------


Minor comment:

Some of the language in the paper does not read sound to me. For example, when paper says “symbolic components provide abstraction while neural components provide generalization”, I find it rather naive, or in the best case, vague, in terms of how it views generalization. Abstraction itself is a very broad concept that is hard to disentangle from generalization when it comes to the applications that paper studies.

**Requested Changes:**

1. I would like to see experiments on several more LLMs and also on more datasets. SoTA LLMs, open source and closed source, needs to be covered in the experiments.

2. Expanding the literature review and acknowledging what is already present in the recent AI literature and framing the contribution of this paper with respect to that literature..

3. The paper needs to be structured better and more efficiently making it easier to read by the community. All the main contributions and experimental results need to come in the main body of the paper while trivial explanations, details, and extended analysis can go into the appendix. 38 pages for the main body of the paper seems too long in my view. Paper can assume its readers already know what is a knowledge graph or they can easily read about it in other sources, and so on.

---

> ### Author Response · Authors · 2025-12-19
> **1/n**
>
> Thank you for the detailed review, and for taking the time to articulate both strengths and concerns. We have revised the paper accordingly.
>
> W1
> > The paper presents a case study about kidney which is too brief in my view. If the paper has coauthors that are experts in kidney disease, that’s fine, otherwise, it would be best for a medical expert to be invited to review those claims and verify their correctness.
>
> We sent samples to five endocrinologists for evaluation. We included:
> 1. Sanity check with GPT-5 Thinking. Table E5: GraphMERT KG and Table C6: LLM KG (Qwen3-32B). That is 52 triples in all.
> 2. FActScore Verification with Qwen3-32B: 10 examples
>
> Unfortunately, they informed us that it was difficult for them to comprehend the fine triple-level descriptions because these descriptions require extensive recall and reasoning. Thus, we would like to request the reviewer to allow us to defer this task to future work.
>
> The kidney case is intended as an illustrative example showing that even large general-purpose LLMs may hallucinate in safety-critical, domain-specific settings. Our goal here is to highlight this issue, not to provide a comprehensive evaluation of LLM capabilities.

---

> ### Author Response · Authors · 2025-12-19
> **2/n**
>
> W2
> > Some of the language in the paper does not seem sound to me. “symbolic components provide abstraction while neural components provide generalization”, I find it rather naive, or in the best case, vague, in terms of how it views generalization. Abstraction itself is a very broad concept that is hard to disentangle from generalization when it comes to the applications that paper studies.
>
> We have removed this phrase from the abstract.

---

> ### Author Response · Authors · 2025-12-19
> **3/n**
>
> W3
>
> > The idea to include a set of useful keywords in the prompt to an LLM is not novel for the TMLR's audience. At the end, it is not clear to me if there is any specific contribution in the method presented in the paper. If someone is specifically interested in a KG or a set of keywords, they can already include those keywords in their prompts to guide the LLMs. The encoder model itself is a simple method and there are already plenty of methods available in the literature for extracting KGs from unstructured text. Whether the KG leads to better accuracies, requires experiments on more advanced models, in my view.
>
> We agree that adding keywords to prompts is a standard technique.
>
> While keyword-based prompting itself is not novel, the question of how to select relevant and faithful keywords in texts that contain linguistically misleading cues remains open.
>
> GraphMERT uses a unified semantic + syntactic input representation and relation-specific embeddings to select candidate tails that are consistent with both the local text and the seed KG ontology, rather than simply reuse surface tokens from the passage. This is exactly why, for the kidney example, GraphMERT’s top predictions concentrate on kidney-related concepts (e.g., kidney, kidney structure) and suppress distractors (“cerebral gray matter”).
>
> What GraphMERT adds is a way to obtain faithful concepts from text without heavy sentence-level annotation (as required by traditional BERT pipelines), while remaining consistent with the domain-specific ontology that LLMs often struggle to follow.

---

> ### Author Response · Authors · 2025-12-19
> **4/n**
>
> C1
> >I would like to see experiments on several more LLMs and also on more datasets.
>
> >First and foremost, the paper needs to compare its contributions to the SoTA models. Paper reports experimenting with powerful models such as Grok on a topic related to kidney disease and it reports that those LLMs are not quite accurate in finding the relationships between the concepts related to kidney disease. The paper does not compare its results to the accuracy of models such as Grok. In the best case, the paper achieves an accuracy of 76.9% on FActScore while it does not report that Grok 4.0 achieves an accuracy of 90% and Grok 4.1 achieves an accuracy of 97% on FActScore.
>
> **1. SoTA models**
>
> We extracted an LLM KG with two extra LLMs: Grok 4.0 (grok-4-fast-reasoning) and Qwen3-14B and evaluated their FActScore and ValidityScore with Qwen3-32B and Gemini 2.0 flash.
> We additionally ran FActScore and ValidityScore evaluations with Gemini 2.0 flash for the Qwen3-14B LLM KG and GraphMERT KG.
>
> We obtained the following results (added to the paper):
>
> Table 5: FActScore* KG evaluation (in percentage points).
> **Bold**: better than Qwen3-32B baseline; ⭐: best in column. Percentages are computed per KG type with repsect to its own #triples.
>
> | KG type              | Model size | #triples | Qwen3-32B as a judge: Context only | Qwen3-32B as a judge: Context + General truth | Gemini 2.0 Flash as a judge: Context only | Gemini 2.0 Flash as a judge: Context + General truth |
> |---|---:|---:|---:|---:|---:|---:|
> | Qwen3-32B (baseline) | 32B        | 515,460  | 40.2 | 48.1 | 49.1 | 44.3 |
> | GraphMERT (ours) | 80M        | 138,201  | **69.8** | **72.2** | **79.7** ⭐ | **84.0** ⭐ |
> | Grok 4 Fast      | 1.7T       | 100,000  | **79.4** ⭐ | **79.7** ⭐ | **73.3** | **67.9** |
> | Qwen3-14B        | 14B        | 100,000  | **46.4** | 45.7 | **62.4** | **57.6** |
>
>
> Table 7: ValidityScore evaluation (in percentage points). **Bold**: better than Qwen3-32B baseline; ⭐: best in column. Percentages are computed per KG type with respect to its own #triples.
>
> | KG type              | Model size | #triples | Qwen3-32B as a judge: yes (Validity-Score) | Qwen3-32B as a judge: maybe | Qwen3-32B as a judge: no | Qwen3-32B as a judge: missing | Gemini 2.0 Flash as a judge: yes (Validity-Score) | Gemini 2.0 Flash as a judge: maybe | Gemini 2.0 Flash as a judge: no | Gemini 2.0 Flash as a judge: missing |
> |---|---:|---:|---:|---:|---:|---:|---:|---:|---:|---:|
> | Qwen3-32B (baseline) | 32B        | 515,460  | 43.0 | 24.1 | 31.4 | 1.5 | 66.0 | 33.6 | 0.3 | < 0.1 |
> | UMLS Seed KG         | N/A        | 28,533   | **53.4** | 10.0 | 34.7 | 1.9 | **82.5** ⭐ | 16.5 | 1.1 | < 0.1 |
> | GraphMERT (ours) | 80M        | 138,201  | **68.7** ⭐ | 18.3 | 10.8 | 2.2 | **81.3** | 18.5 | 0.2 | < 0.1 |
> | Grok 4 Fast          | 1.7T       | 100,000  | **56.9** | 21.9 | 21.2 | < 0.1 | **68.8** | 30.9 | 0.3 | < 0.1 |
> | Qwen3-14B            | 14B        | 100,000  | 31.8 | 21.9 | 46.2 | 0.1 | 53.4 | 45.7 | 0.9 | < 0.1 |
>
> Grok 4.0 achieved 79.7% FActScore with Qwen3-32B as a judge and 67.9% with Gemini 2.0 Flash as a judge.
> The 90% FActScore achieved by Grok 4.0 refers to general-domain factuality benchmarks. However, it doesn’t account for relational knowledge in the form of semantic triples.
> In our setting, factual correctness requires not only identifying medically relevant entities but also assigning the appropriate semantic relation (e.g., cause_of vs. associated_with) as implied between these entities in the target sequence.
> General-purpose LLMs, while being great at discovering entities, tend to perform worse under relation-level evaluation in a narrow domain (medical). This especially negatively impacts their ValidityScore, which reflects the ability to assign a semantically valid relation.
>
> **2. Datasets**
>
> We reported results on other standard medical QA benchmarks in Appendix D to give a sense of downstream utility. However, our primary goal is to assess the validity and factuality of the extracted KG triples themselves, not to optimize for broad QA coverage.
>
> Our model is trained on narrow, diabetes-centric corpora, whereas most popular medical QA benchmarks target broad, multi-specialty medicine. Even when questions mention diabetes, they often require skills that are outside our target use case (e.g., dosage calculations, multi-system differential diagnosis, or cross-specialty knowledge), and are not well aligned with the textual distributions we train on. For this reason, we chose one benchmark with a dedicated endocrinology subset for our GraphRAG evaluation.
>
> A more exhaustive QA evaluation across multiple benchmarks would require designing and tuning a graph-based retrieval and reasoning layer over our multi-relational schema (e.g., how to weight different relations, how far to expand the subgraph). We view this as an orthogonal systems problem and leave it for future work, focusing here on the validity and factuality of the extracted KG itself.

---

> ### Author Response · Authors · 2025-12-19
> **5/n**
>
> C2
> > Expanding the literature review and acknowledging what is already present in the recent AI literature and framing the contribution of this paper with respect to that literature.
>
> > Extracting knowledge/representations from data and then leveraging that knowledge/representation is used all across the AI literature, but the paper does not mention this clearly enough. I give just two examples from a large number of papers that I am aware of.
> I suggest adding a new section in the paper to review the above two lines of research in the literature and thoroughly citing the prior work, not just the two examples that I gave.
>
> 1. We agree that the “extract-then-utilize” paradigm is widely adopted across AI and was not sufficiently introduced in the original draft. In response, we have added a new “Structured Representation Learning” Section 2.1 in the Background section to explicitly situate our work within the broader literature on extracting structured knowledge or representations from data and subsequently leveraging them for downstream reasoning and generalization.
>
> 2. In addition, Section 2.4 has been revised to more clearly position our method relative to existing KG extraction approaches from unstructured text. While leveraging KGs to directly boost LLM accuracy is not the primary focus of this paper, we acknowledge its importance in the literature. To provide appropriate context, we have added a brief review of representative KG-enhanced LLM methods in Section 2.6.

---

> ### Author Response · Authors · 2025-12-19
> **6/n**
>
> C3
> > All the main contributions and experimental results need to come in the main body of the paper while trivial explanations, details, and extended analysis can go into the appendix.
>
> We have revised the introduction, background, and motivational example sections to ensure that all main contributions and key experimental results are presented in the main body, while secondary explanations and extended analyses are moved to the appendix. Specifically, we streamlined the introduction by reducing the discussion on the limitations of LLMs for KG extraction, as this topic is covered in greater detail in the background section. We also relocated less central discussions related to KGs to the appendix. In addition, we simplified the motivational example by retaining only the Gemini 2.5 Pro results in the main text, while moving results from other LLMs to the appendix to improve clarity and readability.

---

> > ### Comment · Reviewer_vaak · 2025-12-23
> >
> > I thank the authors for their detailed response, additional experiments, and the improvements in the paper. I adjust my recommendation.

---

### Review · Reviewer_VVhw · 2025-12-06

**Summary Of Contributions:**

**Summary**\
This paper introduces a compact encoder-only model GraphMERT, which is designed to extract structured knowledge directly from domain-specific text with mixing syntactic sequences and semantic relational cues. GraphMERT employs leafy chain graphs and H-GAT to fuse syntactic-semantic information and then trained with the objectives MLM + MNM to learn both contextual and structural signals. KG results show strong validity and reliability, significantly outperforming LLM-generated KGs in FActScore and ValidityScore.

&nbsp;

**Strengths**
1. The problem setup is clear and timely. The paper effectively motivates the need to automatically derive reliable KGs from text corpora by highlighting limitations of LLM-based KG construction, including reliability issues and potential biases in triplet generation.
2. The overall narrative flows well, and the related work section is well organized. The background and prior research are connected in a way that makes the method easy to follow.
3. GraphMERT shows strong empirical gains over LLM-generated KGs (e.g., FActScore and ValidityScore), demonstrating the benefits of the structured modeling approach.

&nbsp;

**Weaknesses**\
I'm not very familiar with this specific research area, such as KG construction or graph-based Transformer architecture. Therefore, it is difficult for me to precisely assess the novelty of the proposed method relative to existing work.
1. From my perspective, Some components appear somewhat close to prior work (e.g., the span masking resembles SpanBERT, and the attention decay idea feels similar to GradFormer). However, this impression may be due to my limited familiarity with this area.
2. At times, the paper presents KGs as if they are universally beneficial (e.g., in Section 2.2.3 on reliability in critical fields). The claims could be toned down slightly to maintain balance.
3. This is a minor comment, but the transition in the related work section feels slightly abrupt: Sections 2.1–2.6 provide general background, while Section 2.7 begins describing elements of the proposed method. These portions might fit more naturally within a unified preliminaries or method section, though this is not a major issue.
4. While the method is described as domain-agnostic, the evaluation is limited to a single domain, where seed KG size and vocabulary diversity may not reflect other settings. Including an additional domain in the experiments would provide clearer evidence of the method’s generalizability.
5. Because the final triplet relies on a helper LLM, the quality of the resulting relations may be sensitive to the helper model’s domain knowledge. If the helper LLM lacks sufficient expertise in the target domain, it may struggle to construct accurate or meaningful relations, potentially limiting the reliability of the extracted KG.

**Audience:**

Yes

**Audience Explanation:**

I believe that some members of TMLR’s audience would find the paper’s contributions interesting, particularly those working on knowledge graph construction or structured representations. That said, the paper is somewhat long in its current form, and presenting in a more concise and focused manner could make the work even more approachable and engaging for readers.

**Claims And Evidence:**

Yes

**Claims Explanation:**

Although some aspects of the syntactic–semantic modeling are conceptually abstract, the paper presents clear and convincing qualitative and quantitative results, along with informative ablations, that effectively support its main claims.

**Requested Changes:**

1. It may be helpful to moderate some of the broader claims regarding the universal advantages of KGs to present a more balanced perspective.
2. As a minor suggestion, refining the structure of the related work and preliminaries to create a smoother transition could improve readability.
3. Extending the experimental evaluation beyond a single domain would provide a stronger assessment of the method’s generalizability and overall impact.

---

> ### Author Response · Authors · 2025-12-19
> **1/n**
>
> We thank the reviewer for the careful and constructive assessment of our work, as well as for the summary of our contributions and the empirical findings.
>
> W1
> > Some components appear somewhat close to prior work (e.g., the span masking resembles SpanBERT, and the attention decay idea feels similar to GradFormer).
>
> We appreciate the reviewer’s observation.
>
> Our goal is not to claim novelty for span masking or distance-based attention themselves, but to integrate these components into a unified text-KG encoder with leaf injection, relation-specific embeddings, and an MNM objective.
>
> For efficient relation-embedding training under MNM, we replace SpanBERT’s geometric span sampling on arbitrary tokens with masking entire leaf spans, so that gradients propagate through the relation embeddings from each injected leaf.
> Likewise, while GradFormer’s distance-based decay was tested on molecular graphs, we adapt the idea to language sequences by applying a square-root decay to avoid oversuppressing attention to relevant but distant leaf nodes.

---

> ### Author Response · Authors · 2025-12-19
> **2/n**
>
> W2
> > At times, the paper presents KGs as if they are universally beneficial (e.g., in Section 2.2.3 on reliability in critical fields). The claims could be toned down slightly to maintain balance.
>
> We have addressed this (please see our reply to C1)
>
> W3
> > The transition in the related work section feels slightly abrupt: Sections 2.1-2.6 provide general background, while Section 2.7 begins describing elements of the proposed method. These portions might fit more naturally within a unified preliminaries or method section, though this is not a major issue.
>
> We have addressed this (please see reply to C2)
>
> W4
> > While the method is described as domain-agnostic, the evaluation is limited to a single domain, where seed KG size and vocabulary diversity may not reflect other settings. Including an additional domain in the experiments would provide clearer evidence of the method’s generalizability.
>
> Please see our response to C3.
>
> W5
> > Because the final triplet relies on a helper LLM, the quality of the resulting relations may be sensitive to the helper model’s domain knowledge. If the helper LLM lacks sufficient expertise in the target domain, it may struggle to construct accurate or meaningful relations, potentially limiting the reliability of the extracted KG.
>
> We have acknowledged this in the Limitations section.

---

> ### Author Response · Authors · 2025-12-19
> **3/n**
>
> C1
> > It may be helpful to moderate some of the broader claims regarding the universal advantages of KGs to present a more balanced perspective.
>
> We have
> - deleted several paragraphs from former Section 2.2.3: KG Applications in Neurosymbolic Frameworks: RAG, RL, NN Interpretability, and moved the rest to Appendix A;
> - revised Section 2.3: KGs for Domain-specific Superintelligence (now in Appendix A);
> - moderated the paragraph on “Reliability in critical fields” (now it is a part of Section 2.3 “The Representation Dilemma: Neural or Symbolic?”)

---

> ### Author Response · Authors · 2025-12-19
> **4/n**
>
> C2
> > Refining the structure of the related work and preliminaries to create a smoother transition could improve readability.
>
> We agree that restructuring the paper can improve the flow and make it more understandable for the reader. We have moved “Hierarchical Graph Attention Networks” to GraphMERT Architecture and Training. We have joined Section “Graph Transformer Architecture” with Section 2.7: Graph Attention Networks.

---

> ### Author Response · Authors · 2025-12-19
> **5/n**
>
> C3
> > Extending the experimental evaluation beyond a single domain would provide a stronger assessment of the method’s generalizability and overall impact.
>
> We agree that extending the evaluation beyond a single domain would further strengthen claims about generalizability.
> Conceptually, the GraphMERT pipeline is domain-agnostic, but extending it to additional domains is a time-heavy proposition: for a new domain, we must collect a domain-specific corpus, find a high-quality seed KG, and select relations.
> This is because we deliberately focus on domains where no labeled training datasets exist, but a small seed KG is available. For this reason, extracting a KG in a new domain takes time. At the same time, we hope that in this setup, our study unlocks the opportunity to create a KG in other domains where users have data and a small seed KG.
>
> Doing another KG extraction in another new domain is essentially a full pipeline replication per domain and is beyond the scope of the current submission. We explicitly list multi-domain extensions and evaluation as a direction for future work.

---

### Review · Reviewer_Lui9 · 2025-12-13

**Summary Of Contributions:**

The paper introduces GraphMERT, a compact encoder-only model designed to extract reliable knowledge graphs (KGs) from unstructured text. By combining neural abstraction with symbolic reasoning, GraphMERT addresses the scalability and interpretability challenges of neurosymbolic AI.  A key contribution is the joint learning of semantic signals derived from a seed KG and syntactic signals from unstructured text. This is achieved through a proposed masked node modeling.

The authors present experiments on medical texts (e.g., PubMed diabetes papers)  to compare the proposed method with existing LLM-based models in accuracy and validity.

**Audience:**

Yes

**Audience Explanation:**

The paper is of sufficient interest to some members of TMLR, especially those working on the knowledge graph.

**Claims And Evidence:**

Yes

**Claims Explanation:**

## *Strenghts*
The authors provide a clear and well-structured formulation of the problem, with sufficient motivation and a thoughtful positioning of their work in relation to existing literature. The connection to knowledge graphs (KGs) is articulated effectively, highlighting their relevance to the proposed approach. As I am not an expert in this specific domain, I cannot fully assess the completeness of the citations or the novelty of the contribution. Nevertheless, the paper is well organized, flows smoothly, and is accessible to readers.

The empirical evaluations seem to show a clear advantage of the proposed work (GraphMERT) compared to existing LLM-based methods. The adoption of metrics such as FActScore and ValidityScore appears appropriate and well-justified.


## *Weakness*
My main concern regarding this work is that it relies on a helper LLM that introduces a potential limitation. The accuracy of the extracted relations is contingent on the helper model’s domain expertise. In cases where the helper LLM lacks sufficient knowledge of the target domain, the results may be incomplete or erroneous, thereby reducing the overall reliability of the proposed method based on the knowledge graph.

**Requested Changes:**

1. A discussion on the limitations of the proposed model, especially when the helper LLM lacks sufficient knowledge of the target domain.
2. Approaches to help mitigate the issue of a bad LLM helper.

---

> ### Author Response · Authors · 2025-12-19
> **1/n**
>
> We thank the reviewer for the careful summary of our contributions and for highlighting both the strengths of the formulation and evaluation, as well as the important limitation around the helper LLM’s domain expertise.
>
> C1
> > A discussion on the limitations of the proposed model, especially when the helper LLM lacks sufficient knowledge of the target domain.
>
> We agree and have added statements in the limitation section that make this explicit.

---

> > ### Author Response · Authors · 2025-12-19
> >
> > C2
> > > Approaches to help mitigate the issue of a bad LLM helper.
> >
> > We acknowledge this limitation and, in the revised paper, outline practical mitigations in the Limitations section: (i) using domain-specialized helper models when available, (ii) applying simple non-LLM post-filters (e.g., ontology and part-of-speech checks), and (iii) retaining only triples that pass a separate ValidityCheck evaluation.
> >
> > A more systematic study on how to exclude LLM helper and augment GraphMERT to multi-token prediction is left to future work.

---

### Decision · Action_Editor_5khC · 2026-01-17

**Recommendation:** Accept as is

**Audience:**

Yes

**Audience Explanation:**

The task this paper aims to solve is closely related the machine learning community.

**Claims And Evidence:**

Yes

**Claims Explanation:**

The authors have already provided accurate, convincing and clear evidence for their claims.